# SAMPLE-EFFICIENT PRUNING MODEL SELECTION VIA LASSO

## ABSTRACT

We study the problem of selecting a pruned neural network from a set of candidates generated by various pruning methods. The goal of a learner is to identify a near-optimal model that achieves low generalization error. Although model selection techniques such as cross-validation are widely used in practice, they often fail to provide guarantees on generalization error or offer only asymptotic guarantees. To address these limitations, we propose an algorithm that jointly selects a pruned network and updates its parameters using an $L_1$-regularization, thereby encouraging sparsity while ensuring low generalization error. For a given error tolerance $\epsilon$, we establish a sample complexity lower bound of $\Omega\left(\frac{1}{\epsilon^2}\log M\right)$, where $M$ is the number of candidate models, demonstrating that our algorithm remains sample-efficient even when the candidate pool is large. Extensive numerical experiments confirm both the practical effectiveness and the theoretical guarantees of the proposed method.

## 1 INTRODUCTION

Deep neural networks (He et al., 2016; Vaswani et al., 2017; Brown et al., 2020) have become indispensable in modern machine learning, but their large parameter counts and high computational demands create challenges for efficient deployment. Consequently, the growing emphasis on reducing resource demands is fueling interest in neural network pruning. By substantially reducing model size, pruning facilitates cost-effective storage and deployment, making it particularly advantageous for businesses and organizations employing the neural network for inference (Gale et al., 2019; Benbaki et al., 2023). Nevertheless, given the variety of pruning techniques, a central question remains: *how can we systematically select the best pruned model for deployment from a pool of candidates?*

A straightforward solution is to rely on empirical selection methods such as *Cross-Validation* (CV) (Allen, 1974; Geisser, 1975; Stone, 1977). While these methods are widely used in practice, they suffer from a crucial limitation: *their guarantees are asymptotic*. In finite-sample regimes, CV can fail to ensure that the selected model achieves near-optimal generalization performance. This shortcoming is particularly problematic when data collection or computation is costly, since it leaves open the risk of choosing a suboptimal model without a rigorous performance guarantee.

To address this challenge, we conceptualize a pruned neural network as a combination of (i) a representation—the feature map induced by the frozen, pruned hidden layers—and (ii) a last-layer weight vector that linearly combines those features, as in recent representation selection literature (You et al., 2019; Bolya et al., 2021; Shao et al., 2022; Yang et al., 2024). Under this formulation, the pruned neural network selection problem naturally reduces to a feature map–based model selection problem: given a set of candidate feature maps derived from different pruned networks, the learner must identify the optimal representation and learn its corresponding inference head to minimize generalization error.

As there is no formal definition of generalizability in the context of model selection involving multiple models, we introduce the *inter-model excess risk*, which is defined as the difference between the generalization error of a candidate and that of the best model in the pool. This metric captures the essence of model selection: *the chosen network must not only generalize well on its own but also be competitive relative to all alternatives.*

Based on the newly introduced inter-model excess risk measure, we propose an *elimination-based selection algorithm* that first filters out suboptimal models and then identifies a near-optimal one from the remaining candidates. In the elimination step, suboptimal models are iteratively removed using confidence bounds on their risks, ensuring computational efficiency while retaining the best model with high probability. In the subsequent selection step, the last-layer weights of the surviving candidates are updated via $L_1$-regularized loss minimization, promoting additional sparsity. As a result, the algorithm selects either the true best model or one that is empirically superior based on the sampled data. Given a sufficient finite number of samples, the algorithm guarantees the identification of a near-optimal pruned network in terms of generalizability.

Our main contributions are as follows:

- We propose a novel pruning model selection framework that identifies a near-optimal pruned neural network having low generalization error. The proposed algorithm first filters out suboptimal models based on the confidence bounds on their risk, and selects the model with the smallest regularized empirical risk. The algorithm not only identifies a pruned neural network that generalizes well to unseen data, but also improves its sparsity rate without compromising its performance.

- To evaluate generalizability in the context of multi-model classes, we introduce *inter-model excess risk* as an extension of the standard excess risk (Bach, 2024). We show that selecting the model with the minimal inter-model excess risk ensures that it has acquired parameters conducive to generalization among the candidates in the pool. Furthermore, we establish a sample complexity lower bound of $\Omega\left(\frac{1}{\epsilon^2} \log M\right)$, where $M$ is the number of candidate models and $\epsilon$ is an error tolerance. This result implies that the proposed algorithm remains sample-efficient even when $M$ is exponentially large. Compared to existing model selection methods such as CV, our approach offers both finite-sample guarantees and improved computational efficiency.

- We extensively evaluate our algorithms through experiments and show that (1) the proposed algorithm efficiently identifies a pruned network having low generalization error, and (2) the model selected by the algorithm shows higher sparsity rate without deterioration in its performance.

## 2 PRELIMINARIES

### 2.1 NOTATIONS

For a positive integer $N$, $[N] := \{1, \ldots, N\}$ is the index set up to it. For a vector $\mathbf{v} \in \mathbb{R}^d$, $[\mathbf{v}]_j$ represents its $j$-th component. For any index set $I \subseteq [d]$, the vector filtered by $I$ is $\mathbf{v}_I := ([\mathbf{v}]_{1,I}, \ldots, [\mathbf{v}]_{d,I})^\top$, where for all $j \in [d]$, $[\mathbf{v}]_{j,I} := [\mathbf{v}]_j \mathbb{1}\{j \in I\}$. For a matrix $\mathbf{V} \in \mathbb{R}^{m \times n}$, $\mathbf{V}(i, j)$ is the entry of the $i$-th row and $j$-th column.

### 2.2 PROBLEM SETUP

**Pruned neural network selection.** We consider a pruned neural network selection problem. Let $\{f_{\mathbf{W}_m}(\cdot) : \mathcal{X} \to \mathbb{R}^{d_y}\}_{m \in [M]}$ be a set of pruned neural networks where $\mathbf{W}_m$ represents the remaining parameters of the $m$-th network after pruning. Let $\mathbf{W}_m = \{\mathbf{w}_{m,\text{repr}}, \mathbf{w}_m\}$, where $\mathbf{w}_{m,\text{repr}}$ is the parameters of all of the frozen hidden layers and $\mathbf{w}_m \in \mathbb{R}^{d_m}$ denotes the parameters of the last layer. Then, we conceptualize each pruned neural network as a function of the form $f_{\mathbf{W}_m}(\mathbf{x}) = \phi_m(\mathbf{x})^\top \mathbf{w}_m$ where $\phi_m := \phi_{\mathbf{w}_{m,\text{repr}}} : \mathcal{X} \to \mathbb{R}^{d_m}$ denotes the feature extractor induced by the pruned hidden layers, and $\mathbf{w}_m$ corresponds to the last-layer weight vector. From this standpoint, we consider the pruned neural network selection problem as a model selection problem where, a learner must identify an optimal model from a set of candidate models $\{\phi_m(\cdot)^\top \mathbf{w}_m\}_{m \in [M]}$ using *i.i.d.* samples $\{(\mathbf{x}_i, y_i)\}_{i=1}^n$ of size $n$.

**Inter-model excess risk.** However, to claim that one model is more optimal than another, we must first establish a proper comparison criterion. Since the goal is not merely to identify the best model for the given sample, but rather to find the model that performs best with respect to the entire

data distribution, it is natural to adopt generalizability (Definition 9.3 in Hazan et al. (2016)) as the central criterion for comparison. As there is no formal definition of generalizability in the context of model selection involving multiple models, we introduce the *inter-model excess risk*, which not only enables an $(\epsilon, \delta)$-PAC framework across multiple model classes, but also recovers the standard excess risk (Bach, 2024) in the case where there is only a single candidate model. Before introducing the definition of inter-model excess risk, let us first revisit the some standard definitions such as excess risk and generalization error commonly used in the ERM literature.

Let $\mathcal{X} \subset \mathbb{R}^{d_x}$ be the set of inputs, $\mathcal{Y} \subset \mathbb{R}^{d_y}$ be the set of labels, and $\mathcal{P}$ be an unknown distribution over $(\mathcal{X}, \mathcal{Y})$. Let $\{\phi_m(\cdot)^\top \mathbf{w}_m\}_{m \in [M]}$ be the set of candidate models, where $\phi_m : \mathcal{X} \to \mathbb{R}^{d_m}$ and $\mathbf{w}_m \in \mathbb{R}^{d_m \times d_y}$, and let $\ell(\cdot, \cdot)$ be a loss function. For the sake of simplicity in notation, we denote a model with its index, such as referring to model $\phi_m(\cdot)^\top \mathbf{w}_m$ as simply model $m$. First, we define the risk and the optimal parameter of the model $m$ as follows:

**Definition 1** (Risk & Optimal parameter). *The risk of the model $m \in [M]$ is $\mu(m, \mathbf{w}_m) := \mathbb{E}_{(\mathbf{x}, \mathbf{y}) \sim \mathcal{P}}[\ell(\phi_m(\mathbf{x})^\top \mathbf{w}_m, \mathbf{y})]$, the expectation of the loss $\ell$ over $\mathcal{P}$. The optimal parameter of the model $m$ is $\mathbf{w}_m^* := \mathrm{argmin}_{\mathbf{w}} \mu(m, \mathbf{w})$, the parameter minimizing its risk.*

Similarly, given a set of *i.i.d.* samples $\{(\mathbf{x}_i, \mathbf{y}_i)\}_{i=1}^n$ drawn from $\mathcal{P}$, we define the empirical risk and the corresponding empirical estimator as follows:

**Definition 2** (Empirical Risk & Estimator). *The empirical risk of the model $m$ is $\hat{\mu}_n(m, \mathbf{w}_m) := \frac{1}{n} \sum_{i=1}^n \ell(\phi_m(\mathbf{x}_i)^\top \mathbf{w}_m, \mathbf{y}_i)$, the average of its loss over the samples. An estimator of the model $m$ is $\hat{\mathbf{w}}_m^n := \mathrm{argmin}_{\mathbf{w}} \hat{\mu}_n(m, \mathbf{w})$, the parameter minimizing the empirical risk.*

Then, the generalization error and excess risk of the model $m$ is defined as follows:

**Definition 3** (Generalization Error & Excess Risk). *The generalization error of the model $m$ with respect to its estimator $\hat{\mathbf{w}}_m^n$ is $\mu(m, \hat{\mathbf{w}}_m^n)$, the expectation of its loss over $\mathcal{P}$ when it has its estimator $\hat{\mathbf{w}}_m^n$. The excess risk of the model $m$ is $\mu(m, \hat{\mathbf{w}}_m^n) - \mu(m, \mathbf{w}_m^*)$, the difference between the generalization error with respect to estimator and the risk with its optimal parameter.*

When considering only a single model, excess risk serves as a standard measure for evaluating the generalizability of an estimator. In particular, a smaller excess risk indicates that the estimator is more robust to unseen data. For a given tolerance level $\epsilon$, the number of samples required to ensure that the excess risk falls below $\epsilon$ reflects the effectiveness of the estimation method, and the sample complexity of various estimators has been extensively studied in the ERM literature. However, in the multi-model setting, excess risk is not suitable for comparing generalizability across different models. In other words, *a smaller excess risk does not necessarily imply that the corresponding model is more generalizable.* For example, consider two models, $m_1$ and $m_2$, with corresponding estimators $\hat{\mathbf{w}}_{m_1}^n$ and $\hat{\mathbf{w}}_{m_2}^n$. Suppose that the excess risk of $m_1$ is smaller than that of $m_2$. Nevertheless, if the ordering of the generalization error and the minimum risk satisfies $\mu(m_1, \hat{\mathbf{w}}_{m_1}^n) \geq \mu(m_1, \mathbf{w}_{m_1}^*) \gg \mu(m_2, \hat{\mathbf{w}}_{m_2}^n) \geq \mu(m_2, \mathbf{w}_{m_2}^*)$, then model $m_2$ is actually more robust to the underlying data distribution, despite its higher excess risk.

This example highlights the necessity of comparing estimators relative to the best achievable risk across models. That is, each model's generalization should be measured against the smallest minimum risk attainable within the candidate pool. This motivates the need for a new metric—*inter-model excess risk*—which we formally define as follows:

**Definition 4** (Inter-Model Excess Risk). *Let us denote the best model $m^* := \mathrm{argmin}_{m \in [M]} \mu(m, \mathbf{w}_m^*)$, the model with the minimum risk when each model has its optimal parameter. Then, the inter-model excess risk of the model $m$ with respect to its estimator $\hat{\mathbf{w}}_m^n$ is $\mathcal{E}(m, \hat{\mathbf{w}}_m^n) := \mu(m, \hat{\mathbf{w}}_m^n) - \mu(m^*, \mathbf{w}_{m^*}^*)$, the difference between the generalization error of the model $m$ and the minimum risk of the best model.*

In contrast to the excess risk of an individual model, the inter-model excess risk compares the generalization error against the minimum risk attained by the best model. Thus, *the selection of a model characterized by low inter-model excess risk not only ensures that the chosen model has learned well generalized parameters but also signifies that the selected model itself is adept at handling unseen data compared to other models in the candidate pool.* Also, we would like to note that the inter-model excess risk recovers the standard excess risk in the case where there is only a single candidate model.

---

**Algorithm 1** Elimination-based Pruning Model Selection via Lasso (`L1Sel`)

---

1: **Inputs:** Pool of candidates $\{\phi_m(\cdot)^\top \mathbf{w}_m\}_{m \in [M]}$, sample size $N$, number of epochs $H$
2: Sample $\{(\mathbf{x}_i, \mathbf{y}_i)\}_{i=1}^N \sim \mathcal{P}$, set epoch lengths $n_1 < \cdots < n_{H-1} < n_H = N$, regularization parameters $\{\lambda_{n_h}\}_{h=1}^H$ and confidence radii $\{\alpha_{n_h}\}_{h=1}^H$
3: Initialize $\mathcal{M}_1 \leftarrow [M]$
4: **for** $h = 1, \ldots, H-1$ **do**                                      ▷ *Elimination Step*
5:     $\mathbf{w}_m^{n_h} = \arg\min_{\mathbf{w}} \hat{\mu}_{n_h}(m, \mathbf{w}) + \lambda_{n_h} L_1(\mathbf{w}), \quad \forall m \in \mathcal{M}_h$
6:     $\mathrm{ucb}(m, h) := \hat{\mu}_{n_h}(m, \mathbf{w}_m^{n_h}) + \alpha_{n_h}, \mathrm{lcb}(m, h) := \hat{\mu}_{n_h}(m, \mathbf{w}_m^{n_h}) - \alpha_{n_h}, \quad \forall m \in \mathcal{M}_h$
7:     $\hat{m}_h := \arg\min_{m \in \mathcal{M}_h} \mathrm{ucb}(m, h)$ and $\tilde{\mathcal{M}}_h := \{m \in \mathcal{M}_h : \mathrm{lcb}(m, h) > \mathrm{ucb}(\hat{m}_h, h)\}$
8:     $\mathcal{M}_{h+1} \leftarrow \mathcal{M}_h \setminus \tilde{\mathcal{M}}_h$
9: **end for**
10: **for all** $m \in \mathcal{M}_H$ **do**                                  ▷ *Selection Step*
11:     $\mathbf{w}_m^N = \arg\min_{\mathbf{w}} \hat{\mu}_N(m, \mathbf{w}) + \lambda_N L_1(\mathbf{w})$
12: **end for**
13: $m^N = \arg\min_{m \in \mathcal{M}_H} \hat{\mu}_N(m, \mathbf{w}_m^N) + \lambda_N L_1(\mathbf{w}_m^N)$
14: **Return:** $\phi_{m^N}(\cdot)^\top \mathbf{w}_{m^N}^N$

---

Our objective is to design an algorithm $\mathcal{A}$ and verify its required sample size $n$ such that, for given $\epsilon$ and $\delta \in (0, 1)$, the model $m(\mathcal{A}) \in [M]$ selected by the algorithm, along with its estimator $\hat{\mathbf{w}}_{m(\mathcal{A})}^n$ satisfies the following PAC guarantee: with probability at least $1 - \delta$, the inter-model excess risk is less than $\epsilon$, i.e., $\mathbb{P}(\mathcal{E}(m(\mathcal{A}), \hat{\mathbf{w}}_{m(\mathcal{A})}^n) > \epsilon) \leq \delta$.

## 3 MAIN RESULTS

### 3.1 ALGORITHM

In this section, we introduce an algorithm, referred to as `L1Sel`, which aims to identify a near-optimal model from a set of candidate models $\{\phi_m(\cdot)^\top \mathbf{w}_m\}_{m \in [M]}$ based on $N$ *i.i.d.* samples $\{(\mathbf{x}_i, y_i)\}_{i=1}^N$. One intuitive approach is to estimate an appropriate estimator for each model using sampled data and then select the model whose estimated estimator exhibits the smallest approximate inter-model excess risk. However, this strategy can be computationally inefficient, especially when the candidate pool includes clearly sub-optimal models—i.e., models that can be ruled out as non-optimal with far fewer samples. To address this, we propose an *elimination-based selection algorithm* that first filters out sub-optimal models using a subset of the data, and then identifies a near-optimal model from the remaining candidates. Moreover, since our candidate models are pruned neural networks, we expect the selected model—and its associated estimator—to be at least as sparse as the original pruned network. To this end, we adopt Lasso (Tibshirani, 1996) estimation with $L_1$-regularization to estimate the last-layer weights of each candidate model.

As illustrated in Algorithm 1, `L1Sel` samples $N$ data points from the population. Then the algorithm sets $H$ epoch lengths $n_1 < \cdots < n_{H-1} < n_H = N$ of the sampled data. For each epoch $h$, the algorithm utilizes a subset of size $n_h$ of the sampled data. The elimination steps are executed for epochs $h \in [H-1]$, with the final epoch being reserved for the selection step.

**Elimination Step (line 4-9).** For each epoch $h \in [H-1]$, we maintain confidence bounds of each model's minimum risk using a subset of the sampled data of size $n_h$. Then, we obtain $\mathbf{w}_m^{n_h} = \arg\min_{\mathbf{w}} \hat{\mu}_{n_h}(m, \mathbf{w}) + \lambda_{n_h} L_1(\mathbf{w})$, the estimator of the $m$-th model, where $\hat{\mu}_{n_h}(m, \mathbf{w})$ is the empirical risk and $\lambda_{n_h} L_1(\mathbf{w})$ is the $L_1$-regularization term. Based on the estimator we set the upper and lower confidence bounds of each model's minimum risk as $\mathrm{ucb}(m, h) = \hat{\mu}_{n_h}(m, \mathbf{w}_m^{n_h}) + \alpha_{n_h}$ and $\mathrm{lcb}(m, h) = \hat{\mu}_{n_h}(m, \mathbf{w}_m^{n_h}) - \alpha_{n_h}$. Then we remove all of the models $m'$ from the pool that have a lower confidence bound $\mathrm{lcb}(m', h)$ which is larger than the minimum upper confidence bound $\mathrm{ucb}(\hat{m}_h, h)$, i.e., $\mathrm{ucb}(\hat{m}_h, h) < \mathrm{lcb}(m', h)$. We move to the next epoch $h+1$ and repeat the process.

**Selection Step (line 10-13).** At the final epoch $h = H$, we compute $\mathbf{w}_m^N$ for each $m \in \mathcal{M}_H$ using the entire sampled data of size $N$ and select the model with the smallest $L_1$-regularized empirical

risk. By employing $L_1$-regularization, the algorithm performs variable shrinkage, eliminating insignificant entries of each model's last layer parameters. Consequently, the selected pruned network achieves a sparser representation without compromising its performance.

We establish that under the survival of the best model, $m^* \in \mathcal{M}_H$, the model chosen by the selection criterion (line 13) exhibits an inter-model excess risk bounded by $\mathcal{O}(\frac{1}{\sqrt{N}})$ (Lemma 3 and Lemma 13), hence ensuring near-optimal selection if the sample size $N$ is large enough. This result motivates our focus on determining the necessary sample size for the algorithm to adhere to the $(\epsilon, \delta)$-PAC framework. In the following section, we verify the sample complexity required to guarantee: $\mathbb{P}\left(|\mu(m^N, \mathbf{w}_{m^N}^N) - \mu(m^*, \mathbf{w}_{m^*}^*)| \le \epsilon\right) \ge 1 - \delta$.

## 3.2 Sample complexity

In this section, we verify the sample complexity lower bounds for scenarios selecting a near-optimal model from a pool of regressors. Due to space limitations, the results for classifiers are deferred to the Appendix C. First, we introduce some notations to enhance clarity and analytical precision.

In the regression scenario, the algorithm selects from a pool of candidate models that predict a continuous scalar value $y \in \mathcal{Y}$ with the input $\mathbf{x} \in \mathcal{X}$. As $\mathcal{Y} \subset \mathbb{R}$, we notate the parameters $\mathbf{w}_m$ as a $d_m$-dimensional vector $\boldsymbol{\theta}_m \in \mathbb{R}^{d_m}$. Algorithm 1 necessitates the specification of the loss function and $L_1$-norm. We use the squared error as the loss function $\ell(\boldsymbol{\phi}_m(\mathbf{x})^\top \boldsymbol{\theta}, y) = (y - \boldsymbol{\phi}_m(\mathbf{x})^\top \boldsymbol{\theta})^2$, and the $L_1$-norm is $L_1(\boldsymbol{\theta}_m) = \|\boldsymbol{\theta}_m\|_1$. Now, we introduce some regularity assumptions used in the design and analysis of our algorithm.

**Assumption 1** (Boundedness). *There exist constants $\phi_{\max}, R, y_{\max} > 0$ such that for all $m \in [M]$ and $\mathbf{x} \in \mathcal{X}$, it holds that $\|\boldsymbol{\phi}_m(\mathbf{x})\|_\infty \le \phi_{\max}$, $\|\boldsymbol{\theta}_m^*\|_2 \le R$, and $|y| \le y_{\max}$ for all $y \in \mathcal{Y}$.*

**Assumption 2** (Sub-Gaussian). *For all $m \in [M]$, the true error defined by $\xi_{m,i} = y_i - \boldsymbol{\phi}_m(\mathbf{x}_i)^\top \boldsymbol{\theta}_m^*$ is $\sigma^2$-sub-Gaussian. That is, for any $t \in \mathbb{R}$, $\mathbb{E}\left[\exp(t\xi_{m,i})\right] \le \exp(t^2\sigma^2/2)$.*

**Assumption 3** (Compatibility). *For each $m \in [M]$, let the active set $S_m := \{j \in [d_m] : [\boldsymbol{\theta}_m^*]_j \ne 0\}$ denote the indices of non-zero entries in $\boldsymbol{\theta}_m^*$, and define the sparsity level $s_m := |S_m| \ll d_m$ as the cardinality of the active set. We assume that there exists a constant $\psi^2(S_m) > 0$ such that*

$$\psi^2(S_m) \le \frac{s_m \boldsymbol{\theta}^\top \mathbb{E}_{\mathbf{x} \sim \mathcal{X}}[\boldsymbol{\phi}_m(\mathbf{x})\boldsymbol{\phi}_m(\mathbf{x})^\top]\boldsymbol{\theta}}{\|\boldsymbol{\theta}_{S_m}\|_1^2}, \ \forall \boldsymbol{\theta} \in \mathbb{C}(S_m) := \{\boldsymbol{\theta} \in \mathbb{R}^{d_m} : \|\boldsymbol{\theta}_{S_m^c}\|_1 \le 3\|\boldsymbol{\theta}_{S_m}\|_1\}.$$

**Discussion of Assumptions.** Assumption 1 is standard in model selection literature (Foster et al., 2019; Papini et al., 2021; Tirinzoni et al., 2022; Kassraie et al., 2023) to establish results on sample complexity that are independent of the scaling of the representation feature map and parameter. When the labels are bounded, the true error $\xi_i$ is also constrained, and therefore, adheres to the sub-Gaussian assumption with a suitable value of $\sigma$. Assumption 3 guarantees the convergence property of sparse estimators in the high-dimensional statistic literature (Bühlmann & Van De Geer, 2011). Specifically, it ensures that the true set of active parameters are not overly correlated. Under this condition, the Lasso estimator accurately identifies relevant predictors while setting irrelevant coefficients to zero as the dataset grows larger.

Now, we are ready to present the sample complexity lower bound of Algorithm 1 that guarantees arbitrarily small inter-model excess risk with high probability.

**Theorem 1** (Sample Complexity of Algorithm 1: Regression). *Suppose Assumption 1, 2, and 3 hold. Let $\bar{m} := \arg\max_{m \in [M]} s_m/\psi^2(S_m)$ and $d := \max_{m \in [M]} d_m$. For any $\delta \in (0,1)$, $\epsilon > 0$, and $h \in [H]$, by setting the algorithmic parameters in Algorithm 1 as*

$$N \ge \frac{2}{\epsilon^2}\left(AC\epsilon + 2AD + 2B + 4\sqrt{ABD}\right), \lambda_{n_h} = \sqrt{A/n_h}, \alpha_{n_h} = \sqrt{B/n_h} + AC/5n_h,$$

*where*

$$A = 64\phi_{\max}^2\sigma^2\log(4dMH/\delta), \quad B = \frac{1}{2}(y_{\max} + \phi_{\max}R)^4\log(4MH/\delta),$$

$$C = 40s_{\bar{m}}/\psi^2(S_{\bar{m}}), \quad D = R^2,$$

*then we have $\mathbb{P}\left(|\mu(m^N, \boldsymbol{\theta}_{m^N}^N) - \mu(m^*, \boldsymbol{\theta}_{m^*}^*)| \le \epsilon\right) \ge 1 - \delta$.*

**Discussion of Theorem 1.** Theorem 1 establishes a sample complexity of $\Omega\left(\frac{1}{\epsilon^2}\log(dM)\right)$, ensuring that a model selected with this number of samples, with high probability, achieves inter-model excess risk below the target error $\epsilon$. The required sample size $N$ grows only *logarithmically* with the number of candidate models $M$, indicating that our algorithm remains sample-efficient even when $M$ is exponentially large. In particular, when $M = 1$ and $H = 1$, the sample complexity of our proposed algorithm matches the known sample complexity of the standard Lasso estimator (Bühlmann & Van De Geer, 2011).

The key challenge in deriving the sample complexity for model selection lies in analyzing the inter-model excess risk, which compares two inherently incomparable quantities: *the generalization error of the selected model* and *the risk of the best model*. We address this by decomposing the inter-model excess risk into model-specific components that are individually analyzable, leveraging structural properties satisfied by both the optimal model and the empirically selected model. This analytical approach is novel, as it is specifically designed to handle the inter-model excess risk, a concept newly introduced in this work (Lemma 3 and Lemma 13).

## 4 EXPERIMENTS

In this section, we present numerical experiments to evaluate the performance of the proposed algorithm `L1Sel`.

### 4.1 EXPERIMENT 1: GENERALIZATION ERROR

We demonstrate that `L1Sel` effectively selects a model with low generalization error. We report results from two classification and two regression model selection scenarios, each using a pool of 100 pruned neural networks derived from an original network. In this experiment, we note that accuracy on the test set is not the primary objective. Rather, our goal is to evaluate the effectiveness of the proposed method in selecting the relatively better-performing model from a given set of candidates. To this end, we employ random pruning to efficiently generate the candidate models. We generate 20 randomly pruned classification models at sparsity rates of 50%, 60%, 70%, 80%, and 90% and 20 pruned regression models at sparsity rate of 30%, 40%, 50%, 60%, and 70%.

In the classification model selection scenarios, the original network for each is an MLPNet trained on MNIST (LeCun et al., 1998) and a Resnet20 (He et al., 2016) trained on CIFAR10 (Krizhevsky et al., 2009). For the regression model selection scenarios, the original architecture is an MLPNet with a 1-dimensional output, referred to as MLPNetR. The datasets for regression model selection are the *Superconductivity Data* (Super) (Hamidieh, 2018) from the UC Irvine Machine Learning Repository and the *California Housing Dataset* (Cal) (Pace & Barry, 1997).

For each scenario, we assume the entire training dataset as the population and that only a maximum of 50% of the population can be sampled. This configuration is implemented to illustrate the generalizability of the selected model. To verify that the selected model is near-optimal, we must compute the difference between its generalization error and the minimum risk of the best model. However, accessing the true population of any dataset is impractical (e.g., for MNIST, the population data would encompass all handwritten digits worldwide). Hence, employing a bootstrap approach, we resample from the entire training dataset, which is essentially a sample itself, to simulate the process of sampling from a population.

Figure 1 gives the results for the first random seed, with each scenario running for $H = 10$ epochs. Results for the remaining trials are provided in Section D.3. Note that with high probability, $\text{lcb}(m^*, h) < \text{ucb}(m, h)$ for all $m \in [M]$, as else, it would lead to a contradiction, implying $\mu(m', \mathbf{w}^*_{m'}) < \mu(m^*, \mathbf{w}^*_{m^*})$ for some $m' \in [M]$. Thus, the best model $m^*$ survives during the elimination step with high probability. At the selection step, our algorithm ensures that the selected model's generalization error $\mu(m^N, \mathbf{w}^N_{m^N})$ is sufficiently low to keep the inter-model excess risk below an error criterion $\epsilon$. Given that the experiments were conducted with a sample complexity amount of data points, the confidence interval length $2\alpha_N$ is smaller than $\epsilon$ with high probability (Eq. (14) for regression and Eq. (43) for classification). As both $\mu(m^N, \mathbf{w}^N_{m^N})$ and $\mu(m^*, \mathbf{w}^*_{m^*})$ fall within the confidence interval in the figures, it follows that $\mu(m^N, \mathbf{w}^N_{m^N}) - \mu(m^*, \mathbf{w}^*_{m^*}) \leq 2\alpha_N \leq \epsilon$, guaranteeing the selection of a near-optimal model.

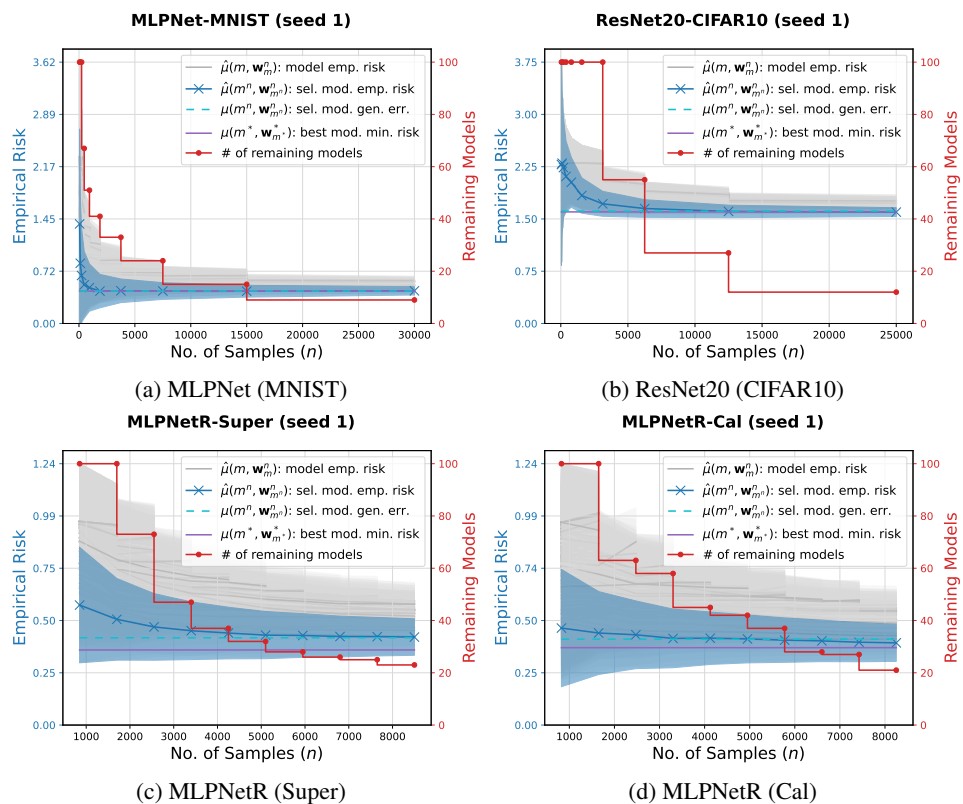

(a) MLPNet (MNIST)

(b) ResNet20 (CIFAR10)

(c) MLPNetR (Super)

(d) MLPNetR (Cal)

Figure 1: Results for (a) MLPNet (MNIST), (b) ResNet20 (CIFAR10) (c) MLPNetR (Super), and (d) MLP-NetR (Cal). The gray lines represent the empirical risk of the surviving models. The gray shades depict their confidence intervals for their minimum risks. The blue line and shades describe the selected model's minimum empirical risk and confidence interval of its minimum risk. The cyan dashed line and purple solid line are the selected model's generalization error and the best model's minimum risk respectively. The red line depicts the remaining models for each epoch. During the elimination step, the algorithm removes the models whose lower confidence bound exceeds the minimum upper confidence bound for each epoch. This results in the steep declines observed in the gray lines and shaded areas at each epoch.

## 4.2 EXPERIMENT 2: COMPARISON WITH EXISTING MODEL SELECTION METHODS

In this experiment, we evaluate our algorithm in a classification model selection setting by comparing the generalization error of the model it selects against those selected by other model selection methods. We compare our method with 5-fold and 10-fold cross-validation, which are commonly used in practice (Ding et al., 2018), denoted as `CV(5)` and `CV(10)`, respectively. Additionally, we evaluate our algorithm against several deep model selection methods, **TS** (Yang et al., 2024), **PARC** (Bolya et al., 2021), and **SFDA** (Shao et al., 2022).

To give a harsher model selection task, we generate a candidate pool having 50 randomly pruned models at sparsity rates of each 50% and 60%, and use a smaller sample size of $N = 10,000$. To make the model selection task more challenging, we construct a candidate pool consisting of 50 randomly pruned models at sparsity levels of 50% and 60%, and we use a smaller sample size of $N = 10,000$. To correspond with the reduced sample size, we also lower the number of training epochs to $H = 5$, while keeping all other hyperparameters unchanged. Since not all methods include a training procedure (Yang et al., 2024; Bolya et al., 2021; Shao et al., 2022), for fair comparison, we train the model selected by each method before evaluating its generalization error.

Table 1 compares our algorithm and other methods in terms of generalization error and runtime. **L1Sel**, **CV(5)**, and **CV(10)** outperform in identifying a model with low generalization error compared to the other three methods. Though the other methods may have advantage in runtime, as they are primarily designed for choosing a representation model for transfer learning (**PARC**

| Selection Method | MLPNet (MNIST) | | ResNet20 (CIFAR10) | |
|---|---|---|---|---|
| | Generalization Error | Runtime (sec) | Generalization Error | Runtime (sec) |
| **L1Sel** | **0.3957 ± 0.0222** | **441.33 ± 35.05** | **1.6762 ± 0.0254** | **2,624.00 ± 186.44** |
| **CV(5)** | 0.3961 ± 0.0222 | 964.13 ± 95.46 | 1.6841 ± 0.0342 | 3,254.94 ± 23.45 |
| **CV(10)** | 0.3961 ± 0.0235 | 2,203.26 ± 151.60 | 1.6905 ± 0.0366 | 6,654.92 ± 386.59 |
| **TS** | 0.4543 ± 0.0569 | 291.76 ± 20.43 | 1.7594 ± 0.0658 | 1,046.94 ± 89.82 |
| **PARC** | 0.4923 ± 0.0829 | **21.17 ± 2.69** | 1.7026 ± 0.0381 | **60.70 ± 4.42** |
| **SFDA** | 0.5518 ± 0.0668 | 213.80 ± 20.67 | 1.8460 ± 0.0633 | 918.68 ± 30.16 |

Table 1: Comparison of the generalization error of the selected model and the runtime of proposed algorithm and other methods. The top three methods involve a learning process, leading to a long runtime, while the bottom three do not.

| Network | | Accuracy(%) | | | | Last Layer Sparsity Rate(%) | | | |
|---|---|---|---|---|---|---|---|---|---|
| Architecture (Dataset) | Pruning Rate | GM | M-FAC | CHITA | L1Sel | GM | M-FAC | CHITA | L1Sel |
| MLPNet (MNIST) | 0.7 | 93.55 | 93.95 | 93.89 | **94.24 ± 0.03** | 7.95 | 6.65 | 7.50 | **18.40 ± 0.77** |
| | 0.8 | 92.88 | 93.64 | 93.69 | **94.01 ± 0.12** | 10.20 | 11.10 | 9.50 | **20.75 ± 0.42** |
| | 0.9 | 90.03 | 92.73 | 92.41 | **93.17 ± 0.16** | 14.85 | 17.55 | 15.00 | **26.15 ± 0.75** |
| | 0.95 | 79.37 | 90.54 | 86.60 | **91.49 ± 0.25** | 19.20 | 27.75 | 19.00 | **32.45 ± 1.67** |
| | 0.98 | 23.00 | 80.57 | 31.13 | **85.18 ± 0.59** | 41.80 | 48.05 | 24.50 | **48.05 ± 1.77** |
| ResNet20 (CIFAR10) | 0.5 | 87.31 | 89.79 | 90.56 | **90.99 ± 0.07** | 7.91 | 7.70 | 7.81 | **19.83 ± 2.32** |
| | 0.6 | 85.89 | 88.88 | 88.71 | **90.36 ± 0.11** | 8.80 | 9.14 | 8.75 | **17.78 ± 1.86** |
| | 0.7 | 82.00 | 86.94 | 82.42 | **89.08 ± 0.12** | 10.59 | 11.09 | 10.47 | **15.95 ± 1.60** |
| | 0.8 | 71.30 | 81.82 | 66.81 | **85.49 ± 0.17** | 14.31 | 15.52 | 13.75 | **18.07 ± 0.97** |
| | 0.9 | 41.96 | 66.20 | 32.97 | **75.81 ± 0.62** | 20.86 | 24.19 | 20.31 | **25.25 ± 0.90** |

Table 2: Test set accuracy and last layer sparsity rate of the pruned networks in each pool and the model selected by the proposed algorithm.

and **SFDA**) or unsupervised domain adaptation (**TS**), they show inferior performance in selecting a model with low generalization error. Compared to CV, our algorithm offers distinct advantages. Since CV selects the model with the highest average accuracy across folds, it can favor models that overfit to specific folds, resulting in inflated average accuracy. Consequently, models selected by CV often exhibit weaker generalizability compared to those chosen by our algorithm. Additionally, our algorithm achieves greater runtime efficiency through its elimination step, which reduces computational overhead. Furthermore, it is the only algorithm with theoretical guarantees on finite sample complexity for selecting a model with low generalization error. This property is particularly crucial in scenarios where data acquisition is costly, ensuring both efficient and reliable model selection.

### 4.3 EXPERIMENT 3: ACCURACY AND SPARSITY RATE OF THE IDENTIFIED MODEL

In this experiment, we show that our algorithm selects *a model with an improved or at least preserved sparsity rate and accuracy* compared to the top-performing model in the candidate pool. For this experiment, we prune MLPNet and ResNet20 with several benchmark pruning algorithms: Global $L_1$-Magnitude Pruning (**GM**) provided in the PyTorch library, **M-FAC** (Frantar et al., 2021), and **CHITA** (Benbaki et al., 2023). For each original network, we apply 5 different pruning rates, generating 3 models per rate. This creates a candidate pool of 3 *state-of-the-art* (SOTA) pruned neural networks for each original network and pruning rate. We run our algorithm for each pool, and measure the selected model's accuracy on the test dataset and its last layer sparsity rate.

The results are aggregated in Table 2. For both MLPNet and ResNet20, the model selected by our algorithm shows higher accuracy in all of the pruning rates. Moreover, we examine the last layer sparsity rate significantly increases for low pruning rates, and even increases when the pruning rates are high. *Experiment 3 shows that our method consistently selects pruned networks with at least improved sparsity and accuracy to the top-performing model in the candidate pool.* Notably, the algorithm enhances these key performance measures even for already highly sparse networks.

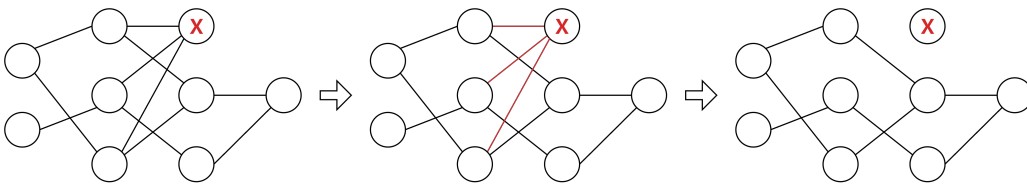

Figure 2: Illustration of `L1Sel+` (Algorithm 2). The network in the first figure illustrates the neural network selected by `L1Sel`. As `L1Sel` promotes sparsity in the last-layer weights, the weights of the penultimate layer connected to these now deactivated last-layer weights (indicated by red lines in the second figure) no longer influence the network output. By further pruning these weights, the network's sparsity can be increased without incurring a loss in performance (third figure).

### 4.4 EXPERIMENT 4: FURTHERING GLOBAL SPARSITY RATE WITH `L1SEL+`

Additionally, we propose `L1Sel+` (Algorithm 2), an algorithm that further improves the global sparsity rate when the candidates are neural network regression models. As regression models have an output dimension of 1, the components of the feature extractor $\phi_m(\mathbf{x})$ connected to the entries of the last layer weights $\boldsymbol{\theta}_m$ that are 0 do not impact the loss. Leveraging this concept, `L1Sel+` prunes the columns of the penultimate layer weights corresponding to the feature map components linked to the 0 value entries of $\boldsymbol{\theta}_m$ (Figure 2). The algorithm ensures at least increasing the global sparsity rate without altering the loss.

---

**Algorithm 2 `L1Sel+`**

1: **Input:** Output of Algorithm 1: $\phi_{m^N}(\cdot)^\top \mathbf{w}_{m^N}^N$
2: Extract the penultimate layer weights $\mathbf{w}_{m^N,\text{penult}}$ from the feature extractor $\phi_{m^N}(\cdot\,;\mathbf{w}_{m^N,\text{feat}})$ where $\mathbf{w}_{m^N,\text{feat}} = (\mathbf{w}_{m^N,\text{feat}-\text{penult}},\ \mathbf{w}_{m^N,\text{penult}})$
3: For every $j$ such that $[\mathbf{w}_{m^N}^N]_j = 0$, prune all of the weights of the $j$-th column of $\mathbf{w}_{m^N,\text{penult}}$
4: **Return:** $\phi_{m^N}(\cdot\,;\mathbf{w}_{m^N,\text{feat}-\text{penult}},\ \mathbf{w}_{m^N,\text{penult}})$

---

Table 3 shows the results applying `CHITA` → `L1Sel` → `L1Sel+` to an MLPNetR (Super) in a sequence. Comparing `CHITA` and `L1Sel`, the last layer sparsity rate shows substantial improvement, and the loss gets lower particularly in high pruning rates. Indeed, the loss after applying `L1Sel+` does not change at all. Notably, `L1Sel+` even stretches the global sparsity rate of already highly pruned networks, enhancing the global sparsity rate from 98% up to 99%.

| Pruning Rate | Loss | | Last Layer Sparsity Rate(%) | | Global Sparsity Rate(%) | | |
|---|---|---|---|---|---|---|---|
| | **CHITA** | **L1Sel(= L1Sel+)** | **CHITA** | **L1Sel(= L1Sel+)** | **CHITA** | **L1Sel** | **L1Sel+** |
| 0.80 | 0.4915 | **0.4185 ± 0.0097** | 0.00 | **66.00 ± 5.68** | 80.02 | 80.35 | **84.37 ± 0.49** |
| 0.85 | 0.7550 | **0.4702 ± 0.0247** | 0.00 | **71.00 ± 3.16** | 85.00 | 85.35 | **89.09 ± 0.32** |
| 0.90 | 0.9736 | **0.5561 ± 0.0278** | 5.00 | **73.50 ± 4.74** | 90.02 | 90.36 | **93.02 ± 0.28** |
| 0.95 | 3.3042 | **0.7191 ± 0.0376** | 5.00 | **72.50 ± 5.40** | 95.00 | 95.33 | **96.38 ± 0.45** |
| 0.98 | 3.9384 | **0.9280 ± 0.0077** | 15.00 | **92.00 ± 2.58** | 98.00 | 98.38 | **99.07 ± 0.20** |

Table 3: Test set loss, last layer sparsity rate, and global sparsity rate of MLPNetR (Super) after applying each method.

## 5 CONCLUSION

In this paper, we study the problem of identifying a near-optimal pruned neural network from a set of candidates. By conceptualizing each neural network as a combination of a feature representation and last-layer weights, we propose a model selection algorithm that integrates representation selection with sparse estimator learning, ensuring low generalization error. We establish that the sample complexity lower bound scales logarithmically with the number of candidates, demonstrating that the proposed algorithm is sample-efficient even for a large number of models. Through numerical experiments, we demonstrate that our algorithm efficiently identifies pruned neural networks with minimal generalization error while improving the sparsity rate without compromising performance.

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

CONTENTS OF APPENDIX

## A  RELATED WORK

**Neural network pruning.**  Neural network pruning methods can be classified by their pruning criterion and its overall pruning strategy. The straightforward criterion is magnitude-based pruning,

which decides whether to prune a parameter or not based on the absolute value of its weight (Hanson & Pratt, 1988; Mozer & Smolensky, 1989; Gale et al., 2019). On the other hand, impact-based methods raise concerns that small weight parameters may play a role in the network structure. Originating from the well-known Optimal Brain Damage/Surgeon framework (LeCun et al., 1989; Hassibi & Stork, 1992), these methods approximate the loss function as a local quadratic model, and seek a parameter or set of parameters to remove subject to minimizing the increase in the loss. Recent studies show improvement in estimating the Hessian matrix (Singh & Alistarh, 2020; Yu et al., 2022) or even furthering to a matrix-free framework (Frantar et al., 2021; Benbaki et al., 2023).

A common practice in pruning is utilizing an update strategy to improve the performance of a pruned neural network compared to its *one-shot*–a network pruned only by applying a pruning criterion. The gradual pruning strategy (Han et al., 2015; Zhu & Gupta, 2017) has gained widespread acceptance within the field. The strategy starts from a pre-trained dense neural network and runs through an iterative process scheduled to gradually reduce the network until it reaches a target sparsity, showing significant improvement for both magnitude-based pruning (Gale et al., 2019) and impact-based ones (Singh & Alistarh, 2020). The dynamic pruning strategy trains a dense neural network by updating its parameters using the gradients of its period-wise optimal pruned variation (Lin et al., 2020). Multi-stage pruning employs a scheduled, iterative process that progressively tightens the sparsity constraint by taking incremental steps towards a greater degree of sparsity (Benbaki et al., 2023). Unlike gradual pruning, this method avoids the overhead of retraining, and hence significantly boosting speed.

While existing frameworks offer valuable pruning algorithms, a comprehensive framework and theoretical guarantee for selecting an optimal pruned neural network remain largely absent in the literature. Developing effective methods that provide clear guidance in this selection process would constitute a significant contribution to the field of lightweight neural network deployment.

**Model Selection.** Model selection is the task of choosing a model from a set of candidates in accordance with a predefined objective. It is employed across a wide range of fields, including variable selection (Marill & Green, 1963; Chen & Chen, 2008; Chandrashekar & Sahin, 2014), selecting a pre-trained deep model for transfer learning (Bolya et al., 2021; Shao et al., 2022) or unsupervised domain adaptation (You et al., 2019; Yang et al., 2024), determining the number of components in mixture models (McLachlan & Rathnayake, 2014; Celeux et al., 2019), and policy selection in interactive learning (Foster et al., 2019; Zitovsky et al., 2023).

Towards seeking a generalizable model, *Akaike Information Criterion* (AIC) (Akaike, 1974) and BIC (Schwarz, 1978) have become the cornerstones for parametric model-based model selection methods. AIC is more focused on prediction while BIC is more focused on finding the true model. Various improvements or balanced approaches have since been proposed, including the *Hannan-Quinn Criterion* (HQC) (Hannan & Quinn, 1979), Extended BIC (Chen & Chen, 2008), and the Bridge Criterion (Ding et al., 2017). Beyond information criteria, other methods have been inspired by Bayesian (Djuric, 1998; Andrieu et al., 2001; Shao et al., 2019), information-theoretic (Wallace & Boulton, 1968; Rissanen, 1978), or decision-theoretic (Dawid, 1984) perspectives.

For model selection frameworks without the requirement for candidates to be parametric, CV (Allen, 1974; Geisser, 1975; Stone, 1977) is widely employed in almost every machine learning pipeline, emphasizing the importance of model selection for successful machine learning deployment. It is important to note that the performance guarantees of these methods are asymptotic with respect to sample size and do not provide explicit quantification of model selection performance with finite samples. For a comprehensive review of various model selection methods, please refer to (Ding et al., 2018) and the references therein.

# B SAMPLE COMPLEXITY LOWER BOUND OF ALGORITHM 1 FOR REGRESSION TASK

In this section, we present the rigorous proof of Theorem 1.

## B.1 PRELIMINARY

Let $\epsilon > 0$ be the error criterion, $n_1 < n_2 < \cdots < n_H = N$ be the epoch lengths, and $\mathcal{M}_h$ be the index set of models remaining in the candidate pool at epoch $h \in [H]$. For notational simplicity, we denote $n := n_h$ for a fixed epoch $h \in [H]$ when it is clear from the context. We denote the empirical process as $\nu_n(m, \boldsymbol{\theta}_m) := \hat{\mu}_n(m, \boldsymbol{\theta}_m) - \mu(m, \boldsymbol{\theta}_m)$ and define model $\bar{m}$ as $\bar{m} := \operatorname{argmax}_{m \in [M]} s_m / \psi^2(S_m)$. We also introduce some events utilized in the subsequent proofs.

- $\mathcal{E} := \left\{ \mu(m^N, \boldsymbol{\theta}_{m^N}^N) - \mu(m^*, \boldsymbol{\theta}_{m^*}^*) > \epsilon \right\}$ : the event that inter-model excess risk of the model selected by Algorithm 1 exceeds $\epsilon$.

- $\mathcal{S}_{n,m} := \{|\nu_n(m, \boldsymbol{\theta}_m^*)| \leq \eta_n\}$: the event that empirical process of the $m$-th model having its optimal parameter is bounded.

- $\mathcal{S}_n := \bigcap_{m \in \mathcal{M}_h} \mathcal{S}_{n,m}$

- $\mathcal{T}_{n,m} := \left\{ \|\frac{1}{n} \sum_{i=1}^n \xi_{m,i} \boldsymbol{\phi}_m(\mathbf{x}_i)\|_\infty \leq \tilde{\lambda}_n \right\}$ : the event that model $m$'s maximum absolute deviation across all feature dimensions is bounded, where $\xi_{m,i} = y_i - \boldsymbol{\phi}_m(\mathbf{x}_i)^\top \boldsymbol{\theta}_m^*$.

- $\mathcal{T}_n := \bigcap_{m \in \mathcal{M}_h} \mathcal{T}_{n,m}$

- $\mathcal{R}_h := \left\{ \bigcap_{h'=1}^h (\mathcal{S}_{n_{h'}} \cap \mathcal{T}_{n_{h'}}) \right\}$: the event that $\mathcal{S}_{n_{h'}}$ and $\mathcal{T}_{n_{h'}}$ both hold for all of the epochs $h' \in [h]$.

Before we present the main proof of Theorem 1, we first introduce the following key lemmas.

**Lemma 1.** *For all $h \in [H]$, if the regularization parameters $\{\lambda_{n_h}\}_{h=1}^H$ and confidence radii $\{\alpha_{n_h}\}_{h=1}^H$ of Algorithm 1 are set as $\lambda_{n_h} = \sqrt{A/n_h}$ and $\alpha_{n_h} = \sqrt{B/n_h} + AC/5n_h$ , where $A$, $B$ and $C$ are the values specified in Theorem 1, then event $\mathcal{R}_H$ occurs with probability at least $1 - \delta$.*

**Lemma 2.** *Under $\mathcal{R}_{H-1}$ and by setting $\alpha_{n_h} = \eta_{n_h} + 8\lambda_{n_h}^2 s_{\bar{m}}/\psi^2(S_{\bar{m}})$ for epochs $h \in [H-1]$, the best model $m^*$ survives for the selection step, i.e., $m^* \in \mathcal{M}_H$.*

**Lemma 3.** *If $m^* \in \mathcal{M}_H$, under $\mathcal{S}_N \cap \mathcal{T}_N$, the inter-model excess risk of the selected model $m^N$ is upper bounded as follows:*

$$\underbrace{\mu(m^N, \boldsymbol{\theta}_{m^N}^N) - \mu(m^*, \boldsymbol{\theta}_{m^*}^*)}_{\text{inter-model excess risk}} \leq 2\eta_N + 2\lambda_N R + 40 \frac{\lambda_N^2 s_{\bar{m}}}{\psi^2(S_{\bar{m}})} .$$

Lemma 1 establishes that by setting $\{\lambda_{n_h}\}_{h=1}^H$ and $\{\alpha_{n_h}\}_{h=1}^H$ in Algorithm 1 according to Theorem 1, a desired event $\mathcal{R}_H = \mathcal{R}_{H-1} \cap \mathcal{S}_N \cap \mathcal{T}_N$ occurs with high probability. Lemma 2 indicates that under $\mathcal{R}_{H-1}$ and by configuring the confidence radii as specified in Theorem 1, the candidate pool retains the best model $m^*$ for the selection step, $m^* \in \mathcal{M}_H$. Lemma 3 demonstrates that under the survival of the best model for the selection step along with the occurrence of event $\mathcal{S}_N \cap \mathcal{T}_N$, the inter-model excess risk of the selected model $m^N$ is bounded by $\mathcal{O}(\frac{1}{\sqrt{N}})$. The proof of Theorem 1 leverages Lemmas 1 and 3 to derive a sample complexity lower bound required for our algorithm to meet the $(\epsilon, \delta)$-PAC criteria.

A brief overview of the following sections is as follows. In Section B.2, we establish Lemma 1. In Section B.3, we verify Lemma 2. In Section B.4, we demonstrate Lemma 3. In Section B.5, we prove Theorem 1. In Section B.6, we provide the proofs for the technical lemmas used in this section.

## B.2 PROOF OF LEMMA 1

*Proof of Lemma 1.* The proof of Lemma 1 utilizes the following lemmas.

**Lemma 4.** *Let $M_h = |\mathcal{M}_h|$ be the number of remaining models for epoch $h$. For a given $\delta_1 \in (0, 1)$, if*

$$\eta_n = \sqrt{\frac{(y_{\max} + \phi_{\max} R)^4 \log(2M_h/\delta_1)}{2n}} ,$$

*event $\mathcal{S}_n$ occurs with probability at least $1 - \delta_1$.*

**Lemma 5.** *Let $M_h = |\mathcal{M}_h|$ be the number of remaining models for epoch $h$. For a given $\delta_2 \in (0, 1)$, if*

$$\tilde{\lambda}_n = \sqrt{\frac{\phi_{\max}^2 \sigma^2 \log(2dM_h/\delta_2)}{n}},$$

*where $d = \max_{m \in [M]} d_m$, event $\mathcal{T}_n$ occurs with probability at least $1 - \delta_2$.*

Note that setting the regularization parameters $\{\lambda_{n_h}\}_{h=1}^H$ and confidence radii $\{\alpha_{n_h}\}_{h=1}^H$ of Algorithm 1 as $\lambda_{n_h} = \sqrt{A/n_h}$ and $\alpha_{n_h} = \sqrt{B/n_h} + AC/5n_h$ implies setting $\{\eta_{n_h}\}_{h=1}^H$ as $\eta_{n_h} = \sqrt{B/n_h}$.

For all $h \in [H]$, by setting $\{\lambda_{n_h}\}_{h=1}^H$ and $\{\alpha_{n_h}\}_{h=1}^H$ as specified in Lemma 1, and by choosing $\delta_1$, $\delta_2$, and $\tilde{\lambda}_{n_h}$ as $\delta_1 = \delta_2 = \frac{\delta}{2H}$ and $\lambda_{n_h} = 8\tilde{\lambda}_{n_h} = \sqrt{A/n_h}$, we have,

$$\eta_{n_h} = \sqrt{B/n_h} = \sqrt{\frac{(y_{\max} + \phi_{\max}R)^4 \log(4MH/\delta)}{2n_h}} \geq \sqrt{\frac{(y_{\max} + \phi_{\max}R)^4 \log(2M_h/\delta_1)}{2n_h}}$$

and

$$\tilde{\lambda}_{n_h} = \frac{1}{8}\sqrt{A/n_h} = \sqrt{\frac{\phi_{\max}^2 \sigma^2 \log(4dMH/\delta)}{n_h}} \geq \sqrt{\frac{\phi_{\max}^2 \sigma^2 \log(2dM_h/\delta_2)}{n_h}},$$

where for the last inequality of each, we use $M \geq M_h$ for all $h \in [H]$. Thus, under the conditions established in Lemma 1, Lemmas 4 and 5 hold.

Using the results of Lemmas 4 and 5, we get

$$\mathbb{P}(\mathcal{R}_H) = 1 - \mathbb{P}(\mathcal{R}_H^{\mathsf{c}})$$

$$= 1 - \left[\mathbb{P}\left(\bigcup_{h=1}^H \mathcal{S}_{n_h}^{\mathsf{c}}\right) + \mathbb{P}\left(\bigcup_{h=1}^H \mathcal{T}_{n_h}^{\mathsf{c}}\right)\right]$$

$$\geq 1 - \left[\sum_{h=1}^H \mathbb{P}\left(\mathcal{S}_{n_h}^{\mathsf{c}}\right) + \sum_{h=1}^H \mathbb{P}\left(\mathcal{T}_{n_h}^{\mathsf{c}}\right)\right]$$

$$\geq 1 - H\delta_1 - H\delta_2$$

$$= 1 - \delta,$$

where the first inequality uses union bound, the second inequality follows by the results of Lemmas 4 and 5, and the last equality comes from the choice of $\delta_1 = \delta_2 = \frac{\delta}{2H}$. $\qquad\square$

### B.3 Proof of Lemma 2

*Proof of Lemma 2.* We introduce the following lemma which provides the validity of the confidence interval.

**Lemma 6.** *For any $h \in [H-1]$, let $n := n_h$. Suppose events $\mathcal{S}_n$ and $\mathcal{T}_n$ both occur. Then, for all $m \in \mathcal{M}_h$, by setting $\lambda_n = 8\tilde{\lambda}_n$ and $\alpha_n = \eta_n + 8\lambda_n^2 s_{\bar{m}}/\psi^2(S_{\bar{m}})$, the following holds:*

$$\mu(m, \boldsymbol{\theta}_m^*) \in [\hat{\mu}_n(m, \boldsymbol{\theta}_m^n) - \alpha_n, \ \hat{\mu}_n(m, \boldsymbol{\theta}_m^n) + \alpha_n].$$

For the remaining, we use proof by contradiction. Assume that $m^*$ gets eliminated at a previous epoch $h' \in [H-1]$. This means there exists a model $m'$ such that $\text{ucb}(m', h') < \text{lcb}(m^*, h')$. On the other hand, the occurrence of $\mathcal{R}_{H-1}$ implies that event $\mathcal{S}_{n_{h'}} \cap \mathcal{T}_{n_{h'}}$ takes place. Thus, by Lemma 6, for each model $m \in \mathcal{M}_{h'}$, we have

$$\mu(m, \boldsymbol{\theta}_m^*) \in [\text{lcb}(m, h'), \ \text{ucb}(m, h')],$$

where $\text{ucb}(m, h') = \hat{\mu}_{n_{h'}}(m, \boldsymbol{\theta}_m^{n_{h'}}) + \alpha_{n_{h'}}$ and $\text{lcb}(m, h') = \hat{\mu}_{n_{h'}}(m, \boldsymbol{\theta}_m^{n_{h'}}) - \alpha_{n_{h'}}$.

However, this represents $\mu(m', \boldsymbol{\theta}_{m'}^*) \leq \text{ucb}(m', h') < \text{lcb}(m^*, h') \leq \mu(m^*, \boldsymbol{\theta}_{m^*}^*)$, which is a contradiction since $\mu(m^*, \boldsymbol{\theta}_{m^*}^*) \leq \mu(m, \boldsymbol{\theta}_m)$ for any $m \in [M]$. Thus, under $\mathcal{R}_{H-1}$ and by setting $\alpha_{n_h} = \eta_{n_h} + 8\lambda_{n_h}^2 s_{\bar{m}}/\psi^2(S_{\bar{m}})$ for epochs $h \in [H-1]$, we conclude that the best model $m^*$ does not get eliminated at any previous epoch $h' \in [H-1]$. $\qquad\square$

### B.4 PROOF OF LEMMA 3

*Proof of Lemma 3.* Establishing Lemma 3 consists of two steps. Step 1 bounds the inter-model excess risk with the sum of the model-wise excess risk of the selected model and an additional $\tilde{\Delta}$ term. Step 2 bounds the sum of these terms and completes the proof.

STEP 1: BOUNDING THE INTER-MODEL EXCESS RISK

As $m^* \in \mathcal{M}_H$, we know that the best model is in the candidate pool for the selection step. We begin with decomposing the inter-model excess risk:

$$\mu(m^N, \boldsymbol{\theta}_{m^N}^N) - \mu(m^*, \boldsymbol{\theta}_{m^*}^*) = [\mu(m^N, \boldsymbol{\theta}_{m^N}^N) - \mu(m^N, \boldsymbol{\theta}_{m^N}^*)] + [\mu(m^N, \boldsymbol{\theta}_{m^N}^*) - \mu(m^*, \boldsymbol{\theta}_{m^*}^*)] \tag{1}$$

Let us denote $\tilde{\mu}_N(m, \boldsymbol{\theta}) := \hat{\mu}_N(m, \boldsymbol{\theta}) + \lambda \|\boldsymbol{\theta}\|_1$ the empirical risk with an $L_1$-regularization term and $\tilde{\Delta} := \max_{m \in \mathcal{M}_H} |\tilde{\mu}_N(m, \boldsymbol{\theta}_m^N) - \mu(m, \boldsymbol{\theta}_m^*)|$ the maximum of the model-wise difference between the minimum empirical risk with $L_1$-regularization and the minimum risk. For each $m \in \mathcal{M}_H$, since $\mu(m, \boldsymbol{\theta}_m^*)$ belongs to the following interval,

$$\mu(m, \boldsymbol{\theta}_m^*) \in \left[ \tilde{\mu}_N(m, \boldsymbol{\theta}_m^N) - \left| \tilde{\mu}_N(m, \boldsymbol{\theta}_m^N) - \mu(m, \boldsymbol{\theta}_m^*) \right|, \tilde{\mu}_N(m, \boldsymbol{\theta}_m^N) + \left| \tilde{\mu}_N(m, \boldsymbol{\theta}_m^N) - \mu(m, \boldsymbol{\theta}_m^*) \right| \right],$$

we have,

$$\tilde{\mu}_N(m, \boldsymbol{\theta}_m^N) - \tilde{\Delta} \le \mu(m, \boldsymbol{\theta}_m^*) \le \tilde{\mu}_N(m, \boldsymbol{\theta}_m^N) + \tilde{\Delta}. \tag{2}$$

Note that since (2) holds for all $m \in \mathcal{M}_H$, for the best model $m^*$ and model selected by Algorithm 1 $m^N$, the followings hold:

$$\tilde{\mu}_N(m^*, \boldsymbol{\theta}_{m^*}^N) - \tilde{\Delta} \le \mu(m^*, \boldsymbol{\theta}_{m^*}^*) \le \tilde{\mu}_N(m^*, \boldsymbol{\theta}_{m^*}^N) + \tilde{\Delta}, \tag{3}$$

$$\tilde{\mu}_N(m^N, \boldsymbol{\theta}_{m^N}^N) - \tilde{\Delta} \le \mu(m^N, \boldsymbol{\theta}_{m^N}^*) \le \tilde{\mu}_N(m^N, \boldsymbol{\theta}_{m^N}^N) + \tilde{\Delta}. \tag{4}$$

By combining (3) and (4), we have

$$\tilde{\mu}_N(m^*, \boldsymbol{\theta}_{m^*}^N) - \tilde{\Delta} \le \mu(m^*, \boldsymbol{\theta}_{m^*}^*) \le \mu(m^N, \boldsymbol{\theta}_{m^N}^*) \le \tilde{\mu}_N(m^N, \boldsymbol{\theta}_{m^N}^N) + \tilde{\Delta}. \tag{5}$$

Then, by plugging (5) into (1), we have

$$\begin{aligned}
(1) &\le \left[ \mu(m^N, \boldsymbol{\theta}_{m^N}^N) - \mu(m^N, \boldsymbol{\theta}_{m^N}^*) \right] + [\tilde{\mu}_N(m^N, \boldsymbol{\theta}_{m^N}^N) + \tilde{\Delta}] - [\tilde{\mu}_N(m^*, \boldsymbol{\theta}_{m^*}^N) - \tilde{\Delta}] \\
&= \left[ \mu(m^N, \boldsymbol{\theta}_{m^N}^N) - \mu(m^N, \boldsymbol{\theta}_{m^N}^*) \right] + [\tilde{\mu}_N(m^N, \boldsymbol{\theta}_{m^N}^N) - \tilde{\mu}_N(m^*, \boldsymbol{\theta}_{m^*}^N)] + 2\tilde{\Delta} \\
&\le \left[ \mu(m^N, \boldsymbol{\theta}_{m^N}^N) - \mu(m^N, \boldsymbol{\theta}_{m^N}^*) \right] + 2\tilde{\Delta},
\end{aligned}$$

where the last inequality holds since $\tilde{\mu}_N(m^N, \boldsymbol{\theta}_{m^N}^N) - \tilde{\mu}_N(m^*, \boldsymbol{\theta}_{m^*}^N) \le 0$ due to the selection of Algorithm 1, i.e., $m^N = \operatorname{argmin}_{m \in \mathcal{M}_H} \tilde{\mu}_N(m^N, \boldsymbol{\theta}_{m^N}^N)$. Therefore, we bound the inter-model excess risk as

$$\underbrace{\mu(m^N, \boldsymbol{\theta}_{m^N}^N) - \mu(m^*, \boldsymbol{\theta}_{m^*}^*)}_{\text{inter-model excess risk}} \le \left[ \mu(m^N, \boldsymbol{\theta}_{m^N}^N) - \mu(m^N, \boldsymbol{\theta}_{m^N}^*) \right] + 2\tilde{\Delta}. \tag{6}$$

STEP 2: BOUNDING $\left[ \mu(m^N, \boldsymbol{\theta}_{m^N}^N) - \mu(m^N, \boldsymbol{\theta}_{m^N}^*) \right] + 2\tilde{\Delta}$

We begin with examining $\tilde{\Delta}$.

$$\begin{aligned}
\tilde{\Delta} &= \max_{m \in \mathcal{M}_H} \left| \tilde{\mu}_N(m, \boldsymbol{\theta}_m^N) - \mu(m, \boldsymbol{\theta}_m^*) \right| \\
&= \max_{m \in \mathcal{M}_H} \left| \hat{\mu}_N(m, \boldsymbol{\theta}_m^N) + \lambda_N \|\boldsymbol{\theta}_m^N\|_1 - \mu(m, \boldsymbol{\theta}_m^*) \right| \\
&= \max_{m \in \mathcal{M}_H} \left| \hat{\mu}_N(m, \boldsymbol{\theta}_m^N) - \hat{\mu}_N(m, \boldsymbol{\theta}_m^*) + \lambda_N \|\boldsymbol{\theta}_m^N\|_1 + \hat{\mu}_N(m, \boldsymbol{\theta}_m^*) - \mu(m, \boldsymbol{\theta}_m^*) \right| \\
&\le \underbrace{\max_{m \in \mathcal{M}_H} \left| \hat{\mu}_N(m, \boldsymbol{\theta}_m^N) - \hat{\mu}_N(m, \boldsymbol{\theta}_m^*) + \lambda_N \|\boldsymbol{\theta}_m^N\|_1 \right|}_{(*)} + \underbrace{\max_{m \in \mathcal{M}_H} |\nu_N(m, \boldsymbol{\theta}_m^*)|}_{\tilde{\Delta}_1}.
\end{aligned}$$

We rewrite the first and second terms in $(*)$, $\hat{\mu}_N(m, \boldsymbol{\theta}_m^N) - \hat{\mu}_N(m, \boldsymbol{\theta}_m^*)$, using notations $\xi_{m,i} := y_i - \phi_m(x_i)^\top \boldsymbol{\theta}_m^*$ and $\zeta_{m,i} := (\boldsymbol{\theta}_m^* - \boldsymbol{\theta}_m^n)^\top \phi_m(\mathbf{x}_i) \phi_m(\mathbf{x}_i)^\top (\boldsymbol{\theta}_m^* - \boldsymbol{\theta}_m^n)$.

$$\hat{\mu}_N(m, \boldsymbol{\theta}_m^N) - \hat{\mu}_N(m, \boldsymbol{\theta}_m^*) = \frac{1}{N} \sum_{i=1}^N \left[ (y_i - \phi_m(x_i)^\top \boldsymbol{\theta}_m^N)^2 - (y_i - \phi_m(x_i)^\top \boldsymbol{\theta}_m^*)^2 \right]$$

$$= \frac{2}{N} \sum_{i=1}^N \xi_{m,i} \phi_m(\mathbf{x}_i)^\top (\boldsymbol{\theta}_m^* - \boldsymbol{\theta}_m^N) + \frac{1}{N} \sum_{i=1}^N \zeta_{m,i}.$$

Returning back to $(*)$, we obtain,

$$(*) = \max_{m \in \mathcal{M}_H} \left| \frac{2}{N} \sum_{i=1}^N \xi_{m,i} \phi_m(\mathbf{x}_i)^\top (\boldsymbol{\theta}_m^* - \boldsymbol{\theta}_m^N) + \frac{1}{N} \sum_{i=1}^N \zeta_{m,i} + \lambda_N \|\boldsymbol{\theta}_m^N\|_1 \right|$$

$$\leq \underbrace{\max_{m \in \mathcal{M}_H} \left| \frac{2}{N} \sum_{i=1}^N \xi_{m,i} \phi_m(\mathbf{x}_i)^\top (\boldsymbol{\theta}_m^* - \boldsymbol{\theta}_m^N) \right|}_{(**)} + \underbrace{\max_{m \in \mathcal{M}_H} \left| \frac{1}{N} \sum_{i=1}^N \zeta_{m,i} + \lambda_N \|\boldsymbol{\theta}_m^N\|_1 \right|}_{(***)}.$$

For $(**)$, under the event $\mathcal{T}_N$,

$$(**) \leq \max_{m \in \mathcal{M}_H} \left| 2\|\frac{1}{N} \sum_{i=1}^N \xi_{m,i} \phi(\mathbf{x}_i)\|_\infty \|\boldsymbol{\theta}_m^N - \boldsymbol{\theta}_m^*\|_1 \right| \leq \max_{m \in \mathcal{M}_H} 2\tilde{\lambda}_N \|\boldsymbol{\theta}_m^N - \boldsymbol{\theta}_m^*\|_1. \tag{7}$$

where the first inequality comes from Hölder's inequality.

To continue the proof, we introduce the following lemmas.

**Lemma 7.** *Under $\mathcal{T}_n$, for all $m \in [M]$, the following inequality holds:*

$$\frac{1}{n} \sum_{i=1}^n \zeta_{m,i} - \mathbb{E}[\zeta_{m,i}] \leq 4 \frac{\lambda_n^2 s_m}{\psi^2(S_m)}.$$

**Lemma 8.** *For each $m \in [M]$, the following holds:*

$$\mathbb{E}[\zeta_{m,i}] = \mathbb{E}_{\mathbf{x} \sim \mathcal{X}} \left[ \left( \phi_m(\mathbf{x})^\top \boldsymbol{\theta}_m^n - \phi_m(\mathbf{x})^\top \boldsymbol{\theta}_m^* \right)^2 \right] = \mu(m, \boldsymbol{\theta}_m^n) - \mu(m, \boldsymbol{\theta}_m^*).$$

Applying Lemmas 7 and 8 to $(***)$, we get

$$(***) \leq \max_{m \in \mathcal{M}_H} \left| 4 \frac{\lambda_N^2 s_m}{\psi^2(S_m)} + \mu(m, \boldsymbol{\theta}_m^N) - \mu(m, \boldsymbol{\theta}_m^*) + \lambda_N \|\boldsymbol{\theta}_m^N\|_1 \right|$$

$$\leq \max_{m \in \mathcal{M}_H} \left| 4 \frac{\lambda_N^2 s_m}{\psi^2(S_m)} + \mu(m, \boldsymbol{\theta}_m^N) - \mu(m, \boldsymbol{\theta}_m^*) + \lambda_N \|\boldsymbol{\theta}_m^*\|_1 + \lambda_N \|\boldsymbol{\theta}_m^N - \boldsymbol{\theta}_m^*\|_1 \right|$$

$$\leq \max_{m \in \mathcal{M}_H} 4 \frac{\lambda_N^2 s_m}{\psi^2(S_m)} + \underbrace{\max_{m \in \mathcal{M}_H} \lambda_N \|\boldsymbol{\theta}_m^*\|_1}_{\tilde{\Delta}_2}$$

$$+ \underbrace{\max_{m \in \mathcal{M}_H} \left[ \mu(m, \boldsymbol{\theta}_m^N) - \mu(m, \boldsymbol{\theta}_m^*) + \lambda_N \|\boldsymbol{\theta}_m^N - \boldsymbol{\theta}_m^*\|_1 \right]}_{\tilde{\Delta}_3} \tag{8}$$

Note that we omit the absolute value notation at the second inequality since $4 \frac{\lambda_N^2 s_m}{\psi^2(S_m)} \geq 0$, $\mu(m, \boldsymbol{\theta}_m^N) - \mu(m, \boldsymbol{\theta}_m^*) \geq 0$ and $\lambda_N \|\boldsymbol{\theta}_m^N\|_1 \geq 0$.

Plugging in (7) and (8) into $(*)$, and with the choice of $\tilde{\lambda}_n = \frac{\lambda_n}{8}$, which implies $\tilde{\lambda}_n \leq \frac{\lambda_n}{4}$, we obtain

$$(*) \leq (**) + (***)$$

$$\leq \max_{m \in \mathcal{M}_H} \left| 2\tilde{\lambda}_N \|\boldsymbol{\theta}_m^N - \boldsymbol{\theta}_m^*\|_1 + 4\frac{\lambda_N^2 S_{\bar{m}}}{\psi^2(S_{\bar{m}})} + \tilde{\Delta}_2 + \tilde{\Delta}_3 \right.$$

$$\leq \max_{m \in \mathcal{M}_H} \left| \frac{1}{2}\lambda_N \|\boldsymbol{\theta}_m^N - \boldsymbol{\theta}_m^*\|_1 \right| + 4\frac{\lambda_N^2 S_{\bar{m}}}{\psi^2(S_{\bar{m}})} + \tilde{\Delta}_2 + \tilde{\Delta}_3 \,.$$

remind that $\bar{m} := \operatorname{argmax}_{m \in [M]} s_m / \psi^2(S_m)$. Thus,

$$\tilde{\Delta} \leq (*) + \tilde{\Delta}_1 \leq \max_{m \in \mathcal{M}_H} \left| \frac{1}{2}\lambda_N \|\boldsymbol{\theta}_m^N - \boldsymbol{\theta}_m^*\|_1 \right| + 4\frac{\lambda_N^2 S_{\bar{m}}}{\psi^2(S_{\bar{m}})} + \tilde{\Delta}_1 + \tilde{\Delta}_2 + \tilde{\Delta}_3 \,. \tag{9}$$

Combining everything, we get,

$$\left[ \mu(m^N, \boldsymbol{\theta}_{m^N}^N) - \mu(m^N, \boldsymbol{\theta}_{m^N}^*) \right] + 2\tilde{\Delta}$$

$$\leq \left[ \mu(m^N, \boldsymbol{\theta}_{m^N}^N) - \mu(m^N, \boldsymbol{\theta}_{m^N}^*) \right] + \max_{m \in \mathcal{M}_H} \lambda_N \|\boldsymbol{\theta}_m^N - \boldsymbol{\theta}_m^*\|_1 + 2\tilde{\Delta}_1 + 2\tilde{\Delta}_2 + 2\tilde{\Delta}_3 + 8\frac{\lambda_N^2 S_{\bar{m}}}{\psi^2(S_{\bar{m}})}$$

$$\leq 2\tilde{\Delta}_1 + 2\tilde{\Delta}_2 + 4\tilde{\Delta}_3 + 8\frac{\lambda_N^2 S_{\bar{m}}}{\psi^2(S_{\bar{m}})} \,, \tag{10}$$

where the last inequality holds since $\left[ \mu(m, \boldsymbol{\theta}_m^N) - \mu(m, \boldsymbol{\theta}_m^*) \right] \leq \tilde{\Delta}_3$ and $\lambda_N \|\boldsymbol{\theta}_m^N - \boldsymbol{\theta}_m^*\|_1 \leq \tilde{\Delta}_3$ for all $m \in \mathcal{M}_H$.

Under the event $\mathscr{S}_N$, we have $\tilde{\Delta}_1 \leq \eta_N$. For $\tilde{\Delta}_2$, we have $\tilde{\Delta}_2 \leq \lambda_N R$ by Assumption 1. The last term $\tilde{\Delta}_3$ is commonly referred to as the oracle inequality. For $\tilde{\Delta}_3$, we invoke the following lemma:

**Lemma 9** (Oracle inequality for regression models)**.** *For any $h \in [H]$, let us denote $n_h = n$. Suppose that event $\mathscr{T}_n$ holds. Then, for any $m \in \mathcal{M}_h$,*

$$\mu(m, \boldsymbol{\theta}_m^n) - \mu(m, \boldsymbol{\theta}_m^*) + \lambda_n \|\boldsymbol{\theta}_m^n - \boldsymbol{\theta}_m^*\|_1 \leq 8\frac{\lambda_n^2 s_m}{\psi^2(S_m)} \,.$$

By Lemma 9, we have

$$\tilde{\Delta}_3 \leq \max_{m \in [M]} 8\frac{\lambda_N^2 s_m}{\psi^2(S_m)} = 8\frac{\lambda_N^2 s_{\bar{m}}}{\psi^2(S_{\bar{m}})} \,.$$

Remind that $\bar{m} = \operatorname{argmax}_{m \in [M]} s_m / \psi^2(S_m)$.

Combining the results of (6) and (10), we complete the proof of Lemma 3:

$$\underbrace{\mu(m^N, \boldsymbol{\theta}_{m^N}^N) - \mu(m^*, \boldsymbol{\theta}_{m^*}^*)}_{\text{inter-model excess risk}} \leq 2\tilde{\Delta}_1 + 2\tilde{\Delta}_2 + 4\tilde{\Delta}_3 + 8\frac{\lambda_N^2 s_{\bar{m}}}{\psi^2(S_{\bar{m}})}$$

$$= 2\eta_N + 2\lambda_N R + 40\frac{\lambda_N^2 s_{\bar{m}}}{\psi^2(S_{\bar{m}})} \,. \tag{11}$$

$\square$

## B.5 Main Proof of Theorem 1

*Proof of Theorem 1.* According to Lemma 1, the constraints on $\{\lambda_{n_h}\}_{h=1}^H$ and $\{\alpha_{n_h}\}_{h=1}^H$ specified in Theorem 1 ensure that the event $\mathscr{R}_H$ occurs with probability at least $1 - \delta$. The occurrence of event $\mathscr{R}_H$ implies the occurrence of event $\mathscr{R}_{H-1}$ and event $\mathscr{S}_N \cap \mathscr{T}_N$.

If $\mathscr{R}_{H-1}$ takes place, Lemma 2 holds as the confidence radii $\{\alpha_{n_h}\}_{h=1}^{H-1}$ are set to meet the conditions of Lemma 2. Hence, with high probability, the candidate pool retains the best model $m^*$ for the final epoch $H$, i.e., the selection step.

Additionally, when $m^* \in \mathcal{M}_H$ and if $\mathscr{S}_N \cap \mathscr{T}_N$ occurs, Lemma 3 applies. Therefore, with high probability, $\mu(m^N, \boldsymbol{\theta}_{m^N}^N) - \mu(m^*, \boldsymbol{\theta}_{m^*}^*) \leq \mathcal{O}(\frac{1}{\sqrt{N}})$.

Recall that $\mathscr{E} := \left\{ \mu(m^N, \boldsymbol{\theta}_{m^N}^N) - \mu(m^*, \boldsymbol{\theta}_{m^*}^*) > \epsilon \right\}$. Starting from the law of total probability, we have

$$
\begin{aligned}
\mathbb{P}(\mathscr{E}) &= \mathbb{P}(\mathscr{E} \mid \mathscr{R}_H)\mathbb{P}(\mathscr{R}_H) + \mathbb{P}(\mathscr{E} \mid \mathscr{R}_H^{\mathsf{c}})\mathbb{P}(\mathscr{R}_H^{\mathsf{c}}) \\
&\leq \mathbb{P}(\mathscr{E} \mid \mathscr{R}_H) + \mathbb{P}(\mathscr{R}_H^{\mathsf{c}}) \\
&\leq \mathbb{P}(\mathscr{E} \mid \mathscr{R}_H) + \delta \\
&= \mathbb{P}(\mathscr{E} \mid \mathscr{R}_{H-1}, \mathscr{S}_N, \mathscr{T}_N) + \delta,
\end{aligned}
\tag{12}
$$

where the second inequality results from Lemma 1. By Lemma 2, the occurrence of event $\mathscr{R}_{H-1}$ and the constraint on $\{\alpha_{n_h}\}_{h=1}^{H-1}$ guarantee that the best model survives for the selection step, $m^* \in \mathcal{M}_H$. For the selection step, we have

$$
\begin{aligned}
\mathbb{P}(\mathscr{E}) &\leq \mathbb{P}(\mathscr{E} \mid \mathscr{R}_{H-1}, \mathscr{S}_N, \mathscr{T}_N) + \delta \\
&\leq \mathbb{P}(\underbrace{2\eta_N + 2\lambda_N R + 40\frac{\lambda_N^2 s_{\bar{m}}}{\psi^2(S_{\bar{m}})} > \epsilon}_{(*)} \mid \mathscr{R}_{H-1}, \mathscr{S}_N, \mathscr{T}_N) + \delta \\
&\leq \delta
\end{aligned}
\tag{13}
$$

where the second inequality employs Lemma 3. Regarding to the last inequality, as $\eta_N$ and $\lambda_N$ both have $1/\sqrt{N}$ terms, by setting $N$ large enough, we can make the LHS of $(*)$ always be smaller than or equal to $\epsilon$, i.e., $\mathbb{P}((*) \mid \mathscr{R}_{H-1}, \mathscr{S}_N, \mathscr{T}_N) = 0$.

To summarize, the sample complexity is any number that exceeds $N$ satisfying the following statement:

$$
2\eta_N + 2\lambda_N R + 40\frac{\lambda_N^2 s_{\bar{m}}}{\psi^2(S_{\bar{m}})} \leq \epsilon.
\tag{14}
$$

From the proof of Lemma 1, we know that setting $\lambda_{n_h} = \sqrt{A/n_h}$ and $\alpha_{n_h} = \sqrt{B/n_h} + AC/4n_h$ implies setting $\eta_{n_h} = \sqrt{B/n_h}$ for all $h \in [H]$. Using $\lambda_N = \sqrt{A/N}$ and $\eta_N = \sqrt{B/N}$, we get

$$
2\sqrt{\frac{B}{N}} + 2\sqrt{\frac{AD}{N}} + \frac{AC}{N} = 2\eta_N + 2\lambda_N R + 40\frac{\lambda_N^2 s_{\bar{m}}}{\psi^2(S_{\bar{m}})}
\tag{15}
$$

where $A$, $B$, $C$, and $D$ are

$$
\begin{aligned}
A &= 64\phi_{\max}^2 \sigma^2 \log(4dMH/\delta), \\
B &= \frac{1}{2}(y_{\max} + \phi_{\max}R)^4 \log(4MH/\delta), \\
C &= 40\frac{s_{\bar{m}}}{\psi^2(S_{\bar{m}})}, \quad D = R^2.
\end{aligned}
$$

By combining (14) and (15), we get

$$2\sqrt{\frac{B}{N}} + 2\sqrt{\frac{AD}{N}} + \frac{AC}{N} \leq \epsilon$$

$$\Rightarrow AC + 2(\sqrt{AD} + \sqrt{B})\sqrt{N} \leq \epsilon N$$

$$\Rightarrow 2(\sqrt{AD} + \sqrt{B})\sqrt{N} \leq \epsilon N - AC$$

$$\Rightarrow 4(AD + B + 2\sqrt{ABD})N \leq \epsilon^2 N^2 - 2AC\epsilon N + A^2C^2$$

$$\Rightarrow \epsilon^2 N^2 - 2\left(AC\epsilon + 2AD + 2B + 4\sqrt{ABD}\right)N + A^2C^2 \geq 0$$

$$\Rightarrow N \geq \frac{1}{\epsilon^2}\Bigg[\left(AC\epsilon + 2AD + 2B + 4\sqrt{ABD}\right)$$

$$+ \sqrt{\left(AC\epsilon + 2AD + 2B + 4\sqrt{ABD}\right)^2 - \epsilon^2 A^2 C^2}\Bigg]$$

$$\Rightarrow N \geq \frac{2}{\epsilon^2}\left(AC\epsilon + 2AD + 2B + 4\sqrt{ABD}\right). \tag{16}$$

For such $N$, we have the desired result $\mathbb{P}(\mathscr{E}) \leq \delta$. $\qquad\square$

### B.6 Proofs for Technical Lemmas

#### B.6.1 Proof of Lemma 4

*Proof of Lemma 4.* First, we show that the complementary event of $\mathscr{S}_{n,m}$ is rare. We can write the complementary event as

$$\mathscr{S}_{n,m}^{\mathsf{c}} = \{|\hat{\mu}_n(m, \boldsymbol{\theta}_m^*) - \mu(m, \boldsymbol{\theta}_m^*)| > \eta_n\}\,,$$

where

$$\mu(m, \boldsymbol{\theta}_m^*) = \mathbb{E}\left[\ell(\boldsymbol{\phi}_m(\mathbf{x}_i)^\top \boldsymbol{\theta}_m^*, y_i)\right] \text{ and } \hat{\mu}_n(m, \boldsymbol{\theta}_m^*) = \frac{1}{n}\sum_{i=1}^{n}\left[\ell(\boldsymbol{\phi}_m(\mathbf{x}_i)^\top \boldsymbol{\theta}_m^*, y_i)\right]\,.$$

Since the data points $(\mathbf{x}_1, y_1), \cdots, (\mathbf{x}_n, y_n)$ are i.i.d., the losses $\ell(\boldsymbol{\phi}_m(\mathbf{x}_i)^\top \boldsymbol{\theta}_m^*, y_i), i \in [n]$ are independent too. Thus, by finding the bounds of the loss $\ell(\boldsymbol{\phi}_m(\mathbf{x}_i)^\top \boldsymbol{\theta}_m^*, y_i)$, we can apply Höeffding's inequality to upper bound the probability of $\mathscr{S}_{n,m}^{\mathsf{c}}$.

By Assumption 1, we have

$$0 \leq \ell(\boldsymbol{\phi}_m(\mathbf{x}_i)^\top \boldsymbol{\theta}_m^*, y_i) \leq (y_{\max} + \phi_{\max} R)^2\,.$$

Applying Höeffding's inequality, we get

$$\mathbb{P}\left(|\hat{\mu}_n(m, \boldsymbol{\theta}_m^*) - \mu(m, \boldsymbol{\theta}_m^*)| > \eta_n\right)$$

$$= \mathbb{P}\left(\left|\frac{1}{n}\sum_{i=1}^{n}\left[\ell(\boldsymbol{\phi}_m(\mathbf{x}_i)^\top \boldsymbol{\theta}_m^*, y_i)\right] - \mathbb{E}\left[\ell(\boldsymbol{\phi}_m(\mathbf{x}_i)^\top \boldsymbol{\theta}_m^*, y_i)\right]\right| > \eta_n\right)$$

$$\leq 2\exp\left(-\frac{2n^2\eta_n^2}{\sum_{i=1}^{n}(y_{\max} + \phi_{\max} R)^4}\right)$$

$$= 2\exp\left(-\frac{2n\eta_n^2}{(y_{\max} + \phi_{\max} R)^4}\right)$$

$$= \delta_0$$

for some $\delta_0 \in (0, 1)$.

Rearranging the last inequality with respect to $\eta_n$, we get

$$\eta_n = \sqrt{\frac{(y_{\max} + \phi_{\max} R)^4 \log(2/\delta_0)}{2n}} \tag{17}$$

Thus, if $\eta_n$ satisfies (17), for given $m \in \mathcal{M}_h$ we have $\mathbb{P}\left(\mathscr{S}_{n,m}^{\mathsf{c}}\right) \leq \delta_0$.

Now for every $m \in \mathcal{M}_h$, we have

$$\mathbb{P}(\mathscr{S}_n) = 1 - \mathbb{P}(\mathscr{S}_n^{\mathsf{c}})$$

$$= 1 - \mathbb{P}\left(\bigcup_{m \in \mathcal{M}_h} \mathscr{S}_{n,m}^{\mathsf{c}}\right)$$

$$\geq 1 - \sum_{m \in \mathcal{M}_h} \mathbb{P}\left(\mathscr{S}_{n,m}^{\mathsf{c}}\right)$$

$$\geq 1 - M_h \delta_0$$

$$= 1 - \delta_1 \,,$$

where the first inequality comes from union bound. Replacing $\delta_0$ in $\eta_n$ as $\delta_1/M_h$, we get,

$$\eta_n = \sqrt{\frac{(y_{\max} + \phi_{\max}R)^4 \log(2M_h/\delta_1)}{2n}} \,. \tag{18}$$

Thus, if $\eta_n$ satisfies (18), $\mathbb{P}\left(\mathscr{S}_n\right) \geq 1 - \delta_1$. $\qquad\square$

### B.6.2 PROOF OF LEMMA 5

*Proof of Lemam 5.* The proof strategy for this Lemma basically follows that of Lemma 4. First, for a given $\delta_0 \in (0, 1)$, we will show that $\mathbb{P}\left(\mathscr{T}_{n,m}^{\mathsf{c}}\right) < \delta_0$. The complementary event is

$$\mathscr{T}_m^{\mathsf{c}} = \left\{\|\frac{1}{n}\sum_{i=1}^n \xi_{m,i}\boldsymbol{\phi}_m(\mathbf{x}_i)\|_\infty > \tilde{\lambda}_n\right\} \,.$$

For each $m \in [M]$ and $j \in [d_m]$, let $z_{m,i,j} := \xi_{m,i}[\boldsymbol{\phi}_m(\mathbf{x}_i)]_j$. As $[\boldsymbol{\phi}_m(\mathbf{x}_i)]_j$ is bounded with $[\boldsymbol{\phi}_m(\mathbf{x}_i)]_j \leq \|\boldsymbol{\phi}_m(\mathbf{x}_i)\| \leq \phi_{\max}$, and $\xi_{m,i}$ is $\sigma^2$-sub-Gaussian by Assumption 2, it follows that $z_{m,i,j}$ is also sub-Gaussian. Specifically, for any $\alpha \in \mathbb{R}$, $\mathbb{E}\left[\exp(\alpha z_{m,i,j})\right] \leq \exp(\alpha^2 \phi_{\max}^2 \sigma^2/2)$.

Applying union bound, we get,

$$\mathbb{P}\left(\|\frac{1}{n}\sum_{i=1}^n \xi_{m,i}\boldsymbol{\phi}_m(\mathbf{x}_i)\|_\infty > \tilde{\lambda}_n\right) = \mathbb{P}\left(\max_{j \in [d_m]} |\frac{1}{n}\sum_{i=1}^n z_{m,i,j}| > \tilde{\lambda}_n\right)$$

$$\leq \sum_{j=1}^{d_m} \mathbb{P}\left(|\frac{1}{n}\sum_{i=1}^n z_{m,i,j}| > \tilde{\lambda}_n\right)$$

$$\leq 2d \exp\left(-\frac{n\tilde{\lambda}_n^2}{\phi_{\max}^2 \sigma^2}\right)$$

$$\leq \delta_0 \,,$$

where $d = \max_{m \in [M]} d_m$. Rearranging the last inequality with respect to $\tilde{\lambda}_n$, we get

$$\tilde{\lambda}_n = \sqrt{\frac{\phi_{\max}^2 \sigma^2 \log(2d/\delta_0)}{n}} \,. \tag{19}$$

Thus, if $\tilde{\lambda}_n$ satisfies (19), for each $m \in \mathcal{M}_h$ we have $\mathbb{P}\left(\mathscr{T}_{n,m}^{\mathsf{c}}\right) \leq \delta_0$.

Now for every model, we have

$$\mathbb{P}(\mathscr{T}_n) = 1 - \mathbb{P}(\mathscr{T}_n^{\mathsf{c}})$$

$$= 1 - \mathbb{P}\left(\bigcup_{m \in \mathcal{M}_h} \mathscr{T}_{n,m}^{\mathsf{c}}\right)$$

$$\geq 1 - \sum_{m \in \mathcal{M}_h} \mathbb{P}\left(\mathscr{T}_{n,m}^{\mathsf{c}}\right)$$

$$\geq 1 - M_h \delta_0$$

$$= 1 - \delta_2 \,,$$

where the first inequality comes from union bound. Replacing $\delta_0$ in $\tilde{\lambda}_n$ as $\delta_2/M_h$, we get,

$$\tilde{\lambda}_n = \sqrt{\frac{\phi_{\max}^2 \sigma^2 \log(2dM_h/\delta_2)}{n}} \, . \tag{20}$$

Thus, if $\tilde{\lambda}_n$ satisfies (20), $\mathbb{P}\left(\mathscr{T}_n\right) \geq 1 - \delta_2$ . $\qquad \square$

### B.6.3 PROOF OF LEMMA 6

*Proof of Lemma 6.* Define $\hat{\Delta}_n := \max_{m \in \mathcal{M}_h} |\hat{\mu}_n(m, \boldsymbol{\theta}_m^n) - \mu(m, \boldsymbol{\theta}_m^*)|$. Since $\mu(m, \boldsymbol{\theta}_m^*)$ belongs to the following interval for any $m \in \mathcal{M}_h$,

$$\mu(m, \boldsymbol{\theta}_m^*) \in \left[\hat{\mu}_n(m, \boldsymbol{\theta}_m^n) - |\hat{\mu}_n(m, \boldsymbol{\theta}_m^n) - \mu(m, \boldsymbol{\theta}_m^*)| , \hat{\mu}_n(m, \boldsymbol{\theta}_m^n) + |\hat{\mu}_n(m, \boldsymbol{\theta}_m^n) - \mu(m, \boldsymbol{\theta}_m^*)|\right],$$

we have,

$$\hat{\mu}_n(m, \boldsymbol{\theta}_m^n) - \hat{\Delta}_n \leq \mu(m, \boldsymbol{\theta}_m^*) \leq \hat{\mu}_n(m, \boldsymbol{\theta}_m^n) + \hat{\Delta}_n \, .$$

Notice that

$$
\begin{aligned}
\hat{\Delta}_n &= \max_{m \in \mathcal{M}_h} |\tilde{\mu}_n(m, \boldsymbol{\theta}_m^n) - \mu(m, \boldsymbol{\theta}_m^*)| \\
&= \max_{m \in \mathcal{M}_h} |\hat{\mu}_n(m, \boldsymbol{\theta}_m^n) - \mu(m, \boldsymbol{\theta}_m^n) + \mu(m, \boldsymbol{\theta}_m^n) - \mu(m, \boldsymbol{\theta}_m^*)| \\
&\leq \max_{m \in \mathcal{M}_h} |\hat{\mu}_n(m, \boldsymbol{\theta}_m^n) - \mu(m, \boldsymbol{\theta}_m^n)| + \max_{m \in \mathcal{M}_h} |\mu(m, \boldsymbol{\theta}_m^n) - \mu(m, \boldsymbol{\theta}_m^*)| \\
&\leq \underbrace{\max_{m \in \mathcal{M}_h} |\hat{\mu}_n(m, \boldsymbol{\theta}_m^n) - \mu(m, \boldsymbol{\theta}_m^n)|}_{\hat{\Delta}_{n,1}} + \underbrace{\max_{m \in \mathcal{M}_h} \mu(m, \boldsymbol{\theta}_m^n) - \mu(m, \boldsymbol{\theta}_m^*) + \lambda_n \|\boldsymbol{\theta}_m^n - \boldsymbol{\theta}_m^*\|_1}_{\hat{\Delta}_{n,2}}
\end{aligned}
$$

Note that under the event $\mathscr{S}_n$, we have $\hat{\Delta}_{n,1} \leq \eta_n$.

By Lemma 9, we have

$$\hat{\Delta}_{n,2} \leq \max_{m \in [M]} 8\frac{\lambda_n^2 s_m}{\psi^2(S_m)} = 8\frac{\lambda_n^2 s_{\bar{m}}}{\psi^2(S_{\bar{m}})} \, ,$$

where $\bar{m} = \operatorname{argmax}_{m \in [M]} s_m/\psi^2(S_m)$.

Combining the upper bounds of $\hat{\Delta}_{n,1}$ and $\hat{\Delta}_{n,2}$, we have

$$\Delta_n \leq \eta_n + 8\frac{\lambda_n^2 s_{\bar{m}}}{\psi^2(S_{\bar{m}})} \, . \tag{21}$$

Therefore, under the event $\mathscr{T}_n$, for all $m \in \mathcal{M}_h$, the following holds:

$$\tilde{\mu}_n(m, \boldsymbol{\theta}_m^n) - \alpha_n \leq \mu(m, \boldsymbol{\theta}_m^*) \leq \tilde{\mu}_n(m, \boldsymbol{\theta}_m^n) + \alpha_n \, ,$$

where $\alpha_n = \eta_n + 8\frac{\lambda_n^2 s_{\bar{m}}}{\psi^2(S_{\bar{m}})}$. $\qquad \square$

### B.6.4 PROOF OF LEMMA 7

*Proof of Lemma 7.* The key idea is to start from (29) but derive a basic inequality different to (30). Note that $\xi_{m,i} := y_i - \boldsymbol{\phi}_m(x_i)^\top \boldsymbol{\theta}_m^*$ and $\zeta_{m,i} := (\boldsymbol{\theta}_m^* - \boldsymbol{\theta}_m^n)^\top \boldsymbol{\phi}_m(\mathbf{x}_i)\boldsymbol{\phi}_m(\mathbf{x}_i)^\top (\boldsymbol{\theta}_m^* - \boldsymbol{\theta}_m^n)$. For each $m \in \mathcal{M}_h$,

$$\hat{\mu}_n(m, \boldsymbol{\theta}_m^n) + \lambda_n \|\boldsymbol{\theta}_m^n\|_1 \leq \hat{\mu}_n(m, \boldsymbol{\theta}_m^*) + \lambda_n \|\boldsymbol{\theta}_m^*\|_1$$

$$\frac{1}{n}\sum_{i=1}^n \left[\left(y_i - \boldsymbol{\phi}_m(\mathbf{x}_i)^\top \boldsymbol{\theta}_m^n\right)^2 - \left(y_i - \boldsymbol{\phi}_m(\mathbf{x}_i)^\top \boldsymbol{\theta}_m^*\right)^2\right] \leq \lambda_n \|\boldsymbol{\theta}_m^*\|_1 - \lambda_n \|\boldsymbol{\theta}_m^n\|_1$$

$$\frac{1}{n}\sum_{i=1}^n \zeta_{m,i} - \frac{2}{n}\sum_{i=1}^n \xi_{m,i}\boldsymbol{\phi}_m(\mathbf{x}_i)^\top(\boldsymbol{\theta}_m^n - \boldsymbol{\theta}_m^*) \leq \lambda_n \|\boldsymbol{\theta}_m^*\|_1 - \lambda_n \|\boldsymbol{\theta}_m^n\|_1 \, . \tag{22}$$

Then, under $\mathscr{T}_n$,

$$\frac{1}{n}\sum_{i=1}^{n}\zeta_{m,i} \leq \frac{2}{n}\sum_{i=1}^{n}\xi_{m,i}\phi_m(\mathbf{x}_i)^\top(\boldsymbol{\theta}_m^n - \boldsymbol{\theta}_m^*) + \lambda_n\|\boldsymbol{\theta}_m^*\|_1 - \lambda_n\|\boldsymbol{\theta}_m^n\|_1$$

$$\leq \left|\frac{2}{n}\sum_{i=1}^{n}\xi_{m,i}\phi_m(\mathbf{x}_i)^\top(\boldsymbol{\theta}_m^n - \boldsymbol{\theta}_m^*)\right| + \lambda_n\|\boldsymbol{\theta}_m^*\|_1 - \lambda_n\|\boldsymbol{\theta}_m^n\|_1$$

$$\leq 2\|\frac{1}{n}\sum_{i=1}^{n}\xi_{m,i}\phi_m(\mathbf{x}_i)\|_\infty\|\boldsymbol{\theta}_m^n - \boldsymbol{\theta}_m^*\|_1 + \lambda_n\|\boldsymbol{\theta}_m^*\|_1 - \lambda_n\|\boldsymbol{\theta}_m^n\|_1$$

$$\leq 2\tilde{\lambda}_n\|\boldsymbol{\theta}_m^n - \boldsymbol{\theta}_m^*\|_1 + \lambda_n\|\boldsymbol{\theta}_m^*\|_1 - \lambda_n\|\boldsymbol{\theta}_m^n\|_1\,,$$

where we use Hölder's inequality in the second inequality. Next, as the choice of $\tilde{\lambda}_n = \frac{\lambda_n}{8}$ implies $\tilde{\lambda}_n \leq \frac{\lambda_n}{4}$, we obtain

$$\frac{1}{n}\sum_{i=1}^{n}\zeta_{m,i} \leq 2\tilde{\lambda}_n\|\boldsymbol{\theta}_m^* - \boldsymbol{\theta}_m^n\|_1 + \lambda_n\|\boldsymbol{\theta}_m^*\|_1 - \lambda_n\|\boldsymbol{\theta}_m^n\|_1$$

$$\leq \frac{1}{2}\lambda_n\|\boldsymbol{\theta}_m^n - \boldsymbol{\theta}_m^*\|_1 + \lambda_n\|\boldsymbol{\theta}_m^*\|_1 - \lambda_n\|\boldsymbol{\theta}_m^n\|_1$$

$$= \frac{\lambda_n}{2}\|\boldsymbol{\theta}_{m,S_m}^n - \boldsymbol{\theta}_{m,S_m}^*\|_1 + \frac{\lambda_n}{2}\|\boldsymbol{\theta}_{m,S_m^c}^n - \boldsymbol{\theta}_{m,S_m^c}^*\|_1$$
$$+ \lambda_n\|\boldsymbol{\theta}_{m,S_m}^*\|_1 - \lambda_n\|\boldsymbol{\theta}_{m,S_m}^n\|_1 - \lambda_n\|\boldsymbol{\theta}_{m,S_m^c}^n\|_1$$

$$\leq \frac{3\lambda_n}{2}\|\boldsymbol{\theta}_{m,S_m}^n - \boldsymbol{\theta}_{m,S_m}^*\|_1 + \frac{\lambda_n}{2}\|\boldsymbol{\theta}_{m,S_m^c}^n - \boldsymbol{\theta}_{m,S_m^c}^*\|_1 - \lambda_n\|\boldsymbol{\theta}_{m,S_m^c}^n\|_1$$

$$= \frac{3\lambda_n}{2}\|\boldsymbol{\theta}_{m,S_m}^n - \boldsymbol{\theta}_{m,S_m}^*\|_1 - \frac{\lambda_n}{2}\|\boldsymbol{\theta}_{m,S_m^c}^n - \boldsymbol{\theta}_{m,S_m^c}^*\|_1\,. \quad (23)$$

Since $\frac{1}{n}\sum_{i=1}^{n}\zeta_{m,i} \geq 0$, from (23) we get $\|\boldsymbol{\theta}_{m,S_m^c}^n - \boldsymbol{\theta}_{m,S_m^c}^*\|_1 \leq 3\|\boldsymbol{\theta}_{m,S_m}^n - \boldsymbol{\theta}_{m,S_m}^*\|_1$. This is the point where we reach at the conclusion for Lemma 10.

Continuing the proof, by Assumption 3, we have

$$\psi^2(S_m) \leq \frac{s_m(\boldsymbol{\theta}_m^n - \boldsymbol{\theta}_m^*)^\top \mathbb{E}_{\mathbf{x}\sim\mathcal{X}}\left[\phi_m(\mathbf{x})\phi_m(\mathbf{x})^\top\right](\boldsymbol{\theta}_m^n - \boldsymbol{\theta}_m^*)}{\|\boldsymbol{\theta}_{m,S_m}^n - \boldsymbol{\theta}_{m,S_m}^*\|_1^2}\,. \quad (24)$$

By adding $\lambda\|\boldsymbol{\theta}_{m,S_m}^n - \boldsymbol{\theta}_{m,S_m}^*\|_1$ to both sides of (23) and by rearranging, we get

$$\frac{2}{n}\sum_{i=1}^{n}\zeta_{m,i} + \lambda_n\|\boldsymbol{\theta}_m^n - \boldsymbol{\theta}_m^*\|_1 \leq 4\lambda_n\|\boldsymbol{\theta}_{m,S_m}^n - \boldsymbol{\theta}_{m,S_m}^*\|_1\,. \quad (25)$$

We lower the LHS of (25) using $\frac{2}{n}\sum_{i=1}^{n}\zeta_{m,i} \geq \frac{1}{n}\sum_{i=1}^{n}\zeta_{m,i} \geq 0$

$$\frac{1}{n}\sum_{i=1}^{n}\zeta_{m,i} + \lambda_n\|\boldsymbol{\theta}_m^n - \boldsymbol{\theta}_m^*\|_1 \leq \frac{2}{n}\sum_{i=1}^{n}\zeta_{m,i} + \lambda_n\|\boldsymbol{\theta}_m^n - \boldsymbol{\theta}_m^*\|_1 \leq 4\lambda_n\|\boldsymbol{\theta}_{m,S_m}^n - \boldsymbol{\theta}_{m,S_m}^*\|_1\,. \quad (26)$$

By plugging (24) into (26), we have

$$\frac{1}{n}\sum_{i=1}^{n}\zeta_{m,i} + \lambda_n\|\boldsymbol{\theta}_m^n - \boldsymbol{\theta}_m^*\|_1$$

$$\leq 4\lambda_n\sqrt{(\boldsymbol{\theta}_m^n - \boldsymbol{\theta}_m^*)^\top\mathbb{E}_{\mathbf{x}\sim\mathcal{X}}\left[\phi_m(\mathbf{x})\phi_m(\mathbf{x})^\top\right](\boldsymbol{\theta}_m^n - \boldsymbol{\theta}_m^*)}\frac{\sqrt{s_m}}{\psi(S_m)}$$

$$= 4\lambda_n\sqrt{\mathbb{E}_{\mathbf{x}\sim\mathcal{X}}[\underbrace{\left(\phi_m(\mathbf{x})^\top\boldsymbol{\theta}_m^n - \phi_m(\mathbf{x})^\top\boldsymbol{\theta}_m^*\right)^2}_{\zeta_{m,i}}]}\frac{\sqrt{s_m}}{\psi(S_m)}$$

$$\leq \mathbb{E}[\zeta_{m,i}] + 4\frac{\lambda_n^2 s_m}{\psi^2(S_m)}\,, \quad (27)$$

where the last inequality follows by using $4uv \leq u^2 + 4v^2$ with

$$u = \sqrt{\mathbb{E}\left[(\phi_m(\mathbf{x})^\top \theta_m^n - \phi_m(\mathbf{x})^\top \theta_m^*)^2\right]} \text{ and}$$

$$v = \lambda_n \sqrt{s_m}/\psi(S_m).$$

We conclude with rearranging (27),

$$\frac{1}{n}\sum_{i=1}^n \zeta_{m,i} - \mathbb{E}[\zeta_{m,i}] \leq \frac{1}{n}\sum_{i=1}^n \zeta_{m,i} + \lambda_n \|\theta_m^n - \theta_m^*\|_1 - \mathbb{E}[\zeta_{m,i}] \leq 4\frac{\lambda_n^2 s_m}{\psi^2(S_m)}.$$

$\square$

### B.6.5 PROOF OF LEMMA 8

*Proof of Lemma 8.* Let $\mathcal{E}(\theta_m^N) := \mu(m, \theta_m^N) - \mu(m, \theta_m^*)$. Then for the mean square error loss, the 2nd order Taylor expansion of $\mathcal{E}(\theta_m^N)$ with respect to $\theta_m^*$ is given by

$$\mathcal{E}(\theta_m^N) = \mathcal{E}(\theta_m^*) + (\nabla_\theta \mathcal{E}(\theta_m^*))^\top (\theta_m^N - \theta_m^*) + \frac{1}{2}(\theta_m^N - \theta_m^*)^\top \nabla_\theta^2 \mathcal{E}(\theta_m^*)(\theta_m^N - \theta_m^*)$$

$$= \frac{1}{2}(\theta_m^N - \theta_m^*)^\top \nabla_\theta^2 \mathcal{E}(\theta_m^*)(\theta_m^N - \theta_m^*), \tag{28}$$

where the second equality holds since $\mathcal{E}(\theta_m^*) = 0$ and since $\mathcal{E}(\theta)$ is a non-negative quadratic function with respect to $\theta$, the minimality of $\mathcal{E}(\theta_m^*)$ implies $\nabla_\theta \mathcal{E}(\theta_m^*) = \mathbf{0}_{d_m}$.

Note that we have

$$\nabla_\theta \mathcal{E}(\theta) = \nabla_\theta \mu(m, \theta) = \nabla_\theta \mathbb{E}\left[(y - \phi_m(\mathbf{x})^\top \theta)^2\right] = \mathbb{E}\left[-2(y - \phi_m(\mathbf{x})^\top \theta)\phi_m(\mathbf{x})\right],$$

and

$$\nabla_\theta^2 \mathcal{E}(\theta) = 2\mathbb{E}\left[\phi_m(\mathbf{x})\phi_m(\mathbf{x})^\top\right].$$

Then, from (28) we have

$$\mathcal{E}(\theta_m^N) = \mathbb{E}\left[\phi_m(\mathbf{x})^\top (\theta_m^N - \theta_m^*)\phi_m(\mathbf{x})^\top (\theta_m^N - \theta_m^*)\right].$$

$\square$

### B.6.6 PROOF OF LEMMA 9

*Proof of Lemma 9.* Utilizing the methodologies outlined in (Bühlmann & Van De Geer, 2011), we derive an upper bound for $\mu(m, \theta_m^n) - \mu(m, \theta_m^*) + \lambda \|\theta_m^n - \theta_m^*\|_1$.

For each $m \in \mathcal{M}_h$, since we have

$$\hat{\mu}_n(m, \theta_m^n) + \lambda_n \|\theta_m^n\|_1 \leq \hat{\mu}_n(m, \theta_m^*) + \lambda_n \|\theta_m^*\|_1. \tag{29}$$

By adding and subtracting $\mu(m, \theta_m^n)$ and $\mu(m, \theta_m^*)$ to both sides respectively, we have

$$\underbrace{\hat{\mu}_n(m, \theta_m^n) - \mu(m, \theta_m^n)}_{\nu_n(m, \theta_m^n)} + \mu(m, \theta_m^n) + \lambda_n \|\theta_m^n\|_1$$

$$\leq \underbrace{\hat{\mu}_n(m, \theta_m^*) - \mu(m, \theta_m^*)}_{\nu_n(m, \theta_m^*)} + \mu(m, \theta_m^*) + \lambda_n \|\theta_m^*\|_1.$$

By rearranging the terms we obtain a basic inequality:

$$\mu(m, \theta_m^n) - \mu(m, \theta_m^*) \leq \nu_n(m, \theta_m^*) - \nu_n(m, \theta_m^n) + \lambda_n \|\theta_m^*\|_1 - \lambda_n \|\theta_m^n\|_1. \tag{30}$$

As the loss is the squared error $\ell(\phi_m(\mathbf{x})^\top \theta, y) = (y - \phi_m(\mathbf{x})^\top \theta)^2$, we can write the concentration of the empirical process as

$$\nu_n(m, \theta_m^*) - \nu_n(m, \theta_m^n) = \frac{1}{n}\sum_{i=1}^n \chi_{m,i} - \mathbb{E}[\chi_{m,i}] + \frac{1}{n}\sum_{i=1}^n \zeta_{m,i} - \mathbb{E}[\zeta_{m,i}], \tag{31}$$

where $\chi_{m,i} := 2\xi_{m,i}\phi_m(\mathbf{x}_i)^\top(\boldsymbol{\theta}_m^* - \boldsymbol{\theta}_m^n)$ and $\zeta_{m,i} := (\boldsymbol{\theta}_m^* - \boldsymbol{\theta}_m^n)^\top\phi_m(\mathbf{x}_i)\phi_m(\mathbf{x}_i)^\top(\boldsymbol{\theta}_m^* - \boldsymbol{\theta}_m^n)$.
For the first concentration term of (31),

$$
\begin{aligned}
\frac{1}{n}\sum_{i=1}^n \chi_{m,i} - \mathbb{E}[\chi_{m,i}] &= \frac{1}{n}\sum_{i=1}^n \chi_{m,i} - \mathbb{E}[\frac{1}{n}\sum_{i=1}^n \chi_{m,i}] \\
&\leq \left|\frac{1}{n}\sum_{i=1}^n \chi_{m,i}\right| + \left|\mathbb{E}[\frac{1}{n}\sum_{i=1}^n \chi_{m,i}]\right| \\
&\leq \left|\frac{1}{n}\sum_{i=1}^n \chi_{m,i}\right| + \mathbb{E}[\left|\frac{1}{n}\sum_{i=1}^n \chi_{m,i}\right|] \\
&\leq 4\tilde{\lambda}_n\|\boldsymbol{\theta}_m^n - \boldsymbol{\theta}_m^*\|_1,
\end{aligned}
\tag{32}
$$

where the second inequality comes from Jensen's inequality. The last inequality holds since, under the occurrence of event $\mathscr{T}_n$,

$$
\left|\frac{1}{n}\sum_{i=1}^n \chi_{m,i}\right| = \left|\frac{2}{n}\sum_{i=1}^n \xi_{m,i}\phi(\mathbf{x}_i)^\top(\boldsymbol{\theta}_m^* - \boldsymbol{\theta}_m^n)\right| \leq 2\|\frac{1}{n}\sum_{i=1}^n \xi_{m,i}\phi(\mathbf{x}_i)\|_\infty\|\boldsymbol{\theta}_m^* - \boldsymbol{\theta}_m^n\|_1
$$

$$
\leq 2\tilde{\lambda}_n\|\boldsymbol{\theta}_m^n - \boldsymbol{\theta}_m^*\|_1.
$$

Plugging in (32) and the result of Lemma 7 to (31), we obtain,

$$
\nu_n(m, \boldsymbol{\theta}_m^*) - \nu_n(m, \boldsymbol{\theta}_m^n) \leq 4\tilde{\lambda}_n\|\boldsymbol{\theta}_m^n - \boldsymbol{\theta}_m^*\|_1 + 4\frac{\lambda_n^2 s_m}{\psi^2(S_m)}.
\tag{33}
$$

Inserting (33) to (30) and by the choice of $\tilde{\lambda}_n = \frac{\lambda_n}{8}$ we get,

$$
\begin{aligned}
&\mu(m, \boldsymbol{\theta}_m^n) - \mu(m, \boldsymbol{\theta}_m^*) \\
&\leq 4\tilde{\lambda}_n\|\boldsymbol{\theta}_m^n - \boldsymbol{\theta}_m^*\|_1 + \lambda_n\|\boldsymbol{\theta}_m^*\|_1 - \lambda_n\|\boldsymbol{\theta}_m^n\|_1 + 4\frac{\lambda_n^2 s_m}{\psi^2(S_m)} \\
&\leq \frac{1}{2}\lambda_n\|\boldsymbol{\theta}_m^n - \boldsymbol{\theta}_m^*\|_1 + \lambda_n\|\boldsymbol{\theta}_m^*\|_1 - \lambda_n\|\boldsymbol{\theta}_m^n\|_1 + 4\frac{\lambda_n^2 s_m}{\psi^2(S_m)} \\
&= \frac{\lambda_n}{2}\|\boldsymbol{\theta}_{m,S_m}^n - \boldsymbol{\theta}_{m,S_m}^*\|_1 + \frac{\lambda_n}{2}\|\boldsymbol{\theta}_{m,S_m^c}^n - \boldsymbol{\theta}_{m,S_m^c}^*\|_1 \\
&\quad + \lambda_n\|\boldsymbol{\theta}_{m,S_m}^*\|_1 - \lambda_n\|\boldsymbol{\theta}_{m,S_m}^n\|_1 - \lambda_n\|\boldsymbol{\theta}_{m,S_m^c}^n\|_1 + 4\frac{\lambda_n^2 s_m}{\psi^2(S_m)} \\
&\leq \frac{3\lambda_n}{2}\|\boldsymbol{\theta}_{m,S_m}^n - \boldsymbol{\theta}_{m,S_m}^*\|_1 + \frac{\lambda_n}{2}\|\boldsymbol{\theta}_{m,S_m^c}^n - \boldsymbol{\theta}_{m,S_m^c}^*\|_1 - \lambda_n\|\boldsymbol{\theta}_{m,S_m^c}^n\|_1 + 4\frac{\lambda_n^2 s_m}{\psi^2(S_m)} \\
&= \frac{3\lambda_n}{2}\|\boldsymbol{\theta}_{m,S_m}^n - \boldsymbol{\theta}_{m,S_m}^*\|_1 - \frac{\lambda_n}{2}\|\boldsymbol{\theta}_{m,S_m^c}^n - \boldsymbol{\theta}_{m,S_m^c}^*\|_1 + 4\frac{\lambda_n^2 s_m}{\psi^2(S_m)},
\end{aligned}
$$

where the last comes from $\lambda_n\|\boldsymbol{\theta}_{m,S_m}^*\|_1 - \lambda_n\|\boldsymbol{\theta}_{m,S_m}^n\|_1 \leq \lambda_n\|\boldsymbol{\theta}_{m,S_m}^* - \boldsymbol{\theta}_{m,S_m}^n\|_1$ and the equality uses the property that $\|\boldsymbol{\theta}_m^n\|_1 = \|\boldsymbol{\theta}_{m,S_m}^n\|_1 + \|\boldsymbol{\theta}_{m,S_m^c}^n\|_1$.

To continue the proof, we use the following lemma:

**Lemma 10.** *Under $\mathscr{T}_n$, for all $m \in [M]$, $(\boldsymbol{\theta}_m^n - \boldsymbol{\theta}_m^*)$ meets the compatibility condition, i.e.,*

$$
\|\boldsymbol{\theta}_{m,S^c}^n - \boldsymbol{\theta}_{m,S^c}^*\|_1 \leq 3\|\boldsymbol{\theta}_{m,S}^n - \boldsymbol{\theta}_{m,S}^*\|_1.
$$

Note that the proof of Lemma 10 is included in the proof of Lemma 7 at Section B.6.4.

By Lemma 10, $\boldsymbol{\theta}_m^n - \boldsymbol{\theta}_m^* \in \mathbb{C}(S_m)$. Thus, by Assumption 3, we have

$$
\psi^2(S_m) \leq \frac{s_m(\boldsymbol{\theta}_m^n - \boldsymbol{\theta}_m^*)^\top \mathbb{E}_{\mathbf{x} \sim \mathcal{X}}\left[\phi_m(\mathbf{x})\phi_m(\mathbf{x})^\top\right](\boldsymbol{\theta}_m^n - \boldsymbol{\theta}_m^*)}{\|\boldsymbol{\theta}_{m,S_m}^n - \boldsymbol{\theta}_{m,S_m}^*\|_1^2}.
\tag{34}
$$

By adding $\lambda\|\boldsymbol{\theta}_{m,S_m}^n - \boldsymbol{\theta}_{m,S_m}^*\|_1$ to both sides and by rearranging we have

$$2\left[\mu(m,\boldsymbol{\theta}_m^n) - \mu(m,\boldsymbol{\theta}_m^*)\right] + \underbrace{\lambda_n\|\boldsymbol{\theta}_{m,S_m^c}^n - \boldsymbol{\theta}_{m,S_m^c}^*\|_1 + \lambda_n\|\boldsymbol{\theta}_{m,S_m}^n - \boldsymbol{\theta}_{m,S_m}^*\|_1}_{\lambda_n\|\boldsymbol{\theta}_m^n - \boldsymbol{\theta}_m^*\|_1}$$

$$\leq 4\lambda_n\|\boldsymbol{\theta}_{m,S_m}^n - \boldsymbol{\theta}_{m,S_m}^*\|_1 + 4\frac{\lambda_n^2 s_m}{\psi^2(S_m)}\,. \tag{35}$$

By plugging (34) into (35), we have

$$2\left[\mu(m,\boldsymbol{\theta}_m^n) - \mu(m,\boldsymbol{\theta}_m^*)\right] + \lambda_n\|\boldsymbol{\theta}_m^n - \boldsymbol{\theta}_m^*\|_1 \tag{36}$$

$$\leq 4\lambda_n\sqrt{(\boldsymbol{\theta}_m^n - \boldsymbol{\theta}_m^*)^\top \mathbb{E}_{\mathbf{x}\sim\mathcal{X}}\left[\boldsymbol{\phi}_m(\mathbf{x})\boldsymbol{\phi}_m(\mathbf{x})^\top\right](\boldsymbol{\theta}_m^n - \boldsymbol{\theta}_m^*)}\frac{\sqrt{s_m}}{\psi(S_m)} + 4\frac{\lambda_n^2 s_m}{\psi^2(S_m)}$$

$$= 4\lambda_n\sqrt{\mathbb{E}_{\mathbf{x}\sim\mathcal{X}}\left[(\boldsymbol{\phi}_m(\mathbf{x})^\top\boldsymbol{\theta}_m^n - \boldsymbol{\phi}_m(\mathbf{x})^\top\boldsymbol{\theta}_m^*)^2\right]}\frac{\sqrt{s_m}}{\psi(S_m)} + 4\frac{\lambda_n^2 s_m}{\psi^2(S_m)}$$

$$\leq \mathbb{E}_{\mathbf{x}\sim\mathcal{X}}\left[(\boldsymbol{\phi}_m(\mathbf{x})^\top\boldsymbol{\theta}_m^n - \boldsymbol{\phi}_m(\mathbf{x})^\top\boldsymbol{\theta}_m^*)^2\right] + 4\frac{\lambda_n^2 s_m}{\psi^2(S_m)} + 4\frac{\lambda_n^2 s_m}{\psi^2(S_m)}\,, \tag{37}$$

where the last inequality follows by using $4uv \leq u^2 + 4v^2$ with

$$u = \sqrt{\mathbb{E}_{\mathbf{x}\sim\mathcal{X}}\left[(\boldsymbol{\phi}_m(\mathbf{x})^\top\boldsymbol{\theta}_m^n - \boldsymbol{\phi}_m(\mathbf{x})^\top\boldsymbol{\theta}_m^*)^2\right]} \text{ and}$$

$$v = \lambda_n\sqrt{s_m}/\psi(S_m)\,.$$

Finally, by Lemma 8, we have from (37)

$$\mu(m,\boldsymbol{\theta}_m^n) - \mu(m,\boldsymbol{\theta}_m^*) + \lambda_n\|\boldsymbol{\theta}_m^n - \boldsymbol{\theta}_m^*\|_1 \leq 8\frac{\lambda_n^2 s_m}{\psi^2(S_m)}\,.$$

$\square$

# C   SAMPLE COMPLEXITY LOWER BOUND OF ALGORITHM 1 FOR CLASSIFICATION TASK

For the classification model selection scenario, the candidates are pruned networks that classify an input $\mathbf{x}$ to one of the predetermined $K$ classes. For a given data point $(\mathbf{x},\mathbf{y})$, the label $\mathbf{y} = (y_1,\cdots,y_K)^\top \in \{0,1\}^K$ is a one-hot vector that indicates the class to which the input $\mathbf{x}$ belongs. It is noteworthy that each model requires only $K-1$ parameter vectors for a $K$-class classification task, as the probability of $\mathbf{x}$ belonging to the $K$-th class is automatically determined by the probabilities of the first $K-1$ classes.

For each model, we denote the parameters $\mathbf{w}_m$ as a matrix consisting of $K-1$ columns $\boldsymbol{\Theta}_m := [\boldsymbol{\theta}_{m,1},\cdots,\boldsymbol{\theta}_{m,K-1}] \in \mathbb{R}^{d_m \times (K-1)}$. The loss function is the *negative log-likelihood*,

$$\ell(\sigma(\boldsymbol{\phi}_m(\mathbf{x})^\top\boldsymbol{\Theta}_m),\mathbf{y}) = -\sum_{k=1}^K y_k\log[\sigma(\boldsymbol{\phi}_m(\mathbf{x})^\top\boldsymbol{\Theta}_m)]_k\,,$$

where the activation function $\sigma(\cdot): \mathbb{R}^{K-1} \to \mathbb{R}^K$ gives

$$\left[\sigma(\boldsymbol{\phi}_m(\mathbf{x})^\top\boldsymbol{\Theta}_m)\right]_k := \frac{\exp(\boldsymbol{\phi}_m(\mathbf{x})^\top\boldsymbol{\theta}_{m,k})\mathbb{1}\{k<K\} + \mathbb{1}\{k=K\}}{1 + \sum_{k=1}^{K-1}\exp(\boldsymbol{\phi}_m(\mathbf{x})^\top\boldsymbol{\theta}_{m,k})}\,.$$

Note that $\sum_{k=1}^K[\sigma(\boldsymbol{\phi}_m(\mathbf{x})^\top\boldsymbol{\Theta}_m)]_k = 1$. The $L_1$-norm of $\boldsymbol{\Theta}_m$ is defined as $L_1(\boldsymbol{\Theta}_m) = \|\boldsymbol{\Theta}_m\|_{1,1} = \sum_{k=1}^{K-1}\|\boldsymbol{\theta}_{m,k}\|_1$. Now we introduce regularity assumptions for the classification scenario.

**Assumption 4** (Boundedness). *There exist constants $\phi_{\max}, R > 0$ such that, for all $m \in [M]$, it holds that $\|\boldsymbol{\phi}_m(\mathbf{x})\|_\infty \leq \phi_{\max}$ and $\|\boldsymbol{\Theta}_m^*\|_{1,1} \leq R$.*

**Assumption 5** (Compatibility). *For each $m \in [M]$, define the active set $S_m := \{(j,k) : \Theta_m^*(j,k) \neq 0\}$ and the sparsity index $s_m := |S_m| \ll d_m \times (K-1)$. For any $\Theta := [\theta_1, \ldots, \theta_{K-1}]^\top \in \mathbb{C}(S_m) := \{\Theta \in \mathbb{R}^{d_m \times (K-1)} : \|\Theta_{S_m^c}\|_{1,1} \leq 3\|\Theta_{S_m}\|_{1,1}\}$, we assume that there exists a constant $\psi^2(S_m) > 0$ such that*

$$\psi^2(S_m) \leq s_m \frac{\sum_{k=1}^{K-1} \sum_{\tilde{k}=1}^{K-1} \theta_k^\top \mathbb{E}_{\mathbf{x} \sim \mathcal{X}}[\phi_m(\mathbf{x})\phi_m(\mathbf{x})^\top]\theta_{\tilde{k}}}{\|\Theta_{S_m}\|_{1,1}^2}.$$

**Assumption 6** (Non-singularity). *For any $m \in [M]$, $\mathbf{x} \in \mathcal{X}$, and $\Theta \in \mathbb{R}^{d_m \times (K-1)}$, we define $\mathbf{A} \in \mathbb{R}^{(K-1) \times (K-1)}$ as $\mathbf{A}(i,j) := [\sigma(\phi_m(\mathbf{x})^\top \Theta)]_i (\delta_{i,j} - [\sigma(\phi_m(\mathbf{x})^\top \Theta)]_j)$, where $\delta_{i,j}$ is the Kronecker delta. We assume there exists a positive constant $0 < \kappa_0 < \infty$ such that $\frac{1}{2}\lambda_{\min}(\mathbf{A}) \geq \frac{1}{\kappa_0}$, where $\lambda_{\min}(\cdot)$ is the minimum eigenvalue of given symmetric matrix.*

**Discussion of Assumptions.** Assumption 4 and 5 extend Assumption 1 and 3 from the regression setting to the classification setting, respectively. Assumption 6 is standard in the multinomial logit model literature (Böhning, 1992; Oh & Iyengar, 2019; Amani & Thrampoulidis, 2021; Perivier & Goyal, 2022) which is also equivalent to the standard assumption for the link function in generalized linear models (Filippi et al., 2010; Li et al., 2017) to ensure non-singularity of the Fisher information matrix.

Now, we present the sample complexity lower bound of Algorithm 1 for classification model selection scenarios.

**Theorem 2** (Sample Complexity of Algorithm 1: Classification). *Suppose Assumption 4, 5, and 6 hold. Let $\bar{m} = \operatorname{argmax}_{m \in [M]} s_m/\psi^2(S_m)$. For any $\delta \in (0,1)$, $\epsilon > 0$, and $h \in [H]$, by setting the algorithmic parameters in Algorithm 1 as*

$$N \geq \frac{2}{\epsilon^2}\left(AC\epsilon + 2AD + 2B + 4\sqrt{ABD}\right), \lambda_{n_h} = \sqrt{A/n_h}, \alpha_{n_h} = \sqrt{B/n_h} + AC/4n_h,$$

*where*

$$A = 32\phi_{\max}^2 \log(4MH/\delta), \quad B = \frac{1}{2}(2\phi_{\max}R + \log K)^2$$

$$C = 16\kappa_0(K-1)s_{\bar{m}}/\psi^2(S_{\bar{m}}), \quad D = R^2,$$

*then, we have $\mathbb{P}\left(|\mu(m^N, \Theta_{m^N}^N) - \mu(m^*, \Theta_{m^*}^*)| \leq \epsilon\right) \geq 1 - \delta$.*

**Discussion of Theorem 2.** Compared to Theorem 1, Theorem 2 also establishes a sample complexity of $\Omega\left(\frac{\log M}{\epsilon^2}\right)$. The absence of the $\log d$ term stems from the difference in the loss functions used. This result demonstrates that our algorithm remains sample-efficient for selecting a classifier from a large pool of candidates, even when the dimensionality of the feature representation is high. Notably, in the special case where $M = 1$, $H = 1$, $K = 2$, and the sparsity level of $\Theta_m^*$ matches the ambient dimension (i.e., $s_m = d_m$), our sample complexity result coincides with the known sample complexity bounds for logistic regression (Hsu & Mazumdar, 2024). To the best of our knowledge, for multi-class classification problems with $K > 2$ under sparsity assumptions, this is the first result to establish such a sample complexity guarantee.

## C.1 PRELIMINARY

The overall flow of the proof for Theorem 2 is similar to that of Theorem 1. The primary distinctions arise from alterations the loss function $\ell(\cdot)$ and the $L_1$-norm. For the sake of brevity, only the points significantly different from the proof of Theorem 1 will be explained.

Let $\epsilon > 0$ be the error criterion, $n_1 < n_2 < \cdots < n_H = N$ be the epoch lengths, and $\mathcal{M}_h$ be the index set of models remaining in the candidate pool at epoch $h \in [H]$. For notational simplicity, we denote $n := n_h$ for a fixed epoch $h \in [H]$ when it is clear from the context. We denote the empirical process as $\nu_n(m, \Theta_m) := \hat{\mu}_n(m, \Theta_m) - \mu(m, \Theta_m)$ and define model $\bar{m}$ as $\bar{m} := \operatorname{argmax}_{m \in [M]} s_m/\psi^2(S_m)$. We also introduce the events employed in the subsequent proofs.

- $\mathscr{E} := \left\{ \mu(m^N, \boldsymbol{\Theta}_{m^N}^N) - \mu(m^*, \boldsymbol{\Theta}_{m^*}^*) > \epsilon \right\}$ : the event that inter-model excess risk of the model selected by Algorithm 1 exceeds the error criterion $\epsilon$.

- $\mathscr{S}_{n,m} := \{ |\nu_n(m, \boldsymbol{\Theta}_m^*)| \leq \eta_n \}$: the event that empirical process of the $m$-th model having its optimal parameter is bounded.

- $\mathscr{S}_n := \bigcap_{m \in \mathcal{M}_h} \mathscr{S}_{n,m}$

- $\mathscr{T}_{n,m} := \left\{ |\nu_n(m, \boldsymbol{\Theta}_m^n) - \nu_n(m, \boldsymbol{\Theta}_m^*)| \leq \tilde{\lambda}_n \|\boldsymbol{\Theta}_m^n - \boldsymbol{\Theta}_m^*\|_{1,1} \right\}$ : event that model $m$'s empirical process concentration is bounded by its $L_1$-regularization parameter concentration.

- $\mathscr{T}_n := \bigcap_{m \in \mathcal{M}_h} \mathscr{T}_{n,m}$

- $\mathscr{R}_h := \left\{ \bigcap_{h'=1}^{h} (\mathscr{S}_{n_{h'}} \cap \mathscr{T}_{n_{h'}}) \right\}$: the event that $\mathscr{S}_{n_{h'}}$ and $\mathscr{T}_{n_{h'}}$ both hold for all of the epochs $h' \in [h]$.

The proof of Theorem 2 involves the following three key lemmas.

**Lemma 11.** *For all $h \in [H]$, if the regularization parameters $\{\lambda_{n_h}\}_{h=1}^H$ and confidence radii $\{\alpha_{n_h}\}_{h=1}^H$ of Algorithm 1 are set as $\lambda_{n_h} = \sqrt{A/n_h}$ and $\alpha_{n_h} = \sqrt{B/n_h} + AC/4n_h$, where $A$, $B$ and $C$ are the values specified in Theorem 2, then event $\mathscr{R}_H$ occurs with probability at least $1 - \delta$.*

**Lemma 12.** *Under $\mathscr{R}_{H-1}$ and by setting $\alpha_{n_h} = \eta_{n_h} + 4\frac{\lambda_{n_h}^2 s_{\bar{m}}}{\psi^2(S_{\bar{m}})}\kappa_0(K-1)$ for epochs $h \in [H-1]$, the best model $m^*$ survives for the selection step, i.e., $m^* \in \mathcal{M}_H$.*

**Lemma 13.** *If $m^* \in \mathcal{M}_H$, under $\mathscr{S}_N \cap \mathscr{T}_N$, the inter-model excess risk of the selected model $m^N$ is upper bounded as follows:*

$$\underbrace{\mu(m^N, \boldsymbol{\Theta}_{m^N}^N) - \mu(m^*, \boldsymbol{\Theta}_{m^*}^*)}_{\text{inter-model excess risk}} \leq 2\eta_N + 2\lambda_N R + 16\frac{\lambda_N^2 s_{\bar{m}}}{\psi^2(S_{\bar{m}})}\kappa_0(K-1),$$

Similar to Section B, in Sections C.2, C.3 and C.4, we prove Lemmas 11, 12 and 13 respectively. In Section C.5, we prove Theorem 2. In Section C.6, we provide the proofs for the technical lemmas used in this chapter.

## C.2 PROOF OF LEMMA 11

*Proof of Lemma 11.* The proof of Lemma 11 utilizes the following lemmas.

**Lemma 14.** *Let $M_h$ be the number of remaining models for epoch $h$. For a given $\delta_1 \in (0,1)$, if*

$$\eta_n = \sqrt{\frac{(2\phi_{\max}R + \log K)^2 \log(2M_h/\delta_1)}{2n}},$$

*event $\mathscr{S}_n$ occurs with probability at least $1 - \delta_1$.*

**Lemma 15.** *Let $M_h$ be the number of remaining models for epoch $h$. For a given $\delta_2 \in (0,1)$, if*

$$\tilde{\lambda}_n = \sqrt{\frac{8\phi_{\max}^2 \log(2M_h/\delta_2)}{n}},$$

*event $\mathscr{T}_n$ occurs with probability at least $1 - \delta_2$.*

Note that setting the regularization parameters $\{\lambda_{n_h}\}_{h=1}^H$ and confidence radii $\{\alpha_{n_h}\}_{h=1}^H$ of Algorithm 1 as $\lambda_{n_h} = \sqrt{A/n_h}$ and $\alpha_{n_h} = \sqrt{B/n_h} + AC/4n_h$ implies setting $\{\eta_{n_h}\}_{h=1}^H$ as $\eta_{n_h} = \sqrt{B/n_h}$.

For all $h \in [H]$, by setting $\{\lambda_{n_h}\}_{h=1}^H$ and $\{\alpha_{n_h}\}_{h=1}^H$ as specified in Lemma 11, and by choosing $\delta_1$, $\delta_2$, and $\tilde{\lambda}_{n_h}$ as $\delta_1 = \delta_2 = \frac{\delta}{2H}$ and $\lambda_{n_h} = 2\tilde{\lambda}_{n_h} = \sqrt{A/n_h}$, we have,

$$\eta_{n_h} = \sqrt{B/n_h}$$

$$= \sqrt{\frac{(2\phi_{\max}R + \log K)^2 \log(4MH/\delta)}{2n_h}} \geq \sqrt{\frac{(2\phi_{\max}R + \log K)^2 \log(2M_h/\delta_1)}{2n_h}}$$

and

$$\tilde{\lambda}_{n_h} = \frac{1}{2}\sqrt{A/n_h} = \sqrt{\frac{8\phi_{\max}^2 \log(4MH/\delta)}{n_h}} \geq \sqrt{\frac{8\phi_{\max}^2 \log(2M_h/\delta_2)}{n_h}},$$

where for the last inequality of each, we use $M \geq M_h$ for all $h \in [H]$. Thus, under the conditions established in Lemma 11, Lemmas 14 and 15 hold.

For the remaining, we conclude the proof using the same argument in Lemma 1, along with the results of Lemma 14 and 15. $\qquad\square$

### C.3   PROOF OF LEMMA 12

*Proof of Lemma 12.* As with Lemma 2, it is sufficient to prove the following lemma.

**Lemma 16.** *For any $h \in [H-1]$, let $n := n_h$. Suppose events $\mathscr{S}_n$ and $\mathscr{T}_n$ both occur. Then, for all $m \in \mathcal{M}_h$, by setting $\lambda_n = 2\tilde{\lambda}_n$ and $\alpha_n = \eta_n + 4\frac{\lambda_n^2 s_{\bar{m}}}{\psi^2(S_{\bar{m}})}\kappa_0(K-1)$, the following holds:*

$$\mu(m, \boldsymbol{\Theta}_m^*) \in [\hat{\mu}_n(m, \boldsymbol{\Theta}_m^n) - \alpha_n, \ \hat{\mu}_n(m, \boldsymbol{\Theta}_m^n) + \alpha_n].$$

Then we conclude the proof using the same argument in Lemma 2, along with the result of Lemma 16. $\qquad\square$

### C.4   PROOF OF LEMMA 13

*Proof of Lemma 13.* With the same argument in **Step 1** in Section B.4, we have

$$\underbrace{\mu(m^N, \boldsymbol{\Theta}_{m^N}^N) - \mu(m^*, \boldsymbol{\Theta}_{m^*}^*)}_{\text{inter-model excess risk}} \leq [\mu(m^N, \boldsymbol{\Theta}_{m^N}^N) - \mu(m^N, \boldsymbol{\Theta}_{m^N}^*)] + 2\tilde{\Delta}, \tag{38}$$

where $\tilde{\Delta} := \max_{m \in \mathcal{M}_H} |\tilde{\mu}_N(m, \boldsymbol{\Theta}_m^N) - \mu(m, \boldsymbol{\Theta}_m^*)|$.

We adapt **Step 2** to accommodate classification scenarios.

STEP 2: BOUNDING $\left[\mu(m^N, \boldsymbol{\Theta}_{m^N}^N) - \mu(m^N, \boldsymbol{\Theta}_{m^N}^*)\right] + 2\tilde{\Delta}$

We start from bounding $\tilde{\Delta}$.

$$\begin{aligned}
\tilde{\Delta} &= \max_{m \in \mathcal{M}_H} \left|\tilde{\mu}_N(m, \boldsymbol{\Theta}_m^N) - \mu(m, \boldsymbol{\Theta}_m^*)\right| \\
&= \max_{m \in \mathcal{M}_H} \left|\hat{\mu}_N(m, \boldsymbol{\Theta}_m^N) + \lambda_N \|\boldsymbol{\Theta}_m^N\|_1 - \mu(m, \boldsymbol{\Theta}_m^*)\right| \\
&= \max_{m \in \mathcal{M}_H} \left|\hat{\mu}_N(m, \boldsymbol{\Theta}_m^N) - \mu(m, \boldsymbol{\Theta}_m^N) + \lambda_N \|\boldsymbol{\Theta}_m^N - \boldsymbol{\Theta}_m^* + \boldsymbol{\Theta}_m^*\|_1 + \mu(m, \boldsymbol{\Theta}_m^N) - \mu(m, \boldsymbol{\Theta}_m^*)\right| \\
&\leq \underbrace{\max_{m \in \mathcal{M}_H} |\hat{\mu}_N(m, \boldsymbol{\Theta}_m^N) - \mu(m, \boldsymbol{\Theta}_m^N)|}_{(*)} \\
&\quad + \underbrace{\max_{m \in \mathcal{M}_H} \lambda_N \|\boldsymbol{\Theta}_m^*\|_1}_{\tilde{\Delta}_2} + \underbrace{\max_{m \in \mathcal{M}_H} \left[\mu(m, \boldsymbol{\Theta}_m^N) - \mu(m, \boldsymbol{\Theta}_m^*) + \lambda_N \|\boldsymbol{\Theta}_m^N - \boldsymbol{\Theta}_m^*\|_1\right]}_{\tilde{\Delta}_3}.
\end{aligned}$$

For $(*)$ under the event $\mathscr{T}_N$,

$$(*) = \max_{m \in \mathcal{M}_H} |\underbrace{\hat{\mu}_N(m, \boldsymbol{\Theta}_m^N) - \mu(m, \boldsymbol{\Theta}_m^N)}_{\nu_N(m, \boldsymbol{\Theta}_m^N)}|$$

$$= \max_{m \in \mathcal{M}_H} \left| \nu_N(m, \boldsymbol{\Theta}_m^N) - \nu_N(m, \boldsymbol{\Theta}_m^*) + \nu_N(m, \boldsymbol{\Theta}_m^*) \right|$$

$$\leq \max_{m \in \mathcal{M}_H} |\nu_N(m, \boldsymbol{\Theta}_m^N) - \nu_N(m, \boldsymbol{\Theta}_m^*)| + \underbrace{\max_{m \in \mathcal{M}_H} |\nu_N(m, \boldsymbol{\Theta}_m^*)|}_{\tilde{\Delta}_1}$$

$$\leq \max_{m \in \mathcal{M}_H} \tilde{\lambda}_N \|\boldsymbol{\Theta}_m^N - \boldsymbol{\Theta}_m^*\|_1 + \tilde{\Delta}_1$$

$$\leq \max_{m \in \mathcal{M}_H} \frac{\lambda_N}{2} \|\boldsymbol{\Theta}_m^N - \boldsymbol{\Theta}_m^*\|_1 + \tilde{\Delta}_1 \,,$$

where the last inequality comes by the choice of $\tilde{\lambda}_N = \frac{\lambda_N}{2}$. Thus,

$$\left[ \mu(m^N, \boldsymbol{\Theta}_{m^N}^N) - \mu(m^N, \boldsymbol{\Theta}_{m^N}^*) \right] + 2\tilde{\Delta}$$

$$\leq \left[ \mu(m^N, \boldsymbol{\Theta}_{m^N}^N) - \mu(m^N, \boldsymbol{\Theta}_{m^N}^*) \right] + \max_{m \in \mathcal{M}_H} \lambda_N \|\boldsymbol{\Theta}_m^N - \boldsymbol{\Theta}_m^*\|_1 + 2\tilde{\Delta}_1 + 2\tilde{\Delta}_2 + 2\tilde{\Delta}_3$$

$$\leq 2\tilde{\Delta}_1 + 2\tilde{\Delta}_2 + 4\tilde{\Delta}_3 \,, \tag{39}$$

where the last inequality holds since $\left[ \mu(m, \boldsymbol{\Theta}_m^N) - \mu(m, \boldsymbol{\Theta}_m^*) \right] \leq \tilde{\Delta}_3$ and $\lambda_N \|\boldsymbol{\Theta}_m^N - \boldsymbol{\Theta}_m^*\|_1 \leq \tilde{\Delta}_3$ for all $m \in \mathcal{M}_H$.

Under the event $\mathscr{S}_N$, we have $\tilde{\Delta}_1 \leq \eta_N$. For $\tilde{\Delta}_2$, we have $\tilde{\Delta}_2 \leq \lambda_N R$ by Assumption 4. For $\tilde{\Delta}_3$, we invoke the following lemma:

**Lemma 17** (Oracle inequality for classification)**.** *For any $h \in [H]$, let us denote $n_h = n$. Suppose that the event $\mathscr{T}_n$ takes place. Then, for any $m \in \mathcal{M}_h$,*

$$\mu(m, \boldsymbol{\Theta}_m^n) - \mu(m, \boldsymbol{\Theta}_m^*) + \lambda_n \|\boldsymbol{\Theta}_m^n - \boldsymbol{\Theta}_m^*\|_{1,1} \leq 4 \frac{\lambda_n^2 s_m}{\psi^2(S_m)} \kappa_0(K-1) \,.$$

Combining (39) and the results of Lemma 17, we prove Lemma 13:

$$\underbrace{\mu(m^N, \boldsymbol{\Theta}_{m^N}^N) - \mu(m^*, \boldsymbol{\Theta}_{m^*}^*)}_{\text{inter-model excess risk}} \leq 2\tilde{\Delta}_1 + 2\tilde{\Delta}_2 + 4\tilde{\Delta}_3$$

$$\leq 2\eta_N + 2\lambda_N R + 16 \frac{\lambda_N^2 s_{\bar{m}}}{\psi^2(S_{\bar{m}})} \kappa_0(K-1) \,. \tag{40}$$

$\square$

## C.5  MAIN PROOF OF THEOREM 2

*Proof of Theorem 2.*  According to Lemma 11, the constraints on $\{\lambda_{n_h}\}_{h=1}^H$ and $\{\alpha_{n_h}\}_{h=1}^H$ specified in Theorem 2 ensure that the event $\mathscr{R}_H$ occurs with probability at least $1 - \delta$. The occurrence of event $\mathscr{R}_H$ implies the occurrence of event $\mathscr{R}_{H-1}$ and event $\mathscr{S}_N \cap \mathscr{T}_N$.

If $\mathscr{R}_{H-1}$ takes place, Lemma 12 holds as the confidence radii $\{\alpha_{n_h}\}_{h=1}^{H-1}$ are set to meet the conditions of Lemma 12. Hence, with high probability, the candidate pool retains the best model $m^*$ for the final epoch $H$, i.e., $m^* \in \mathcal{M}_H$.

Additionally, when $m^* \in \mathcal{M}_H$ and if $\mathscr{S}_N \cap \mathscr{T}_N$ occurs, Lemma 13 applies. Therefore, with high probability, $\mu(m^N, \boldsymbol{\Theta}_{m^N}^N) - \mu(m^*, \boldsymbol{\Theta}_{m^*}^*) \leq \mathcal{O}(\frac{1}{\sqrt{N}})$.

Recall that $\mathscr{E} := \left\{ \mu(m^N, \mathbf{\Theta}_{m^N}^N) - \mu(m^*, \mathbf{\Theta}_{m^*}^*) > \epsilon \right\}$. Starting from the law of total probability, we have

$$
\begin{aligned}
\mathbb{P}(\mathscr{E}) &= \mathbb{P}(\mathscr{E} \mid \mathscr{R}_H)\mathbb{P}(\mathscr{R}_H) + \mathbb{P}(\mathscr{E} \mid \mathscr{R}_H^{\mathsf{c}})\mathbb{P}(\mathscr{R}_H^{\mathsf{c}}) \\
&\leq \mathbb{P}(\mathscr{E} \mid \mathscr{R}_H) + \mathbb{P}(\mathscr{R}_H^{\mathsf{c}}) \\
&\leq \mathbb{P}(\mathscr{E} \mid \mathscr{R}_H) + \delta \\
&= \mathbb{P}(\mathscr{E} \mid \mathscr{R}_{H-1}, \mathscr{S}_N, \mathscr{T}_N) + \delta \,,
\end{aligned}
\tag{41}
$$

where the second inequality results from Lemma 11. By Lemma 12, the occurrence of event $\mathscr{R}_{H-1}$ and the constraint on $\{\alpha_{n_h}\}_{h=1}^{H-1}$ guarantee that the best model survives for the selection step, $m^* \in \mathcal{M}_H$. For the selection step, we have

$$
\begin{aligned}
\mathbb{P}(\mathscr{E}) &\leq \mathbb{P}(\mathscr{E} \mid \mathscr{R}_{H-1}, \mathscr{S}_N, \mathscr{T}_N) + \delta \\
&\leq \mathbb{P}(\underbrace{2\eta_N + 2\lambda_N R + 16\frac{\lambda_N^2 s_{\bar{m}}}{\psi^2(S_{\bar{m}})}\kappa_0(K-1) > \epsilon}_{(*)} \mid \mathscr{R}_{H-1}, \mathscr{S}_N, \mathscr{T}_N) + \delta \\
&\leq \delta
\end{aligned}
\tag{42}
$$

where the first inequality employs Lemma 3. Regarding to the last inequality, as $\eta_N$ and $\lambda_N$ both have $1/\sqrt{N}$ terms, by setting $N$ large enough, we can make the LHS of $(*)$ always be smaller than or equal to $\epsilon$, i.e., $\mathbb{P}((*) \mid \mathscr{R}_{H-1}, \mathscr{S}_N, \mathscr{T}_N) = 0$.

To summarize, the sample complexity is any number that exceeds $N$ satisfying the following statement:

$$
2\eta_N + 2\lambda_N R + 16\frac{\lambda_N^2 s_{\bar{m}}}{\psi^2(S_{\bar{m}})}\kappa_0(K-1) \leq \epsilon \,,
\tag{43}
$$

From the proof of Lemma 11, we know that setting $\lambda_{n_h} = \sqrt{A/n_h}$ and $\alpha_{n_h} = \sqrt{B/n_h} + AC/4n_h$ implies setting $\eta_{n_h} = \sqrt{B/n_h}$ for all $h \in [H]$. Using $\lambda_N = \sqrt{A/N}$ and $\eta_N = \sqrt{B/N}$, we get

$$
2\sqrt{\frac{B}{N}} + 2\sqrt{\frac{AD}{N}} + \frac{AC}{N} = 2\eta_N + 2\lambda_N R + 16\frac{\lambda_N^2 s_{\bar{m}}}{\psi^2(S_{\bar{m}})}\kappa_0(K-1)
\tag{44}
$$

where $A$, $B$, $C$, and $D$ are

$$
\begin{aligned}
A &= 32\phi_{\max}^2 \log(4MH/\delta)\,, \quad B = \frac{1}{2}(2\phi_{\max}R + \log K)^2 \log(4MH/\delta) \\
C &= 16\frac{s_{\bar{m}}}{\psi^2(S_{\bar{m}})}\kappa_0(K-1)\,, \quad D = R^2\,.
\end{aligned}
$$

With the same argument used to derive (16), we get

$$
N \geq \frac{2}{\epsilon^2}\left( AC\epsilon + 2AD + 2B + 4\sqrt{ABD} \right)\,.
$$

For such $N$, we have the desired result $\mathbb{P}(\mathscr{E}) \leq \delta$.

$\square$

## C.6 PROOFS FOR TECHNICAL LEMMAS

### C.6.1 PROOF OF LEMMA 14

*Proof of Lemma 14.* The main difference from Lemma 4 lies in the boundedness of the negative log-likelihood loss when applying the Höeffding's inequality. We define the log-sum-exp function

as $\mathrm{lse}\,[\mathbf{v}] := \log\left[1 + \sum_{k=1}^{K} \exp(v_k)\right]$ for $\mathbf{v}^{\top} \in \mathbb{R}^K$. Then, for each $i \in [n]$,

$$
\ell(\boldsymbol{\phi}_m(\mathbf{x}_i)^{\top}\boldsymbol{\Theta}_m^*, \mathbf{y}_i)
$$

$$
= -\sum_{k=1}^{K-1} y_{k,i}\boldsymbol{\phi}_m(\mathbf{x}_i)^{\top}\boldsymbol{\theta}_{m,k}^* + \mathrm{lse}\left[\boldsymbol{\phi}_m(\mathbf{x}_i)^{\top}\boldsymbol{\Theta}_m^*\right]
$$

$$
\leq -\sum_{k=1}^{K-1} y_{k,i}\boldsymbol{\phi}_m(\mathbf{x}_i)^{\top}\boldsymbol{\theta}_{m,k}^* + \max\left\{0, \boldsymbol{\phi}_m(\mathbf{x}_i)^{\top}\boldsymbol{\theta}_{m,1}^*, \cdots, \boldsymbol{\phi}_m(\mathbf{x}_i)^{\top}\boldsymbol{\theta}_{m,K-1}^*\right\} + \log K
$$

$$
\leq \sum_{k=1}^{K-1} y_{k,i}|\boldsymbol{\phi}_m(\mathbf{x}_i)^{\top}\boldsymbol{\theta}_{m,k}^*| + \max\left\{0, \boldsymbol{\phi}_m(\mathbf{x}_i)^{\top}\boldsymbol{\theta}_{m,1}^*, \cdots, \boldsymbol{\phi}_m(\mathbf{x}_i)^{\top}\boldsymbol{\theta}_{m,K-1}^*\right\} + \log K
$$

$$
\leq \sum_{k=1}^{K-1} y_{k,i}(\phi_{\max}R) + \phi_{\max}R + \log K
$$

$$
\leq 2\phi_{\max}R + \log K \,.
$$

The first inequality holds by the upper bound of the log-sum-exp function. The third inequality holds by Hölder's inequality. Note that $\|\boldsymbol{\theta}_{m,k}^*\|_1 \leq \|\boldsymbol{\Theta}_m^*\|_{1,1} \leq R$. The last inequality comes from $\sum_{k=1}^{K-1} y_{k,i} \leq \sum_{k=1}^{K} y_{k,i} = 1$. This implies that, for all $i \in [n]$, $0 \leq \ell(\boldsymbol{\phi}_m(\mathbf{x}_i)^{\top}\boldsymbol{\Theta}_m, \mathbf{y}_i) \leq 2\phi_{\max}R + \log K$.

Applying Höeffding's inequality, we get

$$
\mathbb{P}\left(|\hat{\mu}(m,\boldsymbol{\theta}_m^*) - \mu(m,\boldsymbol{\theta}_m^*)| > \eta_n\right)
$$

$$
= \mathbb{P}\left(\left|\frac{1}{n}\sum_{i=1}^{n}\left[\ell(\boldsymbol{\phi}_m(\mathbf{x}_i)^{\top}\boldsymbol{\theta}_m, y_i)\right] - \mathbb{E}\left[\ell(\boldsymbol{\phi}_m(\mathbf{x}_i)^{\top}\boldsymbol{\theta}_m, y_i)\right]\right| > \eta_n\right)
$$

$$
\leq 2\exp\left(-\frac{2n^2\eta_n^2}{\sum_{i=1}^{n}(2\phi_{\max}R + \log K)^2}\right)
$$

$$
= 2\exp\left(-\frac{2n\eta_n^2}{(2\phi_{\max}R + \log K)^2}\right)
$$

$$
= \delta_0
$$

for some $\delta_0 \in (0,1)$.

Rearranging the last inequality with respect to $\eta_n$, we get

$$
\eta_n = \sqrt{\frac{(2\phi_{\max}R + \log K)^2 \log(2/\delta_0)}{2n}} \,. \tag{45}
$$

Then with the same argument in Lemma 4, we conclude the proof. $\qquad\square$

### C.6.2 PROOF OF LEMMA 15

*Proof of Lemam 15.* Similar to Lemma 5, it is enough to check the bound of $\chi_i$ which is denoted as follows:

$$
\nu_n(m,\boldsymbol{\Theta}_m^n) - \nu_n(m,\boldsymbol{\Theta}_m^*)
$$

$$
= [\hat{\mu}_n(m,\boldsymbol{\Theta}_m^n) - \mu(m,\boldsymbol{\Theta}_m^n)] - [\hat{\mu}_n(m,\boldsymbol{\Theta}_m^*) - \mu(m,\boldsymbol{\Theta}_m^*)]
$$

$$
= [\hat{\mu}_n(m,\boldsymbol{\Theta}_m^n) - \hat{\mu}_n(m,\boldsymbol{\Theta}_m^*)] - [\mu(m,\boldsymbol{\Theta}_m^n) - \mu(m,\boldsymbol{\Theta}_m^*)]
$$

$$
= \frac{1}{n}\sum_{i=1}^{n}[\chi_i] - \mathbb{E}[\chi_i] \,.
$$

where

$$
\chi_i := -\sum_{k=1}^{K} y_{k,i}\boldsymbol{\phi}_m(\mathbf{x}_i)^{\top}\left(\boldsymbol{\theta}_{m,k}^n - \boldsymbol{\theta}_{m,k}^*\right) + \mathrm{lse}\left[\boldsymbol{\phi}_m(\mathbf{x}_i)^{\top}\boldsymbol{\Theta}_m^n\right] - \mathrm{lse}\left[\boldsymbol{\phi}_m(\mathbf{x}_i)^{\top}\boldsymbol{\Theta}_m^*\right] \,.
$$

To apply Höeffding's inequality, we need the upper and lower bounds of $\chi_i$. Let us denote

$$\chi_{i,1} := -\sum_{k=1}^{K-1} y_{k,i} \phi_m(\mathbf{x}_i)^\top \left( \boldsymbol{\theta}_{m,k}^n - \boldsymbol{\theta}_{m,k}^* \right) \quad \text{and}$$

$$\chi_{i,2} := \mathrm{lse} \left[ \phi_m(\mathbf{x}_i)^\top \boldsymbol{\Theta}_m^n \right] - \mathrm{lse} \left[ \phi_m(\mathbf{x}_i)^\top \boldsymbol{\Theta}_m^* \right] .$$

Then we have $|\chi_i| \le |\chi_{i,1}| + |\chi_{i,2}|$. Define $\mathbf{v}_i \in \mathbb{R}^{K-1}$ such that

$$\mathbf{v}_i := \left[ \phi_m(\mathbf{x}_i)^\top \left( \boldsymbol{\theta}_{m,1}^n - \boldsymbol{\theta}_{m,1}^* \right), \; \phi_m(\mathbf{x}_i)^\top \left( \boldsymbol{\theta}_{m,2}^n - \boldsymbol{\theta}_{m,2}^* \right), \right.$$

$$\left. \cdots, \phi_m(\mathbf{x}_i)^\top \left( \boldsymbol{\theta}_{m,K-1}^n - \boldsymbol{\theta}_{m,K-1}^* \right), \; 0 \right] .$$

Examining $|\chi_{i,1}|$, we get,

$$|\chi_{i,1}| \le \|\mathbf{y}_i\|_\infty \|\mathbf{v}_i\|_1$$

$$= 1 \cdot \sum_{k=1}^{K-1} \left| \phi_m(\mathbf{x}_i)^\top \left( \boldsymbol{\theta}_{m,k}^n - \boldsymbol{\theta}_{m,k}^* \right) \right|$$

$$\le \sum_{k=1}^{K-1} \|\phi_m(\mathbf{x}_i)\|_\infty \|\boldsymbol{\theta}_{m,k}^n - \boldsymbol{\theta}_{m,k}^*\|_1$$

$$\le \phi_{\max} \|\boldsymbol{\Theta}_m^n - \boldsymbol{\Theta}_m^*\|_{1,1}$$

where the first inequality comes from Hölder's inequality.

To bound $|\chi_{i,2}|$, as the log-sum-exp function is convex and continuous, we utilize the mean value theorem. The following holds for $\mathbf{u}_i = c\phi_m(\mathbf{x}_i)^\top \boldsymbol{\Theta}_m^* + (1-c)\phi_m(\mathbf{x}_i)^\top \boldsymbol{\Theta}_m^n$ with some $c \in (0,1)$:

$$|\chi_{i,2}| = \left| \mathrm{lse} \left[ \phi_m(\mathbf{x}_i)^\top \boldsymbol{\Theta}_m^n \right] - \mathrm{lse} \left[ \phi_m(\mathbf{x}_i)^\top \boldsymbol{\Theta}_m^* \right] \right|$$

$$= \left| \nabla \mathrm{lse} \left[ \mathbf{u}_i \right]^\top \left( \boldsymbol{\Theta}_m^n - \boldsymbol{\Theta}_m^* \right)^\top \phi_m(\mathbf{x}_i) \right|$$

$$\le \| \nabla \mathrm{lse} \left[ \mathbf{u}_i \right] \|_\infty \| \left( \boldsymbol{\Theta}_m^n - \boldsymbol{\Theta}_m^* \right)^\top \phi_m(\mathbf{x}_i) \|_1$$

$$\le 1 \cdot \sum_{k=1}^{K-1} \left| \phi_m(\mathbf{x}_i)^\top \left( \boldsymbol{\theta}_{m,k}^n - \boldsymbol{\theta}_{m,k}^* \right) \right|$$

$$\le \sum_{k=1}^{K-1} \|\phi_m(\mathbf{x}_i)\|_\infty \|\boldsymbol{\theta}_{m,k}^n - \boldsymbol{\theta}_{m,k}^*\|_1$$

$$\le \phi_{\max} \|\boldsymbol{\Theta}_m^n - \boldsymbol{\Theta}_m^*\|_{1,1} ,$$

where the second inequality holds as $\| \nabla \mathrm{lse} \left[ \mathbf{u} \right] \|_\infty = \max_{k \in [K-1]} \left\{ \frac{\exp(u_k)}{1 + \sum_{k'=1}^{K-1} \exp(u_{k'})} \right\} \le 1$.

By the above results, we get $|\chi_i| \le 2\phi_{\max} \|\boldsymbol{\Theta}_m^n - \boldsymbol{\Theta}_m^*\|_{1,1}$. Therefore, $-2\phi_{\max} \|\boldsymbol{\Theta}_m^n - \boldsymbol{\Theta}_m^*\|_{1,1} \le \chi_i \le 2\phi_{\max} \|\boldsymbol{\Theta}_m^n - \boldsymbol{\Theta}_m^*\|_{1,1}$.

Applying the Höeffding's inequality, we get

$$\mathbb{P} \left( |\nu_n(m, \boldsymbol{\Theta}_m^n) - \nu_n(m, \boldsymbol{\Theta}_m^*)| > \tilde{\lambda}_n \|\boldsymbol{\Theta}_m^n - \boldsymbol{\Theta}_m^*\|_{1,1} \right)$$

$$= \mathbb{P} \left( \left| \frac{1}{n} \sum_{i=1}^n \chi_i - \mathbb{E}\left[ \chi_i \right] \right| > \tilde{\lambda}_n \|\boldsymbol{\Theta}_m^n - \boldsymbol{\Theta}_m^*\|_{1,1} \right)$$

$$\le 2 \exp \left( -\frac{2n^2 \left( \tilde{\lambda}_n \|\boldsymbol{\Theta}_m^n - \boldsymbol{\Theta}_m^*\|_{1,1} \right)^2}{\sum_{i=1}^n \left( 4\phi_{\max} \|\boldsymbol{\Theta}_m^n - \boldsymbol{\Theta}_m^*\|_{1,1} \right)^2} \right)$$

$$= 2 \exp \left( -\frac{t\tilde{\lambda}_n^2}{8\phi_{\max}^2} \right)$$

$$= \delta_0 .$$

Rearranging the last inequality with respect to $\tilde{\lambda}_n$, we get

$$\tilde{\lambda}_n = \sqrt{\frac{8\phi_{\max}^2 \log(2/\delta_0)}{n}}\,. \tag{46}$$

Then, with the same argument in Lemma 5, we conclude the proof. □

### C.6.3 PROOF OF LEMMA 16

*Proof of Lemma 16.* Define $\hat{\Delta}_n := \max_{m \in \mathcal{M}_h} |\hat{\mu}_n(m, \mathbf{\Theta}_m^n) - \mu(m, \mathbf{\Theta}_m^*)|$. Since $\mu(m, \mathbf{\Theta}_m^*)$ belongs to the following interval for any $m \in \mathcal{M}_h$,

$$\mu(m, \mathbf{\Theta}_m^*) \in \Big[\hat{\mu}_n(m, \mathbf{\Theta}_m^n) - |\hat{\mu}_n(m, \mathbf{\Theta}_m^n) - \mu(m, \mathbf{\Theta}_m^*)|,$$

$$\hat{\mu}_n(m, \mathbf{\Theta}_m^n) + |\hat{\mu}_n(m, \mathbf{\Theta}_m^n) - \mu(m, \mathbf{\Theta}_m^*)|\Big],$$

we have,

$$\hat{\mu}(m, \mathbf{\Theta}_m^n) - \hat{\Delta}_n \leq \mu(m, \mathbf{\Theta}_m^*) \leq \hat{\mu}(m, \mathbf{\Theta}_m^n) + \hat{\Delta}_n\,.$$

Notice that

$$\hat{\Delta}_n = \max_{m \in \mathcal{M}_h} |\hat{\mu}(m, \mathbf{\Theta}_m^n) - \mu(m, \mathbf{\Theta}_m^*)|$$

$$= \max_{m \in \mathcal{M}_h} |\hat{\mu}(m, \mathbf{\Theta}_m^n) - \mu(m, \mathbf{\Theta}_m^n) + \mu(m, \mathbf{\Theta}_m^n) - \mu(m, \mathbf{\Theta}_m^*)|$$

$$\leq \max_{m \in \mathcal{M}_h} |\hat{\mu}(m, \mathbf{\Theta}_m^n) - \mu(m, \mathbf{\Theta}_m^n)| + \max_{m \in \mathcal{M}_h} |\mu(m, \mathbf{\Theta}_m^n) - \mu(m, \mathbf{\Theta}_m^*)|$$

$$\leq \underbrace{\max_{m \in \mathcal{M}_h} |\hat{\mu}(m, \mathbf{\Theta}_m^n) - \mu(m, \mathbf{\Theta}_m^n)|}_{\hat{\Delta}_{n,1}}$$

$$+ \underbrace{\max_{m \in \mathcal{M}_h} \mu(m, \mathbf{\Theta}_m^n) - \mu(m, \mathbf{\Theta}_m^*) + \lambda_n \|\mathbf{\Theta}_m^n - \mathbf{\Theta}_m^*\|_{1,1}}_{\hat{\Delta}_{n,2}}\,.$$

Note that under $\mathscr{S}_n$, we have $\hat{\Delta}_{n,1} \leq \eta_n$. Also, by Lemma 17, we have

$$\hat{\Delta}_{n,2} \leq \max_{m \in [M]} 4\frac{\lambda_n^2 s_m}{\psi^2(S_m)}\kappa_0(K-1) = 4\frac{\lambda_n^2 s_{\bar{m}}}{\psi^2(S_{\bar{m}})}\kappa_0(K-1)\,,$$

where we denote $\bar{m} = \operatorname{argmax}_{m \in [M]} s_m/\psi^2(S_m)$.

Combininig the upper bounds of $\hat{\Delta}_{n,1}$ and $\hat{\Delta}_{n,2}$, we have

$$\hat{\Delta}_n \leq \eta_n + 4\frac{\lambda_n^2 s_{\bar{m}}}{\psi^2(S_{\bar{m}})}\kappa_0(K-1)\,. \tag{47}$$

Therefore, under the event $\{\mathscr{S}_n \cap \mathscr{T}_n\}$, for all $m \in \mathcal{M}_h$, the following holds:

$$\hat{\mu}(m, \mathbf{\Theta}_m^n) - \alpha_n \leq \mu(m, \mathbf{\Theta}_m^*) \leq \hat{\mu}(m, \mathbf{\Theta}_m^n) + \alpha_n\,,$$

where $\alpha_n = \eta_n + 4\frac{\lambda_n^2 s_{\bar{m}}}{\psi^2(S_{\bar{m}})}\kappa_0(K-1)$. □

### C.6.4 PROOF OF LEMMA 17

*Proof of Lemma 17.* For each $m \in \mathcal{M}_h$, since we have

$$\hat{\mu}(m, \mathbf{\Theta}_m^n) + \lambda_n \|\mathbf{\Theta}_m^n\|_{1,1} \leq \hat{\mu}(m, \mathbf{\Theta}_m^*) + \lambda_n \|\mathbf{\Theta}_m^*\|_{1,1}\,.$$

By adding and subtracting $\mu(m, \boldsymbol{\Theta}_m^n)$ and $\mu(m, \boldsymbol{\Theta}_m^*)$ to both sides respectively, we have

$$\underbrace{\hat{\mu}(m, \boldsymbol{\Theta}_m^n) - \mu(m, \boldsymbol{\Theta}_m^n)}_{\nu_n(m, \boldsymbol{\Theta}_m^n)} + \mu(m, \boldsymbol{\Theta}_m^n) + \lambda_n \|\boldsymbol{\Theta}_m^n\|_{1,1}$$

$$\leq \underbrace{\hat{\mu}(m, \boldsymbol{\Theta}_m^*) - \mu(m, \boldsymbol{\Theta}_m^*)}_{\nu_n(m, \boldsymbol{\Theta}_m^*)} + \mu(m, \boldsymbol{\Theta}_m^*) + \lambda_n \|\boldsymbol{\Theta}_m^*\|_{1,1}.$$

By rearranging terms we have

$$\mu(m, \boldsymbol{\Theta}_m^n) - \mu(m, \boldsymbol{\Theta}_m^*) \tag{48}$$

$$\leq |\nu_n(m, \boldsymbol{\Theta}_m^n) - \nu_n(m, \boldsymbol{\Theta}_m^*)| + \lambda_n \|\boldsymbol{\Theta}_m^*\|_{1,1} - \lambda_n \|\boldsymbol{\Theta}_m^n\|_{1,1}$$

$$\leq \tilde{\lambda}_n \|\boldsymbol{\Theta}_m^n - \boldsymbol{\Theta}_m^*\|_{1,1} + \lambda_n \|\boldsymbol{\Theta}_m^*\|_{1,1} - \lambda_n \|\boldsymbol{\Theta}_m^n\|_{1,1}$$

$$\leq \frac{\lambda_n}{2} \|\boldsymbol{\Theta}_m^n - \boldsymbol{\Theta}_m^*\|_{1,1} + \lambda_n \|\boldsymbol{\Theta}_m^*\|_{1,1} - \lambda_n \|\boldsymbol{\Theta}_m^n\|_{1,1}$$

$$= \frac{\lambda_n}{2} \|\boldsymbol{\Theta}_{m,S_m}^n - \boldsymbol{\Theta}_{m,S_m}^*\|_{1,1} + \frac{\lambda_n}{2} \|\boldsymbol{\Theta}_{m,S_m^c}^n - \boldsymbol{\Theta}_{m,S_m^c}^*\|_{1,1}$$

$$+ \lambda_n \|\boldsymbol{\Theta}_{m,S_m}^*\|_{1,1} - \lambda_n \|\boldsymbol{\Theta}_{m,S_m}^n\|_{1,1} - \lambda_n \|\boldsymbol{\Theta}_{m,S_m^c}^n\|_{1,1}$$

$$\leq \frac{3\lambda_n}{2} \|\boldsymbol{\Theta}_{m,S_m}^n - \boldsymbol{\Theta}_{m,S_m}^*\|_{1,1} + \frac{\lambda_n}{2} \|\boldsymbol{\Theta}_{m,S_m^c}^n - \boldsymbol{\Theta}_{m,S_m^c}^*\|_{1,1} - \lambda_n \|\boldsymbol{\Theta}_{m,S_m^c}^n\|_{1,1}$$

$$= \frac{3\lambda_n}{2} \|\boldsymbol{\Theta}_{m,S_m}^n - \boldsymbol{\Theta}_{m,S_m}^*\|_{1,1} - \frac{\lambda_n}{2} \|\boldsymbol{\Theta}_{m,S_m^c}^n - \boldsymbol{\Theta}_{m,S_m^c}^*\|_{1,1}, \tag{49}$$

where the second inequality holds under the event $\mathscr{T}_n$, the third inequality follows by the choice of $\tilde{\lambda}_n = \frac{\lambda_n}{2}$, and the equality uses the property that $\|\boldsymbol{\Theta}_m^n\|_{1,1} = \|\boldsymbol{\Theta}_{m,S_m}^n\|_{1,1} + \|\boldsymbol{\Theta}_{m,S_m^c}^n\|_{1,1}$. Since $\mu(m, \boldsymbol{\Theta}_m^n) \geq \mu(m, \boldsymbol{\Theta}_m^*)$, from (49) we have $\|\boldsymbol{\Theta}_{m,S_m^c}^n - \boldsymbol{\Theta}_{m,S_m^c}^*\|_{1,1} \leq 3\|\boldsymbol{\Theta}_{m,S_m}^n - \boldsymbol{\Theta}_{m,S_m}^*\|_{1,1}$, which means $\boldsymbol{\Theta}_m^n - \boldsymbol{\Theta}_m^* \in \mathbb{C}(S_m)$. Hence, by Assumption 5 we have

$$\psi^2(S_m) \leq s_m \frac{\sum_{k=1}^{K-1} \sum_{\tilde{k}=1}^{K-1} (\boldsymbol{\theta}_k^n - \boldsymbol{\theta}_k^*)^\top \mathbb{E}_{\mathbf{x} \sim \mathcal{X}} [\boldsymbol{\phi}_m(\mathbf{x}) \boldsymbol{\phi}_m(\mathbf{x})^\top] (\boldsymbol{\theta}_{\tilde{k}}^n - \boldsymbol{\theta}_{\tilde{k}}^*)}{\|\boldsymbol{\Theta}_{m,S_m}^n - \boldsymbol{\Theta}_{m,S_m}^*\|_{1,1}^2}. \tag{50}$$

Also, from (49) by adding $\lambda_n \|\boldsymbol{\Theta}_{m,S_m}^n - \boldsymbol{\Theta}_{m,S_m}^*\|_{1,1}$ to both sides and by rearranging we have

$$2\left(\mu(m, \boldsymbol{\Theta}_m^n) - \mu(m, \boldsymbol{\Theta}_m^*)\right) + \underbrace{\lambda_n \|\boldsymbol{\Theta}_{m,S_m^c}^n - \boldsymbol{\Theta}_{m,S_m^c}^*\|_{1,1} + \lambda_n \|\boldsymbol{\Theta}_{m,S_m}^n - \boldsymbol{\Theta}_{m,S_m}^*\|_{1,1}}_{\lambda_n \|\boldsymbol{\Theta}_m^n - \boldsymbol{\Theta}_m^*\|_{1,1}}$$

$$\leq 4\lambda_n \|\boldsymbol{\Theta}_{m,S_m}^n - \boldsymbol{\Theta}_{m,S_m}^*\|_{1,1}. \tag{51}$$

By plugging (50) into (51), we have

$$2\left(\mu(m, \boldsymbol{\Theta}_m^n) - \mu(m, \boldsymbol{\Theta}_m^*)\right) + \lambda_n \|\boldsymbol{\Theta}_m^n - \boldsymbol{\Theta}_m^*\|_{1,1} \tag{52}$$

$$\leq 4\lambda_n \sqrt{\sum_{k,\tilde{k}} (\boldsymbol{\theta}_k^n - \boldsymbol{\theta}_k^*)^\top \mathbb{E}_{\mathbf{x} \sim \mathcal{X}} [\boldsymbol{\phi}_m(\mathbf{x}) \boldsymbol{\phi}_m(\mathbf{x})^\top] (\boldsymbol{\theta}_{\tilde{k}}^n - \boldsymbol{\theta}_{\tilde{k}}^*)} \frac{\sqrt{s_m}}{\psi(S_m)}$$

$$\leq \frac{1}{\kappa_0(K-1)} \sum_{k,\tilde{k}} (\boldsymbol{\theta}_k^n - \boldsymbol{\theta}_k^*)^\top \mathbb{E}_{\mathbf{x} \sim \mathcal{X}} [\boldsymbol{\phi}_m(\mathbf{x}) \boldsymbol{\phi}_m(\mathbf{x})^\top] (\boldsymbol{\theta}_{\tilde{k}}^n - \boldsymbol{\theta}_{\tilde{k}}^*) + 4 \frac{\lambda_n^2 s_m}{\psi^2(S_m)} \kappa_0(K-1), \tag{53}$$

where the last inequality follows by using $4uv \leq u^2 + 4v^2$ with

$$u = \sqrt{\frac{1}{\kappa_0(K-1)} \sum_{k,\tilde{k}} (\boldsymbol{\theta}_k^n - \boldsymbol{\theta}_k^*)^\top \mathbb{E}_{\mathbf{x} \sim \mathcal{X}} [\boldsymbol{\phi}_m(\mathbf{x}) \boldsymbol{\phi}_m(\mathbf{x})^\top] (\boldsymbol{\theta}_{\tilde{k}}^n - \boldsymbol{\theta}_{\tilde{k}}^*)} \text{ and}$$

$$v = \frac{\lambda_n \sqrt{s_m}}{\psi(S_m)} \sqrt{\kappa_0(K-1)}.$$

To continue with the proof, we introduce the following lemma.

**Lemma 18.** *For each $m \in [M]$ with $\boldsymbol{\Theta}_m \in \mathbb{R}^{d_m \times (K-1)}$, the following holds:*

$$\mu(m, \boldsymbol{\Theta}_m) - \mu(m, \boldsymbol{\Theta}_m^*) \geq \frac{1}{\kappa_0(K-1)} \sum_{k, \tilde{k}} (\boldsymbol{\theta}_k^n - \boldsymbol{\theta}_k^*)^\top \mathbb{E}_{\mathbf{x} \sim \mathcal{X}} [\phi_m(\mathbf{x}) \phi_m(\mathbf{x})^\top] (\boldsymbol{\theta}_{\tilde{k}}^n - \boldsymbol{\theta}_{\tilde{k}}^*) . \quad (54)$$

Finally, by Lemma 18, we have from (53)

$$\mu(m, \boldsymbol{\Theta}_m^n) - \mu(m, \boldsymbol{\Theta}_m^*) + \lambda_n \|\boldsymbol{\Theta}_m^n - \boldsymbol{\Theta}_m^*\|_{1,1} \leq 4 \frac{\lambda_n^2 s_m}{\psi^2(S_m)} \kappa_0(K-1) .$$

$\square$

### C.6.5 PROOF OF LEMMA 18

*Proof of Lemma 18.* The model-wise excess risk is

$$\mu(m, \boldsymbol{\Theta}_m) - \mu(m, \boldsymbol{\Theta}_m^*)$$

$$= \mathbb{E}\left[-\sum_{k=1}^{K-1} y_k \phi_m(\mathbf{x})^\top \boldsymbol{\theta}_{m,k} + \log\left(1 + \sum_{k'=1}^{K-1} \exp(\phi_m(\mathbf{x})^\top \boldsymbol{\theta}_{m,k'})\right)\right]$$

$$- \mathbb{E}\left[-\sum_{k=1}^{K-1} y_k \phi_m(\mathbf{x})^\top \boldsymbol{\theta}_{m,k}^* + \log\left(1 + \sum_{k'=1}^{K-1} \exp(\phi_m(\mathbf{x})^\top \boldsymbol{\theta}_{m,k'}^*)\right)\right] .$$

For simple notation, we omit the subscript $m$ everywhere. Also, we denote the model-wise excess risk as $\mathcal{E}(\boldsymbol{\Theta}) := \mu(m, \boldsymbol{\Theta}) - \mu(m, \boldsymbol{\Theta}^*)$.

Viewing each matrix and vector as a tensor, the Taylor expansion of $\mathcal{E}(\boldsymbol{\Theta})$ up to the second order term with $\bar{\boldsymbol{\Theta}} = (\bar{\boldsymbol{\theta}}_1, \cdots, \bar{\boldsymbol{\theta}}_{K-1})$, where for each $\tilde{k} \in [K-1]$, $\bar{\boldsymbol{\theta}}_k = c\boldsymbol{\theta}_k^* + (1-c)\boldsymbol{\theta}_k$ for some $c \in (0, 1)$, gives

$$\mathcal{E}(\boldsymbol{\Theta}) = \mathcal{E}(\boldsymbol{\Theta}^*) + \sum_{k=1}^{K-1} \sum_{j=1}^{d_\phi} [\nabla \mathcal{E}(\boldsymbol{\Theta}^*)]_{k,j} \Delta\theta_{k,j} + \frac{1}{2} \sum_{\tilde{k}=1}^{K-1} \sum_{\tilde{j}=1}^{d_\phi} \sum_{k=1}^{K-1} \sum_{j=1}^{d_\phi} [\nabla^2 \mathcal{E}(\bar{\boldsymbol{\Theta}})]_{k,j,\tilde{k},\tilde{j}} \Delta\theta_{k,j} \Delta\theta_{\tilde{k},\tilde{j}} ,$$

where $\Delta\theta_{k,j}$ is the $j$-th component of $\Delta\boldsymbol{\theta}_k$, $[\nabla \mathcal{E}(\boldsymbol{\Theta}^*)]_{k,j}$ is the $(k, j)$-th component entry of the gradient of $\mathcal{E}(\boldsymbol{\Theta}^*)$, and $[\nabla^2 \mathcal{E}(\bar{\boldsymbol{\Theta}})]_{k,j,\tilde{k},\tilde{j}}$ is the $(\tilde{k}, \tilde{j}, k, j)$-th component of the Hessian of $\mathcal{E}(\bar{\boldsymbol{\Theta}})$.

By the definition of the model-wise excess risk, we have $\mathcal{E}(\boldsymbol{\Theta}^*) = 0$. Also, by Assumption 4, we have $\|\boldsymbol{\Theta}^*\|_F \leq R$, and at that point, $[\nabla \mathcal{E}(\boldsymbol{\Theta}^*)]_{k,j} = 0$ for all $k \in [K-1]$, $j \in [d_\phi]$. Hence, by showing that the second order term of $\mathcal{E}(\boldsymbol{\Theta})$ is larger than the RHS of (54), we conclude the proof.

We organize the second order term of $\mathcal{E}(\boldsymbol{\Theta})$ term and write it in matrix notation. First, we have

$$[\nabla \mathcal{E}(\boldsymbol{\Theta})]_{k,j} = \mathbb{E}\left[-y_k \phi_j + \frac{\exp(\phi(\mathbf{x})^\top \boldsymbol{\theta}_k)\phi_j}{1 + \sum_{k'=1}^{K-1} \exp(\phi(\mathbf{x})^\top \boldsymbol{\theta}_{k'})}\right] ,$$

where $\phi_j$ represents the $j$-th component of $\phi(\mathbf{x})$, and

$$[\nabla^2 \mathcal{E}(\boldsymbol{\Theta})]_{k,j,\tilde{k},\tilde{j}} = \frac{\partial}{\partial\theta_{\tilde{k},\tilde{j}}} [\nabla \mathcal{E}(\boldsymbol{\Theta})]_{k,j}$$

$$= \frac{\partial}{\partial\theta_{\tilde{k},\tilde{j}}} \mathbb{E}\left[-y_k \phi_j + \frac{\exp(\phi(\mathbf{x})^\top \boldsymbol{\theta}_k)\phi_j}{1 + \sum_{k'=1}^{K-1} \exp(\phi(\mathbf{x})^\top \boldsymbol{\theta}_{k'})}\right]$$

$$= \mathbb{E}\left[\frac{\delta_{k,\tilde{k}} \exp(\phi(\mathbf{x})^\top \boldsymbol{\theta}_k)\phi_{\tilde{j}}\phi_j(1 + \sum_{k'=1}^{K-1} \exp(\phi(\mathbf{x})^\top \boldsymbol{\theta}_{k'}))}{(1 + \sum_{k'=1}^{K-1} \exp(\phi(\mathbf{x})^\top \boldsymbol{\theta}_{k'}))^2}\right]$$

$$+ \mathbb{E}\left[\frac{\exp(\phi(\mathbf{x})^\top \boldsymbol{\theta}_k)\phi_j \exp(\phi(\mathbf{x})^\top \boldsymbol{\theta}_{\tilde{k}})\phi_{\tilde{j}}}{(1 + \sum_{k'=1}^{K-1} \exp(\phi(\mathbf{x})^\top \boldsymbol{\theta}_{k'}))^2}\right]$$

$$= \mathbb{E}\left[\left(\delta_{k,\tilde{k}}[\sigma(\phi(\mathbf{x})^\top \boldsymbol{\Theta})]_k - [\sigma(\phi(\mathbf{x})^\top \boldsymbol{\Theta})]_k [\sigma(\phi(\mathbf{x})^\top \boldsymbol{\Theta})]_{\tilde{k}}\right) \phi_j \phi_{\tilde{j}}\right] .$$

Thus,

$$\left[\nabla^2 \mathcal{E}(\bar{\boldsymbol{\Theta}})\right]_{k,j,\tilde{k},\tilde{\jmath}} = \mathbb{E}\left[\left(\delta_{k,\tilde{k}}[\sigma(\bar{\boldsymbol{\Theta}}^\top \phi(\mathbf{x}))]_k - [\sigma(\bar{\boldsymbol{\Theta}}^\top \phi(\mathbf{x}))]_k[\sigma(\bar{\boldsymbol{\Theta}}^\top \phi(\mathbf{x}))]_{\tilde{k}}\right)\phi_j\phi_{\tilde{\jmath}}\right],$$

and finally,

$$\frac{1}{2}\sum_{\tilde{k}=1}^{K-1}\sum_{\tilde{\jmath}=1}^{d_m}\sum_{k=1}^{K-1}\sum_{j=1}^{d_m}\left[\nabla^2\mathcal{E}(\bar{\boldsymbol{\Theta}})\right]_{k,j,\tilde{k},\tilde{\jmath}}\Delta\theta_{k,j}\Delta\theta_{\tilde{k},\tilde{\jmath}}$$

$$= \frac{1}{2}\sum_{\tilde{k}=1}^{K-1}\sum_{k=1}^{K-1}\mathbb{E}\left[\left(\delta_{k,\tilde{k}}[\sigma(\bar{\boldsymbol{\Theta}}^\top\phi(\mathbf{x}))]_k - [\sigma(\bar{\boldsymbol{\Theta}}^\top\phi(\mathbf{x}))]_k[\sigma(\bar{\boldsymbol{\Theta}}^\top\phi(\mathbf{x}))]_{\tilde{k}}\right)\left(\phi(\mathbf{x})^\top\Delta\boldsymbol{\theta}_{\tilde{k}}\right)\left(\phi(\mathbf{x})^\top\Delta\boldsymbol{\theta}_k\right)\right]$$

$$= \frac{1}{2}\mathbb{E}\left[\phi(\mathbf{x})^\top(\Delta\boldsymbol{\Theta})\mathbf{A}(\phi(\mathbf{x}),\bar{\boldsymbol{\Theta}})(\Delta\boldsymbol{\Theta})^\top\phi(\mathbf{x})\right],$$

where $(\Delta\boldsymbol{\Theta}) := [\boldsymbol{\theta}_1 - \boldsymbol{\theta}_1^*, \cdots, \boldsymbol{\theta}_{K-1} - \boldsymbol{\theta}_{K-1}^*]$ and $\mathbf{A}(\phi(\mathbf{x}),\bar{\boldsymbol{\Theta}})$ is the matrix given in Assumption 6,

$$\mathbf{A}(\phi(\mathbf{x}),\boldsymbol{\Theta})(i,j) := [\sigma(\phi(\mathbf{x})^\top\boldsymbol{\Theta})]_i\left(\delta_{i,j} - [\sigma(\phi(\mathbf{x})^\top\boldsymbol{\Theta})]_j\right).$$

On the other hand, we write the RHS of (54) in matrix form as follows:

$$\frac{1}{\kappa_0(K-1)}\sum_{\tilde{k}=1}^{K-1}\sum_{k=1}^{K-1}\mathbb{E}[\Delta\boldsymbol{\theta}_{m,k}^\top\phi(\mathbf{x})\phi(\mathbf{x})^\top\Delta\boldsymbol{\theta}_{m,\tilde{k}}] = \frac{1}{\kappa_0(K-1)}\mathbb{E}\left[\phi(\mathbf{x})^\top(\Delta\boldsymbol{\Theta})\mathbf{J}(\Delta\boldsymbol{\Theta})^\top\phi(\mathbf{x})\right],$$

$$(55)$$

where $\mathbf{J}\in\mathbb{R}^{(K-1)\times(K-1)}$ is the matrix of all ones, i.e., $[\mathbf{J}]_{k,\tilde{k}} = 1$ for all $k,\tilde{k}\in[K-1]$.

Note that $\mathbf{J}$ has eigenvalue $K-1$ with multiplicity 1 and eigenvalue 0 with multiplicity $K-2$. Thus, $\mathbf{J}$ is positive semi-definite, as it is symmetric and only has non-negative eigenvalues.

The end of the proof utilizes Assumption 6. In order to employ Assumption 6, it is necessary to first demonstrate that $\lambda_{\min}(\mathbf{A}(\phi(\mathbf{x}),\bar{\boldsymbol{\Theta}}))$ is at least positive.

**Lemma 19.** *For $\boldsymbol{\Theta}\in\mathbb{R}^{d\times(K-1)}$, let $\bar{\boldsymbol{\Theta}} = c\boldsymbol{\Theta} + (1-c)\boldsymbol{\Theta}^*$ for some $c\in(0,1)$. Then, $\mathbf{A}(\phi(\mathbf{x}),\bar{\boldsymbol{\Theta}})$ is positive definite.*

By Lemma 19, we have $\lambda_{\min}(\mathbf{A}(\phi(\mathbf{x}),\bar{\boldsymbol{\Theta}})) > 0$. Also, as $\mathbf{A}(\phi(\mathbf{x}),\bar{\boldsymbol{\Theta}})$ and $\mathbf{J}$ are both symmetric, we have

$$\mathbf{A}(\phi(\mathbf{x}),\bar{\boldsymbol{\Theta}}) \succeq \lambda_{\min}(\mathbf{A}(\phi(\mathbf{x}),\bar{\boldsymbol{\Theta}}))\mathbf{I}_{K-1} \quad\text{and}\quad \lambda_{\max}(\mathbf{J})\mathbf{I}_{K-1} \succeq \mathbf{J}.$$

Using the above properties, we get

$$\mathcal{E}(\boldsymbol{\Theta}) = \frac{1}{2}\mathbb{E}\left[\phi(\mathbf{x})^\top(\Delta\boldsymbol{\Theta})\mathbf{A}(\phi(\mathbf{x}),\bar{\boldsymbol{\Theta}})(\Delta\boldsymbol{\Theta})^\top\phi(\mathbf{x})\right]$$

$$\geq \frac{1}{2}\lambda_{\min}(\mathbf{A}(\phi(\mathbf{x}),\bar{\boldsymbol{\Theta}}))\mathbb{E}\left[\phi(\mathbf{x})^\top(\Delta\boldsymbol{\Theta})(\Delta\boldsymbol{\Theta})^\top\phi(\mathbf{x})\right]$$

$$\geq \frac{1}{\kappa_0}\mathbb{E}\left[\phi(\mathbf{x})^\top(\Delta\boldsymbol{\Theta})(\Delta\boldsymbol{\Theta})^\top\phi(\mathbf{x})\right]$$

$$= \frac{1}{\kappa_0(K-1)}\lambda_{\max}(\mathbf{J})\mathbb{E}\left[\phi(\mathbf{x})^\top(\Delta\boldsymbol{\Theta})(\Delta\boldsymbol{\Theta})^\top\phi(\mathbf{x})\right]$$

$$\geq \frac{1}{\kappa_0(K-1)}\mathbb{E}\left[\phi(\mathbf{x})^\top(\Delta\boldsymbol{\Theta})\mathbf{J}(\Delta\boldsymbol{\Theta})^\top\phi(\mathbf{x})\right],\qquad(56)$$

where the second inequality comes from Assumption 6 and the equality comes from $\lambda_{\max}(\mathbf{J}) = K-1$. Then, by combining (55) and (56), we conclude the proof. $\qquad\square$

### C.6.6 PROOF OF LEMMA 19

*Proof of Lemma 19.* For simple notation, we use the same notations of Section C.6.5. Furthermore, we denote as $\mathbf{A} := \mathbf{A}(\phi(\mathbf{x}), \bar{\mathbf{\Theta}})$, and $p(k, \mathbf{\Theta}) := [\sigma(\phi(\mathbf{x})^\top \mathbf{\Theta})]_k$.

A symmetric and strictly diagonally dominant matrix having positive diagonal entries is positive definite (Theorem 6.1.10 of Horn & Johnson (2012)). Obviously, $\mathbf{A}$ is symmetric.

Also, all of its diagonal entries $\mathbf{A}(k, k) = p(k, \bar{\mathbf{\Theta}}) - p(k, \bar{\mathbf{\Theta}})^2$ are positive since $p(k, \bar{\mathbf{\Theta}})$ are neither 1 nor 0 for all $k \in [K-1]$. Note that for all $\tilde{k} \neq k$, $p(k, \bar{\mathbf{\Theta}}) = 1$ and $p(\tilde{k}, \bar{\mathbf{\Theta}}) = 0$ only if $\phi(\mathbf{x})^\top \bar{\boldsymbol{\theta}}_k = \infty$ and $\phi(\mathbf{x})^\top \bar{\boldsymbol{\theta}}_{\tilde{k}} \neq \infty$, and $p(k, \bar{\mathbf{\Theta}}) = 0$ while $p(\tilde{k}, \bar{\mathbf{\Theta}}) \neq 1$ only if $\phi(\mathbf{x})^\top \bar{\boldsymbol{\theta}}_k = -\infty$. However, as $\bar{\mathbf{\Theta}}$ lies in line segment between $\mathbf{\Theta}^*$ and $\mathbf{\Theta}$, where $\|\mathbf{\Theta}^*\|_{1,1} \leq R$ and $\mathbf{\Theta} \in \mathbb{R}^{d \times (K-1)}$, at least $-\infty < \bar{\boldsymbol{\theta}}_k < \infty$ for all $k \in [K-1]$. As $\phi(\mathbf{x})$ is bounded by Assumption 4, $\phi(\mathbf{x})^\top \bar{\boldsymbol{\theta}}_k \notin \{-\infty, \infty\}$, and thus, $p(k, \bar{\mathbf{\Theta}}) \notin \{0, 1\}$, for all $k \in [K-1]$.

We finish the proof by showing that $\mathbf{A}$ is strictly diagonally dominant. A strictly diagonal dominant matrix is a square matrix such that for every row (or column), the absolute value of the diagonal entry is strictly larger than the sum of the absolute values of the off diagonal entries. As $\mathbf{A}$ is symmetric, it is suffice to show that $\mathbf{A}$ is strictly diagonally dominant for rows. For $k \in [K-1]$,

$$
\begin{aligned}
\mathbf{A}(k, k) &= p(k, \bar{\mathbf{\Theta}})(1 - p(k, \bar{\mathbf{\Theta}})) \\
&= p(k, \bar{\mathbf{\Theta}}) \left[ \sum_{\tilde{k}=1}^{K} p(\tilde{k}, \bar{\mathbf{\Theta}}) - p(k, \bar{\mathbf{\Theta}}) \right] \\
&= p(k, \bar{\mathbf{\Theta}}) \left[ p(K, \bar{\mathbf{\Theta}}) + \sum_{\substack{\tilde{k} \in [K-1] \\ \tilde{k} \neq k}} p(\tilde{k}, \bar{\mathbf{\Theta}}) \right] \\
&= p(k, \bar{\mathbf{\Theta}}) p(K, \bar{\mathbf{\Theta}}) + \sum_{\substack{\tilde{k} \in [K-1] \\ \tilde{k} \neq k}} p(k, \bar{\mathbf{\Theta}}) p(\tilde{k}, \bar{\mathbf{\Theta}}) \\
&> \sum_{\substack{\tilde{k} \in [K-1] \\ \tilde{k} \neq k}} p(k, \bar{\mathbf{\Theta}}) p(\tilde{k}, \bar{\mathbf{\Theta}}) \\
&= \sum_{\substack{\tilde{k} \in [K-1] \\ \tilde{k} \neq k}} \left| \mathbf{A}(k, \tilde{k}) \right| .
\end{aligned}
$$

$\square$

## D ADDITIONAL EXPERIMENTAL SETTINGS AND RESULTS

All experiments were held in a computing cluster with four NVIDIA GeForce RTX 3090 GPUs, forty Intel(R) Xeon(R) Silver 4210R CPUs, and 187GB of RAM.

### D.1 SUMMARY OF NETWORK-DATASET PAIRS USED IN THE EXPERIMENTS

Summary of the network-dataset pairs used in the experiments are given in Table 4.

|  | Exp. 1 | Exp. 2 | Exp. 3 | Exp. 4 |
|---|---|---|---|---|
| Measure to Demonstrate | Generalization Error | Generalization Error | Accuracy & Sparsity Rate | Loss & Sparsity Rate |
| Dataset for Model Selection | Half of the Training Dataset | $N = 10,000$ | Half of the Training Dataset | Half of the Training Dataset |
| Dataset for Measurement | Entire Training Dataset | Entire Training Dataset | Test Dataset | Test Dataset |
| Used Network & Dataset | All | MLPNet (MNIST) ResNet20 (CIFAR-10) | MLPNet (MNIST) ResNet20 (CIFAR-10) | MLPNetR (Super) |
| Used Benchmarks | - | **CV**, **TS**, **PARC**, **SFDA** | **GM**, **M-FAC**, **CHITA** | **CHITA** |

Table 5: Summary of the objectives and methods of the experiments.

| Network (Dataset) | No. of Params. | Input Dim. | Feature Map Dim. | Training Set Size | Test Set Size |
|---|---|---|---|---|---|
| MLPNet (MNIST) | 30K | 784 | 20 | 60,000 | 10,000 |
| ResNet20 (CIFAR-10) | 200K | 3,072 | 64 | 50,000 | 10,000 |
| MLPNetR (Super) | 4K | 81 | 20 | 17,010 | 4,253 |
| MLPNetR (Cal) | 1K | 8 | 20 | 16,512 | 4,128 |

Table 4: Summary of network-dataset pairs used in the experiments. The number of parameters refer to the ones of the original neural network before applying any pruning method. For $y = \phi(\mathbf{x})^\top \mathbf{w}$, the input dimension is the dimension of $\mathbf{x}$, and the feature map dimension is that of $\phi(\mathbf{x})$.

For MLPNet (MNIST) and ResNet20 (CIFAR-10), we utilize the checkpoints from the official repository of (Benbaki et al., 2023). We use the Superconductivity Data (Super) (Hamidieh, 2018) and California Housing Dataset (Cal) (Pace & Barry, 1997) to train the MLPNetR (Super) and MLPNetR (Cal) respectively. Each dataset was standardized and randomly split into training and test datasets with an 80% to 20% ratio. The mean squared error on the test set is 0.1039 for MLP-NetR (MNIST) and 0.2416 for MLPNetR (Cal) after training. We provide the checkpoints of each network along with the training and test datasets.

### D.2 DETAILS ON THE EXPERIMENTAL SETTINGS

The objectives and methods of the experiments are summarized in Table 5.

**Experiment 1.** Figure 1 is the result of `L1Sel` executing an elimination-selection process and choosing a pruned neural network from $M = 100$ candidate models. We explained how we generated the pruned neural networks in Section 4.1. We used $H = 10$ epochs for each model selection scenario. The epoch lengths $n_1 < \cdots < n_9 < n_{10} = N$ can be set using the optimal method found through experimentation. For classification model selection, we double the epoch length with each increasing epoch. For regression model selection, we increase the epoch length by a constant increment. In both cases, the final epoch uses the entire sample.

We use hyperparameters $\lambda_0$ and $\alpha_0$, named initial regularization parameter and initial confidence radius, and set the regularization parameter and confidence radius for each epoch as $\lambda_{n_h} = \lambda_0/\sqrt{n_h}$ and $\alpha_{n_h} = \alpha_0/\sqrt{n_h}$. If we had complete information of every variable and constant in Theorems 1 and 2, we could set $\lambda_0$ and $\alpha_0$ so that $\lambda_{n_h}$ and $\alpha_{n_h}$ slightly exceed the theoretical values. However, due to the lack of precise values for certain variables and constants, such as $s_{\bar{m}}/\psi^2(S_{\bar{m}})$, we employ conservative estimates for $\alpha_0$ and $\lambda_0$. We set the initial confidence radius $\alpha_0$ as 10 for the classification model selection scenarios, and 8 for the regression model selection scenarios. For the initial regularization parameter $\lambda_0$, we consider the input and/or feature map dimension of each network-dataset pair. The values are set as follows: 20 for MLPNet (MNIST), 60 for ResNet20 (CIFAR-10), 2 for MLPNetR (Super), and 0.5 for MLPNet (Cal).

**Experiment 2.** For comparing our algorithm with other model selection methods, we design a more challenging model selection task. The total number of models in each pool are same, 100, but

this time we generate the candidate pool with $50$ randomly pruned models at only sparsity rates, $50\%$ and $60\%$. Also, we use a smaller sample size of $N = 10,000$, and corresponding to sample size reduction, we adjust the number of epochs to $H = 5$. Other hyperparameters remain unchanged.

**Experiment 3.** As explained in Section 4.3, we utilize three pruning benchmarks, Global $L_1$-Magnitude Pruning (**GM**) provided in the PyTorch library, **M-FAC** (Frantar et al., 2021), and **CHITA** (Benbaki et al., 2023), to create a candidate pool for each network and pruning rate. We generated the pruned networks in a one-shot manner using the official codes of **CHITA** and **M-FAC** without altering any hyperparameters. As a result, the test set accuracy may differ from those reported originally. Since **GM** is deterministic, we gave Gaussian noise to the pruning rate when generating the models.

Given that there are only three pruned models in each pool, the need for elimination is minimal. Thus, we set the number of epochs as $H = 2$. Except for this change, we maintain consistency by using the same total number of samples $N$ (half of the entire training dataset), initial confidence radius $\alpha_0$ and initial regularization parameter $\lambda_0$ as in Experiment 1.

**Experiment 4.** For Experiment 4, we only use one pruned model, MLPNetR (Super) pruned with **CHITA**, for each sparsity rate. Thus, we set the number of epochs as $H = 1$. Except for this, we use the the same $N$, $\alpha_0$ and $\lambda_0$ for MLPNetR (Super) in Experiment 1.

### D.3 ADDITIONAL RESULTS OF EXPERIMENT 1

We report the additional results corresponding to the rest of the random seeds of Experiment 1. The figures show consistent results. As both $\mu(m^N, \mathbf{w}_{m^N}^N)$ (cyan dashed line) and $\mu(m^*, \mathbf{w}_{m^*}^*)$ (purple line) reside in the confidence interval (blue shade) in the figures, the algorithm guarantees a near-optimal selection with high probability.

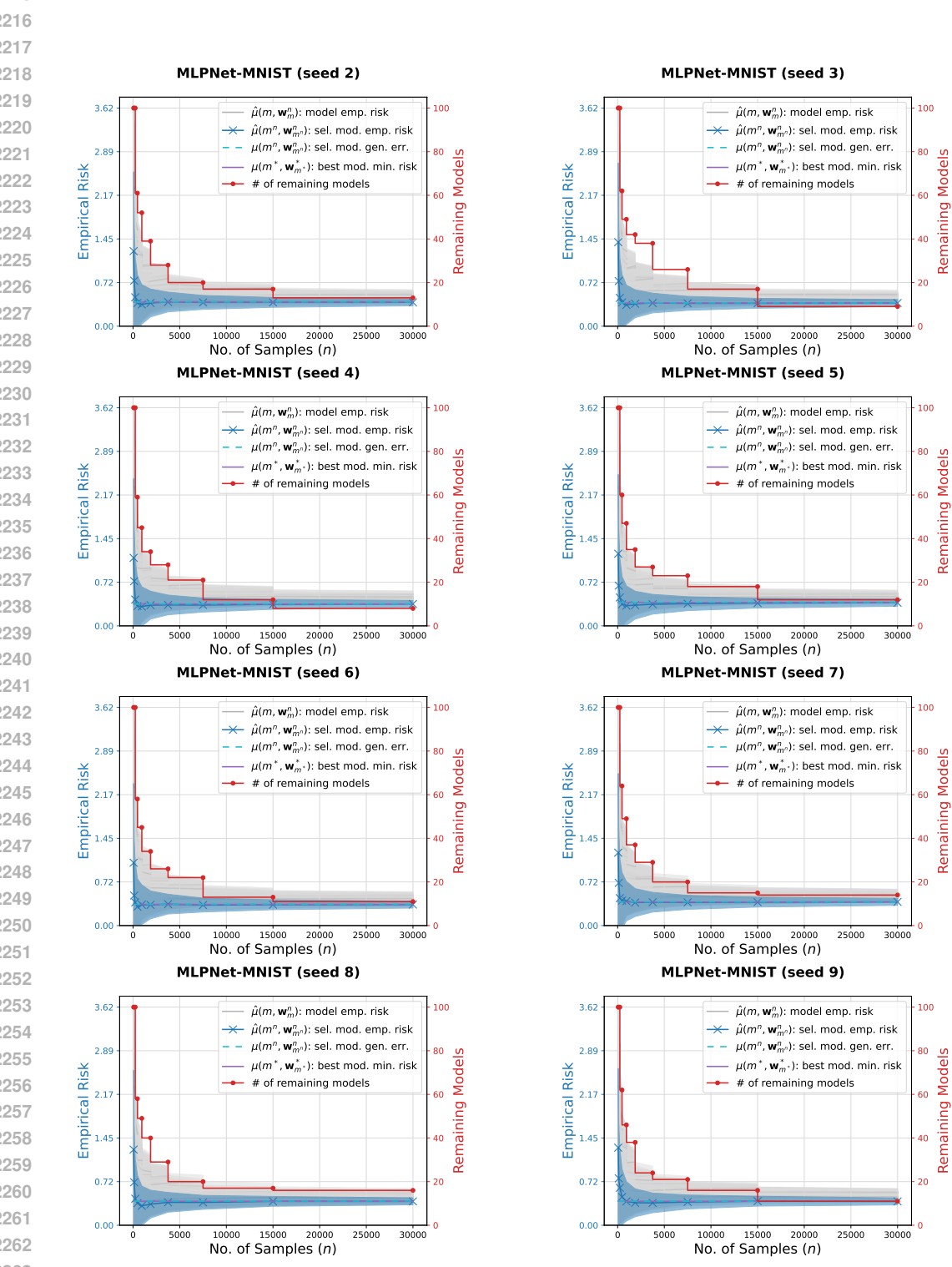

Figure 4: Additional results for MLPNet (MNIST).

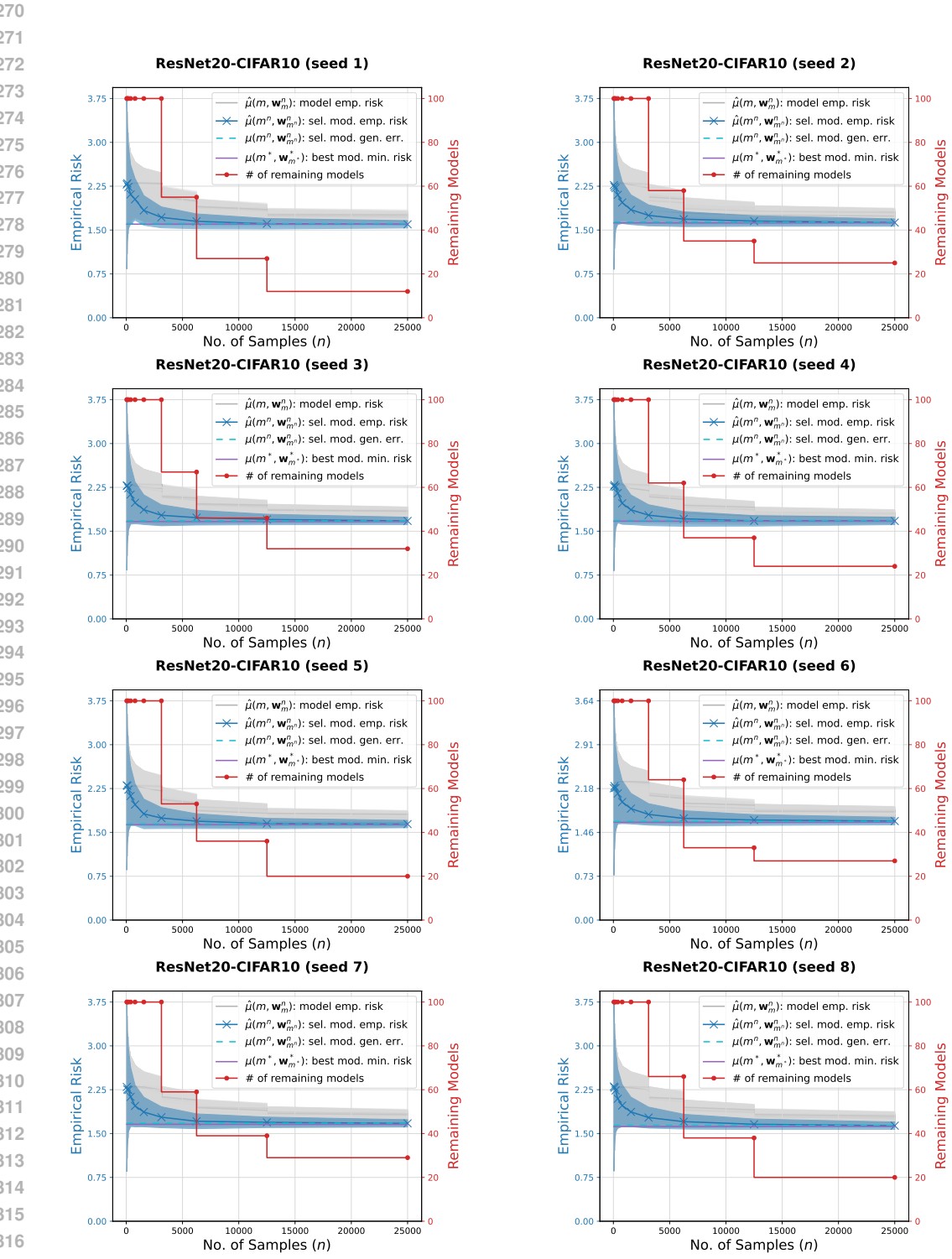

Figure 5: Additional results for ResNet20 (CIFAR10).

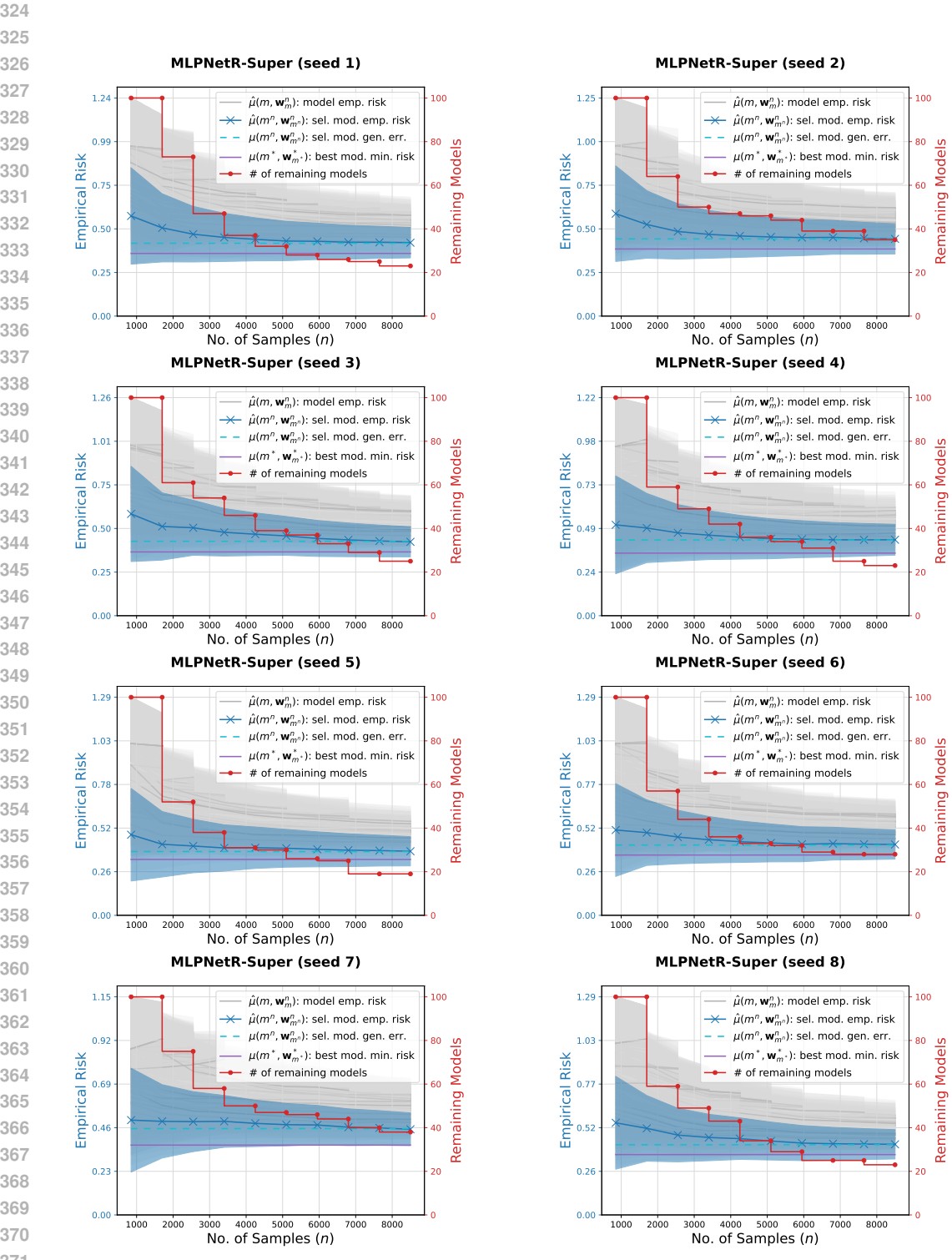

Figure 6: Additional results for MLPNetR (Super).

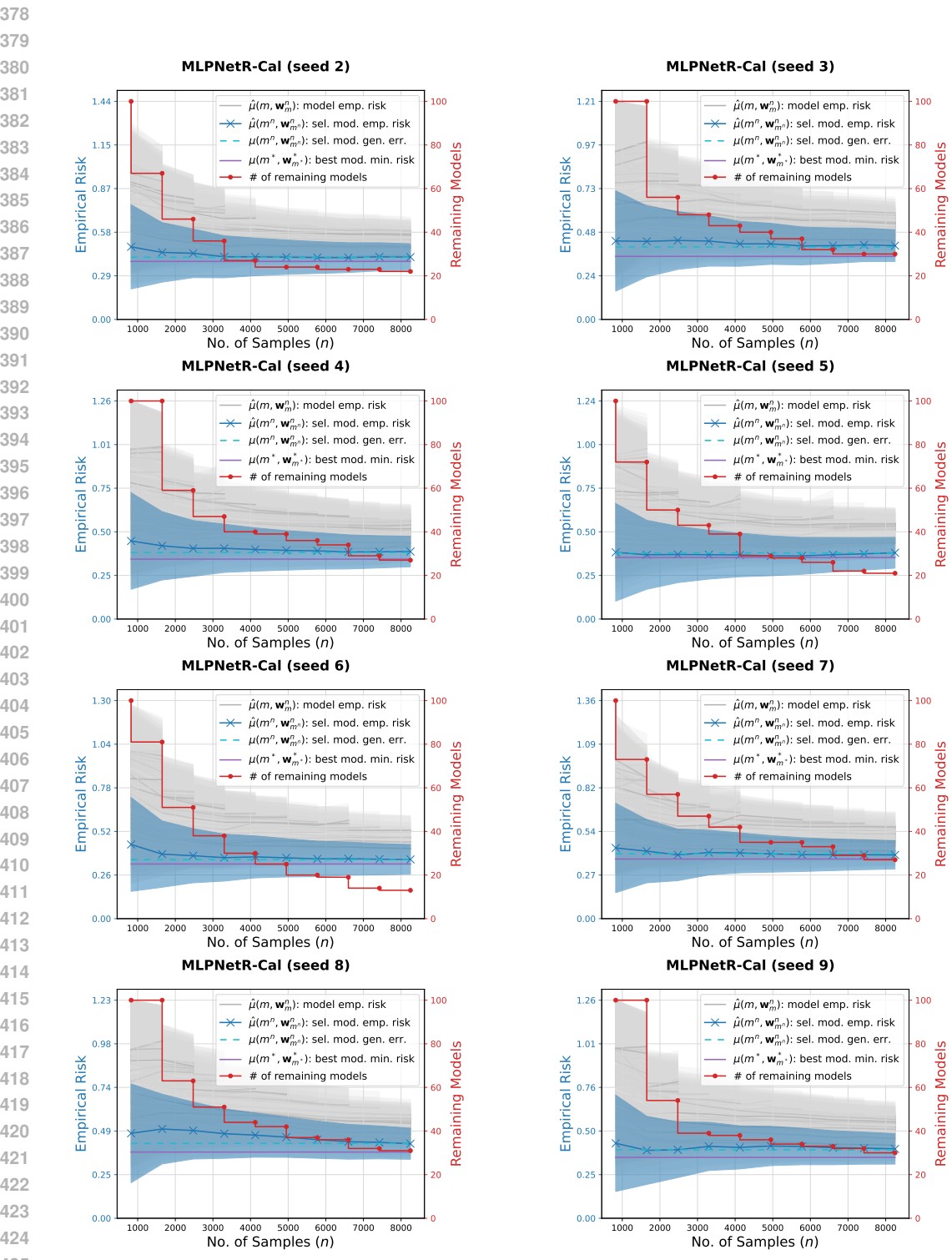

Figure 7: Additional results for MLPNetR (Cal).

