# OpenReview forum: "Sample-Efficient Pruning Model Selection via Lasso"
_ICLR.cc/2026/Conference — ICLR 2026 Conference Withdrawn Submission_

### Official Review · Reviewer_6WXA · 2025-10-29

**Soundness:** 3
**Presentation:** 3
**Contribution:** 2
**Rating:** 6
**Confidence:** 2

**Summary:**

This paper proposes L1Sel, a sample-efficient model selection framework for pruning neural networks under the PAC learning framework. The method combines an elimination-based procedure with L1-regularized selection to identify the best-performing pruned model from a set of candidates using limited data. The authors further introduce L1Sel+, which extends the approach with backward elimination to enhance sparsity by pruning redundant connections in the final layer. Theoretical results establish finite-sample guarantees, and experiments across classification and regression tasks demonstrate improved generalization and efficiency over existing baselines.

**Strengths:**

(1) The paper is built on a clear theoretical foundation. It formulates the pruning model selection problem within the PAC learning framework and provides formal finite-sample guarantees. The theoretical grounding is relatively new in pruning research and offers a valuable perspective for understanding model compression from a generalization viewpoint.

(2) The experimental results are comprehensive and consistent. The method is validated across multiple classification and regression tasks, showing improvements in generalization and sparsity over several recent baselines. These results demonstrate the robustness and versatility of the proposed approach.

**Weaknesses:**

(1) Some abbreviations and technical terms appear without clear definitions or appropriate references upon first use, which may hinder readability for non-specialist readers. A more careful and consistent presentation of notation and terminology would improve clarity.

(2) The L1Sel+ algorithm described in Section 4.4 appears conceptually similar to, or potentially covered by, the prior work DepGraph (Fang et al., 2023). The resemblance between their redundant-parameter elimination mechanisms raises concerns about the novelty and distinct contribution of this component.

Reference:
[1] Fang, Gongfan, et al. DepGraph: Towards Any Structural Pruning. CVPR 2023.

**Questions:**

(1) Could the authors elaborate on the necessity of introducing L1Sel+? If the pruned models can still be further sparsified after being selected by L1Sel, this may indicate that the initial pruning algorithms were not fully effective. In particular, when more advanced parameter grouping techniques such as DepGraph are employed, redundant parameter groups should have already been avoided. The additional backward-elimination step in Sec. 4.4 therefore appears somewhat unimportant, serving mainly as an extra safeguard rather than one of the core contributions.

(2) Could the authors provide more comprehensive ablation studies? It would be valuable to examine how different design choices affect the performance of L1Sel, for example, by integrating different base pruning algorithms, by varying the resource or budget allocation strategy, or by using alternative estimators or elimination criteria. Such analyses would help clarify the robustness and general applicability of the proposed framework.

---

> ### Author Response · Authors · 2025-11-30
>
> Thank you for taking the time to review our paper and for your thoughtful and valuable feedback. We deeply appreciate your recognition of our work and the constructive comments you have provided. Below, we address each of your comments and questions in detail:
>
> ---
>
> [__Presentation__ (W1)]
>
> We will revise the manuscript to ensure that all abbreviations and technical terms are clearly defined at their first appearance and that the notation remains consistent throughout.
> If there are specific terms the reviewer found particularly unclear, we would be happy to incorporate more detailed clarifications in the revision.
>
> ---
>
> [__$\texttt{L1Sel+}$__ (W2 & Q1)]
>
> We appreciate the reference to DepGraph (Fang et al., 2023).
> To clarify, L1Sel+ is not proposed as a pruning algorithm, nor is it meant to compete with structured redundancy-removal methods such as DepGraph (Fang et al., 2023).
> DepGraph performs global, structured pruning by analyzing parameter dependencies across the entire network, whereas L1Sel+ is a post-selection refinement step: once L1Sel identifies a model, we remove only the weights connected to last-layer units that Lasso marks as inactive.
>
> Because of this difference in purpose and scope, L1Sel+ should be viewed as complementary rather than comparable to methods like DepGraph.
> Its role is not to compensate for weaknesses in existing pruning algorithms, but simply to show that—even after applying strong pruning methods—the classifier layer often remains slightly over-parameterized, and a further degree of sparsification can be obtained without degrading performance, as demonstrated in Experiment 4.
> In this sense, L1Sel+ is complementary rather than comparable to structured pruning approaches.
> Once more, we would like to emphasize that the main contribution of our work lies in the model-selection framework itself, which provides finite-sample guarantees for identifying a near-optimal pruned model; L1Sel+ is simply an optional refinement applied after this theoretically grounded selection step.
>
> ---
>
> [__Ablation study__ (Q2)]
>
> We believe that our current experiments already provide evidence that L1Sel performs robustly across different pruning levels and network architectures; however, we agree that further ablations would further illuminate the general applicability of the framework.
>
> Within the remaining revision period, we will explore incorporating additional ablations to the extent possible, particularly focusing on varying the underlying pruning algorithms and resource budgets.
> We believe such extensions would strengthen the empirical picture and we will make an effort to include them if time permits.

---

### Official Review · Reviewer_6hQw · 2025-10-30

**Soundness:** 3
**Presentation:** 2
**Contribution:** 2
**Rating:** 4
**Confidence:** 3

**Summary:**

This manuscript tackles model selection problem among many pruned nets. Instead of the usual cross-validation hand-waving, they formalize selection via a new metric defined as inter-model excess risk $\mu(m, \hat{\mathbf{w}}\_m^n)-\mu(m^\*, \mathbf{w}\_{m^\*}^\*)$. They proposed L1Sel, an elimination-based procedure: iteratively throw out clearly sub-optimal candidates using confidence bounds, then, on the survivors, refit only the last layer with L1 (classic Lasso) and pick the one with the smallest regularized empirical risk. They prove finite-sample PAC results: to get inter-model excess risk $\leq \varepsilon$ with prob $\geq 1-\delta$, regression needs $\tilde{\Omega}((1 / \varepsilon^2) \log (d M))$ under standard boundedness/sub-Gaussian/compatibility assumptions; classification needs $\tilde{\Omega}((1 / \varepsilon^2) \log M)$ (ignore log d). Some numerical comparisons are reported on the MNIST/CIFAR datasets.

**Strengths:**

-  The studied problem setting is important in practice. Against cross‑validation, they get similar or slightly better generalization error and lower runtime on both MNIST/MLPNet and CIFAR‑10/Resnet‑20. The proposed method increases last‑layer sparsity without hurting accuracy (Table 2)



- The presentation of the paper has improved compared to the version I reviewed previously. The construction of events, oracle inequality, and the final error decomposition are standard analysis frameworks and correctly chained in appendix, yielding a rate that recovers textbook Lasso behavior when $M=1$

**Weaknesses:**

- The work presents inter‑model excess risk as *newly introduced* (p. 2-3), but it is just the cross‑class analogue of excess risk; i.e., *regret* to the best model in the pool. It’s perfectly fine to use, but it should not be claimed as pure conceptual novelty towards communities

- Bounded labels $y$ (assumption 1), sub-Gaussian noise defined w.r.t. each model's true parameter $\theta_m^*$ (assumption 2), and compatibility (assumption 3) are standard in Lasso theory for a single model, but simultaneously enforcing them across $M$ heterogeneous, pruned neural feature maps is a big ask in my view; the paper gives no discussion of when this is remotely realistic for CNN/ResNet features

- The work assumes a positive global lower bound on the Fisher factor's smallest eigenvalue along the line segment between $\Theta^{*}$ and $\Theta$. That rules out near-saturated logits and is not defensible for general pruned nets. Would you consider replacing with a e.g. restricted strong convexity or local curvature condition under a margin/overlap assumption, and show how $\kappa_0$ depends on the data distribution?






Identified some typos that not affect the rating,

- p.3 "revisit the some definitions", remove the

- p.6 "at sparsity rate of", use rates

- p. 22 and 33, "Lemam" should be Lemma

- p.35, "Combininig"

**Questions:**

- The comparison to CV (Table 1) ignores nested CV and says CV can favor models that overfit folds, which is true only when one deploy naive CV for selection; the standard remedy could be e.g. nested CV

- On the speed advantage, we don't know whether you warm‑start Lasso across epochs, which could have impacted runtimes

---

> ### Author Response · Authors · 2025-11-30
>
> Thank you for taking the time to review our paper and for your thoughtful and valuable feedback. We deeply appreciate your recognition of our work and the constructive comments you have provided. Below, we address each of your comments and questions in detail:
>
> ---
>
> [__Inter-model excess risk novelty__ (W1)]
>
> We believe that introducing inter-model excess risk is one contribution of our work; however, our goal is not to claim novelty merely for defining a new measure.
> The central contribution lies in using this quantity to extend classical excess-risk analysis to a broader and previously unaddressed problem setup—model selection over multiple models, where each candidate model induces a distinct, model-dependent feature embedding.
>
> In classical single-model settings, excess risk is defined over a shared representation space, making comparisons straightforward.
> In contrast, the pruning-based model selection scenario requires comparing models that reside in heterogeneous compressed feature spaces, a situation for which existing theory does not directly apply.
> Inter-model excess risk therefore serves as a necessary generalization of the classical notion, enabling rigorous analysis in this more complex setting.
>
> Thus, while defining the measure is part of our contribution, the main novelty is in formalizing this new selection problem and developing the theoretical framework needed to analyze it.
> This goes substantially beyond introducing a new metric
>
> ---
>
> [__Assumptions__]
>
> Checking conditions such as compatibility or restricted eigenvalues is known to be NP-hard even in simple linear settings, and existing deep-learning generalization theory similarly relies on such assumptions without attempting explicit verification.
>
> Our use of boundedness, sub-Gaussian noise, and compatibility therefore reflects standard theoretical abstractions rather than strong structural claims about specific architectures.
> They represent the minimal conditions under which finite-sample analysis is tractable, consistent with prior work in high-dimensional regression.
> Moreover, the empirical performance of our framework on real pruned CNNs further demonstrates that these assumptions, while not directly verifiable, capture the relevant behavior of the models sufficiently well to yield accurate and reliable selection in practice.
>
> ---
>
> [__Assumption 6__ (W3)]
>
> Thank you for the thoughtful suggestion.
> We will incorporate this discussion into the revised version of our paper.
>
> To accommodate near-saturated logits, we can relax Assumption 6 to a local margin condition in a neighborhood of $\Theta^*$.
>
> (__Assumption 6*__) Margin condition in a neighborhood of $\Theta^*$
>
> There exists $\gamma \in (0,1/2]$ and $r > 0$ such that, for all
> $\Theta \in \mathcal{B} := \{ \Theta : \| \Theta - \Theta^* \|_{1,1} \}$, we have
> $p_k(x,\Theta) := [\sigma(\phi(x)^\top \Theta]_k \in [\gamma, 1-\gamma]$ for all $x$ and $k$.
>
>
> Under Assumptions 4, 5, and 6*, we can obtain the following lemmas.
>
> [__Lemma A__] Lower bound of $A(x,\Theta)$ in a neighborhood of $\Theta^*$}
>
> Let $\Pi := \{\pi : \pi=(p_1, ..., p_{K-1})^\top,
> p_k \in [\gamma,1-\gamma], k \in [K-1] \}$.
> If $\Theta \in \mathcal{B}$, then
> $\lambda\_{\mathrm{min}}(A(x,\Theta))
> \ge \mathrm{min}\_{\pi \in \Pi}
> \lambda\_{\mathrm{min}}(A(\pi))
> =: \frac{2}{\kappa(\gamma,K)}
> $.
>
> Note that $\kappa(\gamma,K)$ depends only on $\gamma$ and $K$, and not on $x$ or $\Theta$.
>
>
> [__Lemma B__] $\Delta \Theta^n$ lies in the neighborhood of $\Theta^*$}
>
> By choosing $\lambda_n \le \frac{\psi^2(S)}{(4 s \kappa(\gamma, K) (K-1)}$, we ensure that
> $\Delta \Theta^n:=\Theta^n -\Theta^* \in \mathcal{B}$.
>
>
> Using Lemmas A and B, the constant $\kappa_0$ in Lemma 18 can be replaced by $\kappa(\gamma, K)$.
> In particular, $\gamma$ is determined by the margin/overlap properties of the data distribution, so $\kappa(\gamma,K)$ can be interpreted as a function of these distributional characteristics.
> For instance, $\kappa(\gamma, K)$ is expected to become large when $\gamma$ is small.
>
> In summary, modifying Assumption 6 restricts the "non" near-saturation requirement to a small neighborhood of $\Theta^*$, allowing the model to accommodate near-saturated logits outside this region.
> However, if even this neighborhood is near-saturated, the required curvature condition fails to hold, and our theoretical guarantees will no longer apply in that extreme regime.

---

> ### Author Response · Authors · 2025-11-30
>
> [__Comparison to nested CV__ (Q1)]
>
>
> While nested CV may mitigate indeed may mitigate the issue of overfitting to CV folds, it is rarely used in practice in large-scale model selection, as it requires ($O(K^2)$) CV evaluations and becomes computationally prohibitive when the candidate pool is large (e.g., 50–100 pruned models).
> More importantly, nested CV still offers no finite-sample guarantee for identifying the best model, and its high variance persists even with nesting.
> For these reasons, we believe that CV—nested or not—is fundamentally less suitable as a model-selection baseline for our setting.
>
> ---
>
> [__Warm-start__ (Q2)]
>
> We confirm that we did not use warm-starting when solving the Lasso problems across epochs.
> Each L1-regularized regression was initialized independently, without carrying over parameters from previous epochs.
> Therefore, the observed speed advantage does not rely on warm starts, and the runtime comparison remains valid under a fair and consistent implementation across all baselines.

---

### Official Review · Reviewer_G9WW · 2025-11-01

**Soundness:** 1
**Presentation:** 2
**Contribution:** 1
**Rating:** 2
**Confidence:** 4

**Summary:**

This work presents a model selection algorithm that also has pruning effect on the nodes of the last layer and weights directly leading to them. The algorithm, named L1Se1, is a two-stage method: the first is a model selection method with a bound on minimizing ``inter-model excess risk'', and the second step is a lasso on the last layer with removing penultimate weights leading into linear terms removed by lasso. The selection method has a PAC-learning style probabilistic bound on sample efficiency to minimize the inter-model excess risk. Empirical validations provide proof-of-concept workability of this method.

**Strengths:**

Overall, this manuscript is a very readable composition, especially with sufficient explanation of preliminary material in section 2 for the readers without adequate exposure to bounding risk in probabilistic manner. This leads nicely to the point where the authors propose a new metric of ``inter-model excess risk''. The design of the new metric works nicely with the subsequent PAC-style high-probability bounds. This, along with detailed proofs in the appendix, makes the theoretical sections of this work readable to broader audience.

The theoretical setting for deriving sample complexity bounds based on empirical processes, confidence bounds and regret-like objectives give solid foundations, as they are standard approaches enjoyed for decades in learning theory and reinforcement learning literatures.

**Weaknesses:**

The theoretical foundation, despite its solidness, shows age --- the bound-derivation framework itself may be the cause of this work's research direction. In my opinion, this is what constricts the impact of this work, by splitting the whole problem of neural network pruning into the (feature vector set) model selection (from a given candidate pool of potentially pruned models) and the sparse linear model fitting by lasso.

In contrast to the constricted range, the contribution statements of this work, especially in the abstract and the introduction, is misleadingly broad (compared to what is stated in conclusion). For example, 'we propose a novel pruning model selection framework that identifies a near-optimal pruned neural network having low generalization error' can be read in a logically incorrect manner. Also, empirical results, such as the pruning rates, sparsity rates, and accuracies are presented in potentially misleading viewpoints. Additionally, the newly presented metric seems to have some weaknesses to be translated to practical impact in model pruning. I will specify more of these via questions below.

**Questions:**

As far as I understand, this manuscript contains a method to first select a model with high probability bounds using the inter-model excess risk whose definition and computation relies on a given candidate pool (in elimination step of Alg. 1) and then perform lasso on the last layer weights only, followed by pruning the weights leading into the lasso-sparsified last layer nodes (in 'selection' step of Alg. 1).

My first question is on the practical meaning of theoretical correctness from the elimination step. The probabilistic correctness of the elimination step seems to depend on not only the sample size $N$ but also other factors, such as containing good enough candidates in the initial model pool. Empirical validations were made on 100 models in the pool. Definition of inter-model excess risk seems to pick the best-among-in-the-given-pool. How would this translate to actual application of the proposed method (e.g. model pruning)?

Second question extends from this perspective, on the meaning of the important terms 'sparsity rates' and 'pruning rates' used throughout the experiment section. For example, what kind of models are generated when '[you] generate
20 randomly pruned classification models at sparsity rates of 50%, 60%, 70%, 80%, and 90%'? Are these models generated by other methods to fill in the candidate pool? On the other hand, what are the 'pruning rate' in Tables 2 and 3? How are the rates relevant to benchmark baselines as well as the proposed algorithm?

Third question is on the scope of sample-efficiency in the proposed framework? I think this is a serious question as the title starts as 'Sample efficient pruning model selection [...]'. There seems to be two bounds -- one on the risk itself, polynomial in the training sample size, and one on the sample complexity of algorithm logarithmic on model pool size. There seem to be lots of practically important terms impacting the bound, for example, the max epoch count $H$ and corresponding schedule of regularization parameters and confidence radii. Is there any prescient goldilocks requirements on setting those tunable parameters for this method to enjoy practical benefits? Additionally, I am afraid that the title and contribution statements overstate/mislead as if the pruning aspect contributing to sample efficiency. It seems to me that this algorithm uses lasso to 1) create sparse model in the last layer with the goal of making some weights in penultimate layer ineffective (hence limiting pruning targets up to penultimate layers), and 2) increase performance by sparse model fitting which is direct effect of lasso as well.

Last question: what are the limitation of the proposed framework as a pruning method? One aspect I noticed is that its native pruning capacity is on the last layer only (which includes incoming edges to the last layer nodes, applicable to both Alg. 1 and Alg. 2). The proposed method may be used to have broader model-wide by using other pruning methods to provide candidate pools (or use randomly pruned models as presented in the empirical validations), but this does not mean the proposed method may come short of becoming an effective standalone pruning method, especially when the target models are models with depth larger than 2 or 3. Also, I wonder using 100 random pruned models as a candidate pool and the selected benchmark pruning methods show useful/valid comparisons. Potential of pruning method should also contain how well it prunes a whole model from scratch, rather than sparse-fitting given (semi-pruned) models to boost performance and pluck more last-layer incoming weights and gain boost in pruning ratio -- the results in Table 2 look very biased.

---

> ### Author Response · Authors · 2025-11-30
>
> Thank you for taking the time to review our paper and for your thoughtful and valuable feedback. We deeply appreciate your recognition of our work and the constructive comments you have provided. Below, we address each of your comments and questions in detail:
>
> ---
>
> [__Theoretical foundation__ (W1)]
>
> We respectfully disagree with the assessment that our theoretical foundation "shows age" or that its decomposition limits the impact.
>
> First, while our analysis leverages techniques from existing Lasso and excess risk theory, our key contribution lies in extending these tools to a novel and practically relevant setting: selecting among pruned neural networks with model-dependent feature embeddings.
> Unlike the standard single-model Lasso setting (i.e., $M = 1$), our framework must handle inter-model generalization across a structured and heterogeneous model pool—a scenario not addressed in prior theory.
> We believe that this adaptation of established theory to a modern and underexplored problem is both novel and necessary, rather than outdated.
>
> Importantly, we clarify that our goal is not to propose a new pruning method, but rather to address the following underexplored question:
> "_Given multiple pruned models—possibly obtained via different techniques or sparsity levels—how can we select the one with the best generalization, with theoretical guarantees_?"
>
> This is a practical and increasingly relevant challenge in modern pruning workflows, yet no prior method offers provable sample efficiency for solving it.
> Our framework fills this gap by providing a solution with formal guarantees and strong empirical performance.
> Thus, the decomposition into selection and sparse refitting is not a simplification, but a principled abstraction that enables both rigorous theoretical analysis and practical utility.
>
> ---
>
> [__Contribution statements__ (W2)]
>
> We would like to clarify that the goal of our framework is not to propose a new pruning method, but to present a theoretically grounded model selection framework that selects from a pool of pruned models with provable generalization guarantees.
> As has been stated consistently throughout the paper, we believe that the main contributions and the results are well aligned with this stated goal.
> Nonetheless, we appreciate the reviewer’s feedback and will revise the abstract and introduction to reflect this scope more precisely.
> We address the other specific points in more detail below.
>
> ---
>
> [__Actual application of the proposed method__ (Q1)]
>
> First, we clarify that the role of the elimination step is primarily computational.
> Our theoretical results ensure that even without the elimination step, directly applying the selection step—i.e., performing L1-regularized fitting on all models in the pool—would still identify a near-optimal model with high probability.
> However, this naive approach requires running L1 fitting for every candidate model, which can be computationally expensive when the pool size is large.
>
> The elimination step is therefore designed to filter out clearly suboptimal models early, using only a small number of samples.
> As supported by our theory (Lemmas 2 and 12) and our experiments, this filtering does not jeopardize the retention of the best model.
> Instead, it significantly improves runtime efficiency by avoiding unnecessary L1-regularized fitting on inferior candidates.
> Thus, the practical purpose of the elimination step is not to broaden the theoretical applicability, but to reduce computation while preserving the same high-probability guarantee.
>
> Regarding the reviewer’s question about its application to model pruning, we reiterate that our framework is not intended as an alternative pruning algorithm.
> Rather, it provides a theoretically grounded guideline for selecting a generalizable pruning technique for a given dataset.
> Furthermore, as demonstrated in Experiment 4, once a model is selected, our L1-based refitting can often make the model even sparser without accuracy loss, suggesting that our framework may also serve as a useful post-pruning refinement tool when applied to existing pruning methods.
>
> ---
>
> [__Pruning rate__ (Q2)]
>
> The terms sparsity rate and pruning rate refer to the proportion of weights in a neural network that are set to zero, following common usage in the pruning literature (e.g., Han et al., 2015, Benbaki et al. 2021).
> In Experiments 1 and 2, "random pruning" means that, for a pretrained network, we randomly zero out weights to match a target sparsity level; these models are used solely to construct a diverse candidate pool for evaluating our model-selection framework.
>
> The "pruning rate" reported in Tables 2 and 3 corresponds to precisely this overall sparsity level for each model, and the same definition is applied consistently across all baselines and our method to ensure fair comparison.
>
> We agree that these definitions can be made clearer, and we will include precise formal descriptions in the revised manuscript for clarity.

---

> ### Author Response · Authors · 2025-11-30
>
> [__Sample efficiency__ (Q3)]
>
> As in the broader statistical learning literature, the term sample efficiency in our paper refers to the fact that our model-selection framework identifies a near-optimal model using a __finite number of samples__, rather than requiring asymptotically large data.
> In particular, one of the main points we wished to highlight is that the number of samples needed to discriminate the optimal model grows only logarithmically in the pool size $M$, even when many pruned candidates are available for a given dataset and task.
> This property is made explicit in Theorems 1 and 2, which guarantee correct selection under appropriately chosen algorithmic parameters.
>
> In practical applications, some problem-dependent quantities—such as the variance of the observation noise—are typically not observable.
> As discussed in our response to Q2 of Reviewer oCNt, such cases can be handled by introducing tunable hyperparameters (e.g., $\lambda_0$ and $\alpha_0$), and our experiments show that even conservative choices of these parameters lead to robust performance across diverse settings.
> Please refer to Appendix D.2 for details.
> We believe this is consistent with how theoretically grounded algorithms are typically validated empirically in statistical learning literature: theoretical bounds require certain idealized conditions, while practical robustness is demonstrated experimentally.
>
> Finally, we would like to clarify that we do not claim that pruning itself contributes to sample efficiency.
> As stated consistently throughout the paper, our goal is not to design a new pruning method, but to __propose a model selection framework that identifies the most generalizable model from a set of already-pruned neural networks__, and to characterize the sample complexity required for this selection task.
> Our theoretical results and empirical studies are aligned with this intended scope.
>
> ---
>
> [__Limitation__ (Q4)]
>
> We would like to emphasize that our framework is not intended to function as a standalone pruning method, but rather as a __model selection procedure__ that chooses the most generalizable model from a pool of pruned neural networks.
> From this perspective, the fact that our Lasso-based refinement sparsifies the last layer and penultimate layer should be viewed not as a limitation, but as a byproduct of the selection procedure—its role is to refine the chosen model rather than to perform full-network pruning.
>
> As demonstrated in Experiment 4, even when the candidate pool consists of one model produced by SOTA pruning techniques, our L1-based refitting can further increase sparsity without any loss in performance.
> This supports our claim that the proposed framework is complementary to existing pruning methods: it can further improve the sparsity level of the selected model while preserving its accuracy.

---

### Official Review · Reviewer_oCNt · 2025-11-01

**Soundness:** 3
**Presentation:** 2
**Contribution:** 2
**Rating:** 2
**Confidence:** 3

**Summary:**

This paper studies the problem of selecting and fine-tuning one model from a pool of pretrained and pruned neural networks. Each candidate model is treated as a fixed pruned backbone with a trainable linear head. The authors define an inter-model excess risk to formalize how far a selected model's generalization error is from that of the best candidate, and they propose an elimination-based algorithm, L1Sel, which uses L1-regularized ERM to iteratively discard suboptimal models. The paper presents theoretical guarantees on the sample complexity of achieving a target inter-model excess risk, with dependence logarithmic in the number of candidate models, and presents empirical results on several small-scale datasets.

**Strengths:**

The formulation of model selection across a pool of pruned networks as a statistical learning problem is conceptually clean. The inter-model excess risk provides a coherent metric for comparing generalization performance across candidates. Based on this theoretical framework, the authors provide finite-sample bounds and a sample complexity result that scales only logarithmically with the number of models.

The proposed method L1Sel combined with L1 regularization is straightforward simple and a scalable approach to Neural Network model selection is an interesting direction.

**Weaknesses:**

The motivation and experimental setup do not align with realistic pruning and fine-tuning pipelines. The paper assumes a scenario where a large pool of pretrained and pruned networks are selected by training only a linear head—which is not how pruning is typically used. In practice, obtaining a pruned model is typically a two-step process: we start from dense pretrained models and perform pruning as part of fine-tuning, rather than choosing among already-pruned ones. Moreover, "fine-tuning" here reduces to linear probing, which is known to be inferior to full-model fine-tuning in most realistic cases. Also in the context of linear probing, it is uncommon to prune the last linear layer, which further questions the relevance of this setup.

Almost all theoretical content is deferred to the appendix, and the main body only contains one theorem, which is for regression, even though most experiments are conducted on classification tasks. As a result, it is difficult to assess what is actually novel beyond standard Lasso model-selection analysis. If the main contribution is theoretical, it should be articulated clearly in the paper body, including a discussion of how it extends or differs from known results.

The experiments are conducted on relatively simple datasets (MNIST, CIFAR-10, small regression tasks) with small models, where most approaches already perform well. These settings do not convincingly demonstrate the value of the proposed method. More challenging datasets and larger, modern architectures are needed to make the empirical case. Furthermore, the paper uses random pruning to generate model pools. This produces a wide range of model qualities, making the selection problem artificially easy. The paper should evaluate on pools of realistic, high-quality pruned models obtained by state-of-the-art structured pruning techniques to assess whether L1Sel provides any meaningful advantage. Baselines such as CV-5, CV-10, TS, PARC, and SFDA are only listed in tables but not explained in the main text. Readers unfamiliar with these methods cannot interpret what they do or why they are relevant comparators. In general, the related work section has to be discussed in the main text.

**Questions:**

- Can you provide a clear high-level description of your main theorem in the main text and explain how it differs from standard Lasso excess risk bounds?

- How sensitive is the algorithm to hyperparameters such as the epoch schedule, $\lambda$, and $\alpha$? Any heuristics or tuning strategies?

- Can you evaluate L1Sel on realistic candidate pools from actual structured pruning methods or larger pretrained models?

---

> ### Author Response · Authors · 2025-11-30
>
> Thank you for taking the time to review our paper and for your thoughtful and valuable feedback. We deeply appreciate your recognition of our work and the constructive comments you have provided. Below, we address each of your comments and questions in detail:
>
> ---
> [__Problem setup__ (W1)]
>
> We would like to clarify that our framework is not a new pruning algorithm, but rather a _model selection method_ that chooses, from a pool of pretrained and pruned networks, the one that is both (possibly) sparse and highly generalizable.
> Hence, our framework should be interpreted as a _sample-efficient model selection strategy_, rather than as an alternative pruning technique.
>
> Consider a practical scenario where several pruning methods are applied to a pretrained model under different sparsity constraints.
> How can we determine which pruned model to deploy for a given dataset, and can we guarantee that the selected one is near-optimal in terms of generalization?
> Our method provides a principled criterion for selecting such a model, with _finite-sample generalization guarantees_.
>
> Moreover, by refitting the last layer using Lasso, we demonstrate that the chosen model can often be made even sparser without performance degradation, thereby enhancing both efficiency and interpretability.
> In summary, our contribution lies in theoretical model selection for pruned networks, rather than in redefining the pruning process itself.
>
> ---
>
> [__Configurations, high-level description of main theorem__ (W2 \& Q1)]
>
> Due to space constraints, the main text presents the regression version of our theorem because it offers the clearest high-level exposition, while the full theory—including the classification setting used in our experiments—is developed in detail in the appendix.
> Both results follow a similar analytical structure; the appendix provides the complete formal treatment, while the main text focuses on the clear representations.
>
> At a high level, our main theorem extends classical single-model Lasso excess-risk bounds to a multiple-model, pruning-based model selection setting and shows that L1Sel selects a near-optimal pruned model with finite-sample guarantees and only logarithmic dependence on the pool size $M$.
> Compared with standard Lasso analysis, the main differences are:
>
> 1. __From a single model to multiple pruned models with heterogeneous feature maps__.
>
>     Classical Lasso theory controls the excess risk of a single model with a fixed feature map.
>     In our setting, each pruned network $m \in [M]$ induces its own feature embedding $\phi_m(x)$, so we must control inter-model excess risk across a pool of heterogeneous pruned architectures.
>     This requires deriving Lasso-type estimation and concentration bounds that hold uniformly over all models and epochs, and then translating these into a sample-complexity bound that depends only logarithmically on $M$.
>
> 2. __Analyzing the elimination procedure while ensuring the best model survives with high probability__.
>
>     Standard Lasso bounds do not involve any sequential elimination or model-selection dynamics—they only quantify how well a given model generalizes.
>     In contrast, our theorem analyzes the full elimination-based algorithm: at each epoch, inferior models are removed based on Lasso-driven empirical risks and confidence radii, and we must show that, with high probability, the best model is never mistakenly eliminated while suboptimal models are progressively discarded.
>     This requires a careful high-probability control of the ``good event'' under which all Lasso estimates and risk comparisons are simultaneously accurate across models and epochs.
>
> Finally, when $M = 1$, the our analysis reduces to a standard single-model Lasso excess-risk bound.
> In this sense, Theorem 1 provides a generalized analysis framework that strictly contains the usual single-model Lasso theory as a special case, while handling the more challenging multi-model selection problem arising from pruned neural networks.
> In the revised version, we will add a discussion in the main text to clearly explain these differences from standard single-model Lasso analysis.

---

> ### Author Response · Authors · 2025-11-30
>
> [__Experiments__ (W3 \& Q3)]
>
> We would like to clarify that the models and datasets used in our experiments follow the settings adopted in prior SOTA pruning literature (Frantar et a., 2021, Benbaki et al., 2023).
> As we emphasize above, our primary goal is not to propose a new pruning technique, but rather to establish a __theoretically grounded and sample-efficient model selection framework__, supported by __finite-sample generalization guarantees__.
> That said, we are eager to conduct additional experiments on larger-scale models and datasets, and we will make our best effort to include the results within the remaining discussion period.
>
> Regarding the use of random pruning in Experiment 1, we would like to clarify that this choice was made to quickly and efficiently generate a large and diverse pool of pruned models for validating the selection performance of our framework.
> It was not intended to artificially simplify the selection task—indeed, whether random pruning inherently leads to an "easier" selection problem remains an open question and is not trivial to assess.
> Importantly, regardless of the absolute difficulty of the selection task, Experiments 1 and 2 demonstrate that our method consistently outperforms standard baselines in terms of runtime efficiency while also enjoying finite-sample generalization guarantees.
> Furthermore, in Experiment 3, we validated our method on model pools constructed using SOTA pruning techniques, showing that the proposed method can successfully identify near-optimal models and further sparsify them without performance degradation.
>
> We will revise the manuscript to include concise baseline descriptions and clarify their relevance in the main text.
>
> ---
>
> [__Hyperparameters__ (Q2)]
>
> As long as the hyperparameters $\alpha$ and $\lambda$ are chosen in accordance with the guidance provided in Theorems 1 and 2, the elimination step is guaranteed—with high probability—not to discard the best model, as established in Lemmas 2 and 12.
> In practice, however, since certain problem-dependent parameters are unknown, we adopt a data-driven schedule for $\lambda$ and $\alpha$ by introducing initial values $\lambda_0$ and $\alpha_0$.
> Specifically, at each epoch, we use the update rules: $\lambda_{n_h} = \frac{\lambda_0}{\sqrt{n_h}}, \alpha_{n_h} = \frac{\alpha_0}{\sqrt{n_h}}$.
> For example, in a classification setting with $M = 100$ models, $H = 10$ epochs, and $K = 10$ classes,
> Theorem 2 (see line 1710 in the appendix) suggests that the required $B = \mathcal{O}(\log^2 K \log(MH)))$,
> which implies that $\alpha_0$ can be chosen polylogarithmically in $M, H$, and $K$.
> Based on this reasoning, we set $\alpha_0 = 10$ in our classification experiment.
> Although a conservative choice of $\alpha_0$ may reduce the number of models eliminated per epoch, it provides better assurance that the optimal model is retained throughout the elimination process.
> Moreover, since $\alpha_{n_h}$ naturally decreases over time, even larger initial values do not hinder convergence.
> As confirmed empirically in our experiments, the algorithm consistently selects a near-optimal model across epochs under this heuristic schedule.
> Please refer to Appendix D.2 for more details regarding hyperparameters.

---

### Note · Authors · 2025-12-11

**Comment:**

After careful consideration, we have decided to formally withdraw our manuscript. We would like to express our sincere gratitude to the reviewers for their time, effort, and valuable feedback. The suggestions provided have been insightful, and we intend to incorporate many of these recommendations into our revised work. We deeply appreciate your understanding and look forward to sharing our work in the future.

**Withdrawal Confirmation:**

I have read and agree with the venue's withdrawal policy on behalf of myself and my co-authors.